# Sensitivities of subgrid-scale physics schemes, meteorological forcing, and topographic radiation in atmosphere-through-bedrock integrated process models: A case study in the Upper Colorado River Basin

Zexuan Xu[1], Erica R. Siirila-Woodburn[1], Alan M. Rhoades[1], Daniel Feldman[1]

[1] Earth and Environmental Sciences Area, Lawrence Berkeley National Laboratory, United States

*Correspondence to:* Zexuan Xu (zexuanxu@lbl.gov)

**Abstract.**

Mountain hydrology is controlled by interacting processes extending from the atmosphere through the bedrock. Integrated process models (IPM), one of the main tools needed to interpret observations and refine conceptual models of the mountainous water cycle, require meteorological forcing that simulates the atmospheric process to predict hydroclimate then subsequently impacts surface-subsurface hydrology. Complex terrain and extreme spatial heterogeneity in mountainous environments drive uncertainty in several key considerations in IPM configurations, and require further quantification and sensitivity analyses. Here, we present an IPM using the Weather Research and Forecasting (WRF) model which force an integrated hydrologic model, ParFlow-CLM, implemented over a domain centered over the East River Watershed (ERW), located in the Upper Colorado River Basin (UCRB). The ERW is a heavily-instrumented 300 $km^2$ region in the headwaters of the UCRB near Crested Butte, CO, with a growing atmosphere-through-bedrock observation network. Through a series of experiments in water year 2019 (WY19), we use four meteorological forcings derived from commonly used reanalysis datasets, three subgrid-scale physics scheme configurations in WRF, and two terrain shading options within WRF to test the relative importance of these experimental design choices on key hydrometeorological metrics including precipitation, snowpack, as well as evapotranspiration, groundwater storage, and discharge simulated by the ParFlow-CLM. Our hypothesis is that uncertainty from synoptic-scale forcings produce a much larger spread in surface-through-subsurface hydrologic fields than subgrid-scale physics scheme choice. Results reveal that WRF subgrid-scale physics configuration lead to larger spatiotemporal variance in simulated hydrometeorological conditions, whereas variance across meteorological forcing with common subgrid-scale physics configurations is more spatiotemporally constrained. Despite reasonably simulating precipitation, a delay in simulated discharge peak is due to a systematic cold bias across WRF simulations, suggested the need for bias correction. Discharge shows greater variance in response to the WRF simulations across subgrid-scale physics schemes (26%) rather than meteorological forcing (6%). Topographic radiation option has minor effects on the watershed-average hydrometeorological processes, but adds profound spatial heterogeneity to local energy budgets (+/-30 $W/m^2$ in shortwave radiation and 1 K air

temperature differences in late summer).  This is the first presentation of sensitivity analyses that provide support to  help guide the scientific community to develop observational constraints on atmosphere-through-bedrock processes and their interactions.

## 1.   Introduction

An improved predictive understanding of watershed dynamics and response to perturbations is particularly important for mountainous watersheds due to the multitude of natural services they provide even while those services are highly vulnerable to anthropogenic and natural environmental change (Hubbard et al., 2018; Siirila-Woodburn et al., 2021). The Upper Colorado River Basin (UCRB), which supports 40 million people and ecosystems, has experienced major hydrological change in recent decades (James et al., 2014). Discharge has decreased by ~9.3% per degree Celsius of warming, due to processes extending from the atmosphere through the subsurface (Milly and Dunne, 2020). Drought is common to the region, however, the current multi-decade drought is unprecedented in at least the last 1200 years (Williams et al., 2022). To better estimate how aridification of the UCRB might continue, processes that shape the water cycle in this region must be considered holistically, including atmospheric processes such as large-scale vapor transport, precipitation and radiation, land surface processes such as evapotranspiration and snowpack metamorphosis, and surface-through-subsurface hydrological processes. Atmospheric and land surface processes all interact and influence river discharge through riverine processes, infiltration, and subsurface flow and storage, but their impact varies depending on the temporal and spatial scales of analysis (Siirila-Woodburn et al., 2021). Unfortunately, there is a dearth of observational data that can constrain these processes at their respective scales which has resulted in persistent model simulation biases in the predictive the mountainous hydrologic cycle with direct implications for water resource management (Sturm et al., 2017; Rhoades et al., 2018a,b,c; Xu et al., 2019). Lundquist et al. (2019) highlighted that calibrated models, which themselves have numerous deficiencies, have likely outpaced the skill of observationally-based gridded products in advancing the understanding of the integrated mountainous hydrologic cycle. A wide range of physical based and statistical models have been used over the complex terrain of the western U.S. For example, Alder et al. (2019) and Rahimi et al. (2022) have evaluated the choice of downscaled climate data and the sensitivities of grid resolution. Buban et al. (2022) also investigated the use of PRISM as a reference dataset to assess climate model performance. Observational campaigns, combined with coordinated modeling activities, represent a potential path forward towards enhancing our predictive understanding of the hydrologic cycle in complex terrain and, ultimately, advancing model development that can better aid water resource management (Lundquist et al, 2019; Feldman et al., 2021).

Here, we explore how modeling activities can best support that path forward. Process models provide an essential tool for quantifying linear and non-linear interacting processes across spatiotemporal scales that arise in mountains and can help to fill observational gaps. However, the processes that are represented in these process models are a mixture of fundamental physics and subgrid-scale parameterizations, many of which were not developed with a focus on performance in mountainous environments, and/or are based on decades-old field and laboratory data that do not adequately capture the range of

environmental conditions over which those processes occur. Advances in process modeling in complex terrain must recognize connections between processes in the atmosphere, at the surface and in the subsurface. At the same time, making connections between processes across the atmosphere-through-bedrock continuum is highly non-trivial (Meixner et al., 2016, Zhuang et al., 2022). Furthermore, snow processes must be resolved at much finer scales than atmospheric processes, such that snow process investigations and accurate snow process modeling requires high-resolution downscaling of WRF (e.g. Winstral and Marks, 2014). Cross-scale interactions in complex terrain are challenging to resolve at their native scales with currently available advanced computing resources (Siirila-Woodburn et al., 2021). While discipline-specific process models, such as those used to explore and predict atmospheric or subsurface processes have advanced scientific understanding in a myriad of ways through sustained engagement with extensive user communities (Gutowski et al., 2020), Integrated Process Models (IPMs), in which these discipline-specific process models are integrated, are relatively novel and are still being vetted for various scientific applications in complex terrain.

Zhang et al. (2016) and Davison et al. (2017) demonstrated the utility of coupling process models built to explore discipline-specific processes as a mechanism to advance interdisciplinary research. Furthermore, Camera et al (2020) discussed the one-way vs. two-way coupling of IPM to understand process interactions in the mountainous hydrologic cycle. The capabilities and details of the IPM have been discussed in a series of findings. For example, Maina et al. (2020) explored how the horizontal resolution of atmospheric forcing datasets (40 km to 0.5 km) in the Cosumnes River watershed, California, simulated by a widely-used regional climate model (Weather Research and Forecasting (WRF; Powers et al., 2017), result in differences in surface and subsurface hydrologic metrics when used to force the integrated hydrologic model (ParFlow-CLM; Ashby and Falgout, 1996; Jones and Woodward, 2001; Maxwell, 2013, Maxwell et al., 2015), which has been widely applied in the UCRB (Maina et al., 2022; Foster and Maxwell, 2018; Pribulick et al., 2016). We expand upon those various sensitivity analyses in this study, including the influences of large-scale meteorological forcing and subgrid-scale physics schemes choice on the surface-through-subsurface response of the integrated hydrologic model. The goal of this work is to provide the mountain hydrology research community with assessed several literatures supported configurations IPM that can inform ongoing and future field campaigns and their process-modeling needs in the UCRB.

Standalone WRF simulations have been widely investigated in complex terrain, and provide context for the unfilled gaps in IPM investigation and development in complex terrain. For example, several papers detailed the role of subgrid-scale physics configuration on precipitation and snowpack processes in the UCRB (Rasmussen et al., 2011; Liu et al., 2011; Liu et al., 2017; Rasmussen et al, 2020). Outside of the UCRB, Orr et al. (2017) found cloud microphysics schemes have significant impacts on monsoon precipitation simulation in the complex-terrain Himalayan regions, with the Morrison microphysics scheme producing the best agreement with observations. Conversely, Comin et al. (2018) found that the Morrison microphysics scheme produced excessive snowfall and exhibited poor performance when evaluated in the Andes, while the Goddard (WDM6) scheme exhibited the best performance with respect to observed snowfall. In terms of land surface process, Jin et al. (2010)

explored that land surface model complexity improves temperature simulation, but has a minimal impact on simulated precipitation. Additionally, Mallard et al. (2017) evaluated that the sensitivity of near-surface temperatures and precipitation to changes in land use representation is smaller than the model error for those fields, while Rudisill et al., (2021) found that the details of snow cover in the initial conditions of a WRF simulation in complex terrain are key to ensuring the skill of that simulation, not just in 2-meter air temperature but also in the surface energy budget. Meanwhile, Rahimi et al (2022) found minimal sensitivity of SWE in WRF simulations across the entire western United States to microphysics schemes, but found large effects due to model resolution. On the other hand, the effects of meteorological forcing as the lateral boundary conditions of WRF simulations have also been recognized. For instance, Xu et al. (2018) identified that the simulations of hydroclimate in California using WRF are largely driven by large-scale forcing datasets. Taken together, the published literature suggests a one-size-fits-all WRF model configuration for hydrological studies in complex terrain may not be possible. In other words, the WRF configuration is likely case- and region-specific, and could depend either on the representation of processes within the WRF simulation domain or the boundary conditions of WRF forced by the large-scale meteorological forcing. The options of subgrid-scale physics schemes and large-scale meteorological forcing datasets need to be fully tested to understand their sensitivities to atmospheric and hydrological processes in the ERW.

Furthermore, few studies have assessed how these choices impact the subsequent simulation of surface-through-subsurface hydrologic processes. These types of analysis are needed because the WRF model can be configured in myriad ways for a given domain, and feedbacks to the surface and subsurface hydrology can yield a potentially large range of results. The aforementioned IPM study by Maina et al. (2020) showed that biases of 5-10% in basin-average surface water storage can result from forcing resolution differences in WRF alone, with localized differences in groundwater head by several meters. Schreiner-McGraw and Ajami (2020) show that water partitioning across four commonly used meteorological forcing datasets differs substantially within a Sierra Nevada watershed, and that the combination of precipitation uncertainty, soil parameterization, and topographic position all impact the severity to which these differences in forcing exert on the hydrology. However, neither standalone WRF nor WRF-Hydro explicitly simulate streamflow and three-dimensional groundwater processes. Groundwater in WRF-Hydro is highly simplified (shallow soil layers and a bucket model) while ParFlow simulates the full continuum of variably saturated flow in three dimensions. Therefore, one-dimensional land surface model alone cannot be used to better understand the configuration impacts on the greater hydrologic cycle, given the importance of lateral groundwater flow contributions to streamflow, especially in complex mountainous terrain.

In spite of the range of WRF sensitivity investigations, the connections between uncertainty in a WRF configuration and its influence on surface-through-subsurface hydrology is underexplored and therefore the focus of this work. It should be noted that our investigation is not to explore general principles behind IPM uncertainty quantification and error propagation, but rather to present a concrete use-case to guide the advancement of atmosphere-through-bedrock modeling and its connections to mountainous hydrological science. Using an IPM, we address an outstanding question: does synoptic-scale meteorological

forcing or meso-to-micro scale atmospheric processes have a more direct effect on surface and subsurface hydrologic processes in a mountainous watershed?

In order to answer this question, we undertake a series of experiments with different synoptic-scale meteorological forcing datasets, and different, plausible choices for meso-to-micro scale parameterizations in the IPM. This is informed by prior

standalone WRF studies that have utilized different shortwave and longwave radiation, microphysical, and surface and planetary boundary layer schemes (Skamarock et al., 2019). Additionally, topographical shortwave shading effects are tested to understand how spatial heterogeneity in the surface radiation budget influences evapotranspiration and snowpack accumulation and ablation processes (Arthur et al., 2018). Then we explore how the surface and subsurface hydrology fields respond to these various experimental setup choices, especially discharge in the ERW of the UCRB (described below).


With a discrete set of simulations, we establish the relative importance of these choices. We also establish the relative importance of subgrid-scale parameterizations that affect water and energy budgets. Our hypothesis is that synoptic-scale forcings produce a much larger spread in surface-through-subsurface hydrology fields than subgrid-scale physics scheme choice. If our hypothesis is confirmed, then scientific efforts to advance the predictive hydrology, through modeling, of the

UCRB should prioritize improving large-scale weather products and analyses. Conversely, if the hypothesis is falsified, model subgrid-scale physics scheme choice produces more variability in hydrologic response, then scientific efforts should prioritize the development of smaller scale atmospheric and hydrological process representations affected by surface heterogeneity in the ERW.

In this study, we also used the distributed hydrological model ParFlow-CLM to quantify streamflow and groundwater storage, since the hydrological processes included in WRF are over-simplified. Therefore, this article is organized as follows: first, we present details of study site and hydroclimate in the water year, as well as the IPM including the coupling between WRF and ParFlow-CLM and the justifications for using WRF and ParFlow-CLM as the atmospheric and surface-through-subsurface process models in the IPM, respectively. Then, we describe the WRF experiments that we performed to test the relative

importance of synoptic-scale boundary forcing and meso-to-micro scale model subgrid-scale physics schemes for driving ERW integrated hydrological simulations. Next, we present the simulated discharge, evapotranspiration and groundwater storage using ParFlow-CLM, to quantify the responses to changing WRF configurations. We conclude by contextualizing these results in light of the ongoing field campaign activities in the ERW.

## 2. Study Site

This investigation focused principally on modeling and analysis of the ERW, a mountainous headwater catchment of the UCRB near Crested Butte, Colorado (Hubbard et al., 2018). This 300 km$^2$ watershed of the Upper Colorado River Basin is at

a high-level, similar to other basins in the UCRB that it has very large gradients in precipitation (e.g., a factor of 2 range in precipitation between the northern and southern boundary of the ERW) and surface-through-subsurface hydrology. The ERW has a continental, subarctic climate with long, cold winters and short, cool summers. At an average elevation of 3266 meters above sea level, the watershed has a mean annual temperature of 0°C, and distinct winter and growing seasons that influence hydrologic and biogeochemical cycles. River discharges are driven primarily by snowmelt in late spring to early summer, with mid- to late-summer monsoonal rainfall inducing rapid but punctuated increases in streamflow. The ERW receives ~1200 mm/yr of precipitation and we focus here on Water Year 2019 (Oct 1, 2018 - Sep 30, 2019).

The ERW has become a mountainous community testbed for improving predictive understanding of multi-scale atmosphere-through-bedrock system dynamics and is the center piece of such focused activities because it is one of two major tributaries that form the Gunnison River, which in turn accounts for nearly half of the Colorado River's discharge at the Colorado–Utah border. In the past decade, several synthesis research efforts have been established in this region, including a wide range of fieldwork and modeling activities (Hubbard et al., 2018). The ERW has become one of the most heavily-instrumented mountainous watersheds in the world, which makes it an ideal location      for this research given the potentially large number of observational constraints available for the IPM efforts presented here. For example, The SAIL-based observations (Feldman et al., 2022) will be used in a future study to compare with IPM skill once the SAIL campaign is completed (2021-2023). Although a wide range of  precipitation, temperature and hydrological data have been collected, it is still challenging to use these to characterize atmospheric, surface and subsurface processes and their interactions at relevant scales.          .

## 3.   Methods

### 3.1. WRF models

The Weather Research & Forecasting (WRF) model version 4.0 is used in this study (Powers et al., 2017). WRF was chosen because of its widespread use in the investigation of atmospheric and land processes, and contextualizing observations in complex terrain (Rasmussen et al., 2011; Rasmussen et al., 2014). The WRF model is a fully coupled atmospheric and land surface model with a range of user-specific options for subgrid-scale physics schemes. WRF is a regional climate model that requires boundary and initial conditions provided by either global climate model (GCM) outputs or atmospheric reanalyses datasets. Our configuration of the WRF model is designed with three nested domains, with an outer, middle and inner domains at grid resolution of 4.5 km, 1.5 km and 0.5 km, respectively, centered around Crested Butte, Colorado where the East River watershed is located (Figure 1). All WRF simulations are initialized on Sep 15, 2018 but we discard the first 15 days of each simulation as spin-up.

While the stand-alone WRF model has been used extensively to advance the understanding of atmospheric processes, it has lower fidelity and applicability to investigate surface-through-subsurface hydrologic processes, and consequently is limited as an assessment and modeling tool for understanding integrated mountainous hydrologic cycle. Therefore, in order to provide an estimate of the entire hydrologic budget, we use a one-way coupling between WRF and an integrated hydrologic model, ParFlow-CLM (Maxwell et al., 2015, described in further detail below), that simulates the hydrological response of key variables not otherwise quantifiable in standalone WRF, such as discharge and groundwater storage.

Figure 2 summarizes this approach graphically. It shows that the one-way coupling enables an exploration of sensitivities of modeled hydrologic quantities (many of which can be observed) to combinations of atmospheric, surface, and sub-surface process representations. We do not choose a single configuration of WRF or ParFlow-CLM for this one-way coupling, but rather explore the uncertainty in representing atmospheric processes for integrated mountainous hydrology by analysing simulations with multiple, plausible configurations with multiple, plausible meteorological forcings. We recognize that the output from WRF simulations may be dependent on initial conditions, which are inherently difficult to constrain (e.g., Walser and Schär, 2004), but the experimental configuration described here seeks to be insulated from that dependency by running WRF simulations with initial conditions derived from different meteorological forcings.

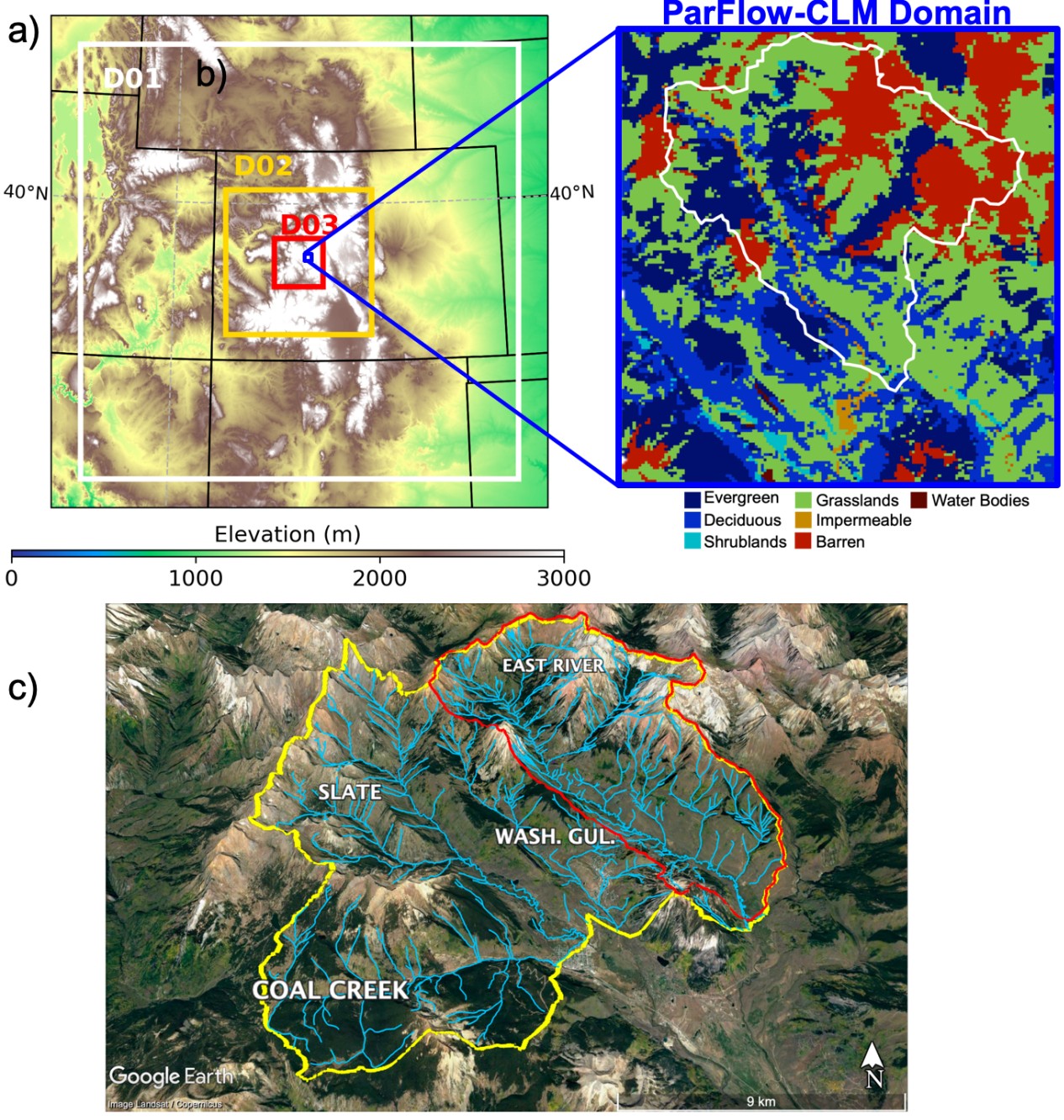

*Figure 1. a): Three nested WRF domains D01 (4.5 km grid resolution, 201 by 201 grid cells or 900 by 900 km extent), D01 (1.5 km grid resolution, 201 by 201 grid cells or 300 by 300 km extent), and D03 (0.5 km grid resolution, 201 by 201 grid cells or 100 by 100 km extent) and their associated elevations (left). The Global Multi-resolution Terrain Elevation Data 2010 (GMTED2010) elevation data in meters above mean sea-level is used in the WRF simulation. b): the innermost ParFlow-CLM*

*domain and spatial extent of the East River Watershed (white line) and associated land cover type derived from the National Land Cover Dataset (NLCD) (Homer et al, 2020) and upscaled to 100 m (right). c) Topography and stream network in the ERW and other nearby watersheds.*

A major experimental design decision when simulating the integrated mountainous hydrologic cycle is the computational cost associated with the simulations (e.g., simulated years per actual day) that are determined by model horizontal, vertical, and timestep resolutions as well as subgrid scale physics parameterization complexity. The computational expense incurred here to explore the sensitivities of WRF configuration choices were significant: one simulated year requires approximately 100,000 CPU hours on LBNL's Lawrencium lr6 supercomputing system. As such, it was highly impractical to simulate the entire configuration space of meteorological forcing and subgrid-scale parameterization choice. A discrete sub-sample of configurations, as presented here, is used to isolate and systematically determine which combination of subgrid scale parameterization choice is superior for a given domain such as the ERW. We therefore adopted a parsimonious approach to explore the space of possible WRF configurations, described below.

### 3.1.1. Subgrid-scale physics schemes

Three well-established suites of subgrid scale physics schemes for WRF are evaluated in this study (Table 1). One scheme was developed by NCAR and is used for a wide range of simulations over domains extending across the entire Conterminous United States (CONUS) (Liu et al., 2017). Another scheme that we consider here has been used for decadal-length hydroclimate simulation over California (Huang et al., 2017; Xu et al., 2018; Ullrich et al., 2018), and since it was initially developed by researchers at the University of California, Davis, it is denoted as UCD here. More recently, Flores et al. (2016) and Rudisill et al. (2021) implemented a WRF configuration that focused on exploring land-atmosphere interactions in complex terrain. This configuration was developed by researchers at Boise State University, and is referred to as BSU here. We recognized that this study would be computationally constrained given our prioritization of the use of sub-km horizontal resolution IPM simulations, and this is why we did not exhaustively sample the model configuration matrix.

*Table 1: Microphysics, radiation, land surface model, surface layer, and planetary boundary layer schemes used for the three different WRF configurations of the IPM tested here.*

| Subgrid-scale physics schemes | NCAR (CONUS) | BSU | UCD |
|---|---|---|---|
| Microphysics | Thompson | Thompson | WSM6 |
| Shortwave radiation | RRTMG | CAM | RRTMG |
| Longwave radiation | RRTMG | CAM | RRTMG |

| | | | |
|---|---|---|---|
| Land surface model | Noah | Noah-MP | Noah |
| Surface layer | Eta similarity | Monin-Obukhov | Revised MM5 |
| Planetary Boundary layer | Mellor-Yamada-Janjic scheme | Mellor-Yamada-Janjic scheme | UW (Bretherton and Park) |

### 3.1.2. Meteorological forcing

40 Each of these WRF configurations must specify a set of initial and lateral boundary conditions at the synoptic scale and, at least in the outer domain, are typically derived from high-resolution atmospheric reanalyses. The reanalysis from the National Centers for Environmental Prediction (NCEP), Climate Forecast System Reanalysis version 2 (CFSR2), The Modern-Era Retrospective analysis for Research and Applications - Version 2 (MERRA2), European Centre for Medium-Range Weather Forecasting Reanalysis version 5 (ERA5) were used in this study.

45

ERA5 is the fifth generation ECMWF atmospheric reanalysis of the global climate on a 30 km grid resolution (Hersbach et al, 2020), and combines model data with observations from across the world into a globally complete and consistent dataset. The CFSR2 is also global and is designed to provide an operational product for forecasting and analysis purposes at 0.3 degree grid resolution (Saha et al, 2010). The CFSR2 data were generated by an advanced assimilation schemes, atmosphere-land-ocean-

50 sea ice coupling, assimilates satellite radiances. MERRA2 is another atmospheric reanalysis based on data assimilation (Gelaro et al., 2017), which is the first long-term global reanalysis to assimilate space-based observations of aerosols and represent their interactions with other physical processes in the climate system. In addition, the NCEP FNL (NCEP, 2000) operational global analysis and forecast data are on a 0.25-degree grid resolution from the Global Data Assimilation System (GDAS) (Kleist et al, 2009). All meteorological forcing datasets are processed at 6-hourly by the WRF Preprocessing System (WPS).

55 ### 3.1.3 Topographic radiation

Topographic effects for shortwave radiation flux calculations in complex terrain are evaluated (Arthur et al., 2018). One is the "slope_rad" namelist option, which modifies surface solar radiation flux according to terrain slope by correcting it based on the solar zenith angle relative to the local surface normal vector. This adjustment ensures that the solar radiation received at the surface in WRF is consistent with the geometric projection of incoming sunlight onto local, non-flat surfaces. The other

60 namelist option, "topo_shading", allows for shadowing of neighboring grid cells. When "topo_shading" is active, WRF determines if any topography intersects a line drawn between a given grid point and the location of the sun at the time-step of the WRF run. If so, a topographic shadow is cast on that grid point and the direct component of the incoming solar radiation is set to 0. In this study, simulations in which "slope_rad" and "topo_shading" are jointly enabled are termed "3DRad" and when jointly disabled are termed "no3DRad", in the inner domain of the WRF simulation.

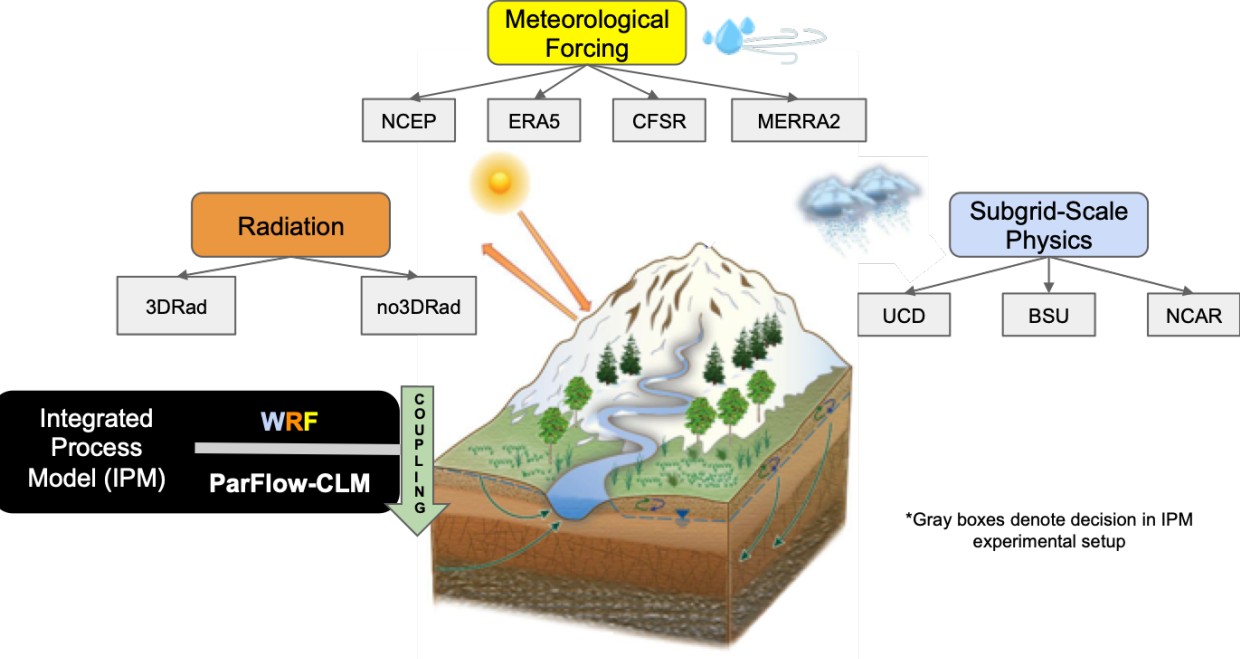

***Figure 2.*** *Conceptual framework for developing a set of different WRF configurations of the IPM to evaluate the sensitivities of subgrid-scale physics parameterization choice, meteorological forcing, and radiation scheme in the representation of mountain water and energy budgets.*

*Table 2:  East River Watershed WRF experiment configurations. Three subgrid-scale physics schemes, four meteorological forcings, and the topographic radiation options were assessed.*

| Subgrid-scale  physics schemes | Meteorological forcing | Topographic radiation |
|---|---|---|
| BSU | CFSR2 | 3DRad_inner |
|  |  | no3DRad_inner |
|  | ERA5 | 3DRad_inner |
|  |  | no3DRad_inner |
|  | MERRA2 | 3DRad_inner |
|  | NCEP | 3DRad_inner |
| UCD | CFSR2 | 3DRad_inner |

| | ERA5 | 3DRad_inner |
|---|---|---|
| NCAR | CFSR2 | 3DRad_inner |
| | ERA5 | 3DRad_inner |

## 3.2 ParFlow-CLM description

ParFlow is a physically based surface-subsurface hydrologic model that solves the coupled flow of saturated and variability-
saturated groundwater and overland surface water (Ashby and Falgout, 1996; Jones and Woodward, 2001; Maxwell, 2013).
The three-dimensional form of Richards equation is used to solve for lateral and vertical groundwater flow in the subsurface
and the kinematic wave approximation is used to solve two-dimensional overland flow. ParFlow is coupled to the land surface
model, the Common Land Model (CLM), which calculates a coupled water energy balance at every surface cell of the domain
(Dai et al., 2003) and incorporates spatially distributed vegetative processes by including specified land use types
parameterized by the International Geosphere-Biosphere Program standard database. Hourly meteorological forcing derived
from WRF drives ParFlow-CLM, and includes the following eight variables: precipitation, two-meter surface air temperature,
longwave radiation, shortwave radiation, 10-meter east-west and south-north wind speeds, atmospheric pressure, and specific
humidity. We also forced ParFlow-CLM with PRISM precipitation and temperature fields by evenly distributing daily
precipitation and temperature across a diurnal cycle of 24 hours within a day


The ParFlow-CLM subsurface domain is 30-meter deep at 100-meter horizontal resolution. The WRF outputs are re-grided
using bilinear interpolation to match the ParFlow-CLM grid cells. The model parameters are based on a variety of geological
and soil parameters, and calibrated using streamflow measurements. More details can be found in Foster et al. (2019) and
Pribulick et al. (2016). The computational expense of ParFlow-CLM is also less substantial than that of WRF for this model
configuration, but still requires high performance computing. Excluding the time for a multi-year initial condition spinup, a
single water year of the ParFlow-CLM simulations on 64 cores on the NERSC's Cori supercomputing system is approximately
1,000 CPU hours.

## 3.3 Reference Datasets

The Parameter-elevation Relationships on Independent Slopes Model (PRISM) dataset (Daly et al., 2008) was used here as a
point of comparison in evaluating model uncertainty across subgrid-scale physical schemes and meteorological forcing datasets
for precipitation and temperature. PRISM uses observations from quality-controlled meteorological stations along with a
topographic correction method against elevation based on empirical regressions to create daily gridded 800-meter total
precipitation, and daily average, minimum and maximum two-meter surface temperature. Although PRISM was generated
using statistical    models, it has been widely used for climate and hydrological model assessments (e.g., Lund    quist et al.,

2020) and associated uncertainty analyses (e.g., Buban et al., 2020). In the assessment of subgrid-scale physics schemes and meteorological conditions, the percent difference in cumulative precipitation is compared against PRISM by calculating by (max - min)/min*100, where max and min are the maximum and minimum cumulative precipitation values from the simulations within each group, respectively.

Snowpack Telemetry (SNOTEL) data have been widely used in snowpack assessment (Serreze et al, 1999; Fassnacht et al, 2003), and we use three SNOTEL stations (Butte, Schofield Pass, Upper Taylor) within the WRF inner domain to assess the snowpack simulation skill of each IPM configuration. Significant heterogeneity is sampled by the three SNOTEL stations (within or near the ERW) due to the complex topography. For example, the Butte station is located downstream of the ERW and, on average, receives approximately 0.8 m of precipitation, and reaches 0.4 m in maximum snow water equivalent over

the year. On the other hand, the Schofield Pass station is located upstream of the ERW and, on average, receives 1.2 m of precipitation and reaches 0.9 m in maximum snow water equivalent. In addition, we use the snow water equivalent product of the Airborne Snow Observatory (ASO; Painter et al, 2016) on 04/07/2019 to evaluate the spatial pattern skill of the snowpack simulation across WRF configurations (Figure S-8). The raw ASO product has 50-meter spatial resolution, and is regridded to the same grid resolution as WRF outputs (500 meters) for comparison purposes using bilinear interpolation, as documented in

Oaida et al. (2019). Since the spatial resolution of ASO data are significantly finer than the WRF outputs, we acknowledge that the underestimation by ASO could be due to the point-to-grid errors (Oaida et al. 2019). Notably, ASO SWE estimates are lower than SNOTEL SWE measurements (ASO: 389 mm at Butte, 938 mm at Schofield Pass; SNOTEL: 490 mm at Butte, 1260 mm at Schofield Pass). In addition to SNOTEL station data, stream gauges measurement of discharge at the pumphouse, the outlet of the ERW, is used to evaluate the ParFlow-CLM simulation results.

**4.  Results**

**4.1 Sub-grid physical schemes vs meteorological forcings**

We start by presenting a number of time-series of spatial averages over the ERW for WY19. They indicate the gross performance of the IPM across the water year, and whether a configuration produces generally reasonable estimates relative to observational products. Figure 3 shows cumulative precipitation, two-meter surface air temperature, and snow water

equivalent (SWE) aggregated over the ERW, and the *in-situ* assessments compared against two SNOTEL stations are in Figure S-3 For cumulative precipitation, each configuration produces amounts higher than PRISM (cumulative precipitation of 1201 mm), and the UCD simulates the highest cumulative precipitation. For surface air temperature, the seasonal cycle and daily variability are captured by all configurations, however, exhibit systematic cold biases relative to PRISM (annual average two-meter surface air temperature of 0.6 degrees Celsius). In terms of SWE, all model configurations concur in their representation

of the snowpack accumulation season and melt season in late spring and into summer, except UCD which simulates an earlier peak timing of SWE.

The spread in cumulative precipitation when comparing across different meteorological forcing dataset is apparent (Figure 3). Although UCD and NCAR configurations show a greater difference in precipitation forced by ERA5 and CFSR2, the

consistency across BSU configurations is notable, which also shows the closest agreement with PRISM. When comparing the relative roles of subgrid-scale physics scheme choices to meteorological forcings, the percent difference of cumulative precipitation, calculated with (maximum - minimum)/minimum*100, across BSU-CSFR2, UCD-CSFR2 and NCAR-CSFR2 schemes is nearly 34% of the minimum cumulative precipitation simulated by BSU-CFSR2, compared to the 4.6% of the simulations across BSU configurations with different meteorological forcing (CSFR2, ERA5, MERRA2 and NCEP).


BSU simulations are generally in agreement with PRISM. However, the UCD simulations are outliers relative to the other simulations, with cumulative precipitation of 1706 mm, or 42% higher at the end of the water year, with the most notable differences occurring in March through September. NCAR simulations show general agreement with PRISM and BSU throughout the water year, save for June through September. The two-meter surface air temperature time-series reveals that

the UCD simulation is systematically colder throughout the winter and spring, regardless of which meteorological forcing dataset is used. The persistent cold bias simulated by the UCD, NCAR and BSU schemes has been found in previous WRF studies within western US mountain regions (Xu et al., 2018, Rudisill et al. 2021). The SWE time-series again shows a similar relationship with precipitation, with the outlier being UCD-ERA5, in terms of the seasonal timing of when snowpack peaks and melts (Figure S-1).  Comparing the monthly average between UCD-ERA5 (Figure S-4) and BSU-ERA5 (Figure S-5), the

early snowmelt observed in the UCD scheme is likely a result of warmer temperatures at low-altitude region that melt the snow earlier in the water year. However, the high-altitude regions remain cold enough to maintain snowpack through early-mid summer.

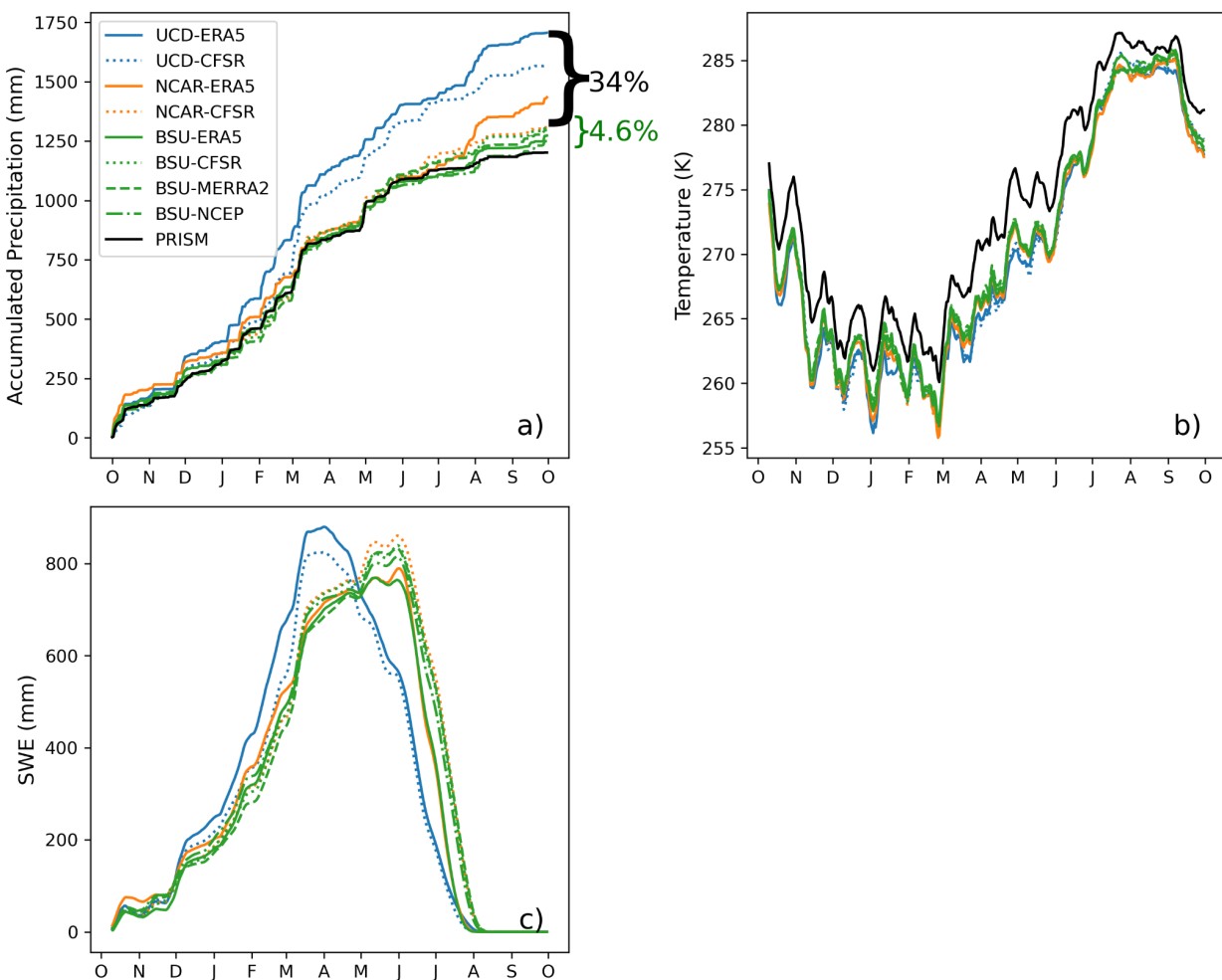

**Figure 3.** *a) Cumulative precipitation, b) two-meter surface air temperature and c) snow water equivalent (SWE) simulated within the ERW using an IPM with different subgrid-scale physics schemes and meteorological forcings. The cumulative precipitation and temperature results are compared relative to PRISM. 10-day moving averages of daily temperature are shown in b). The percent difference in cumulative precipitation across subgrid-scale physics schemes (black brackets) and meteorological forcing (green brackets), calculated by (maximum - minimum)/minimum\*100, are provided on the right y-axis.*

In addition to the domain-averages, spatial heterogeneity due to land-surface cover and topographic effects are shown in Figure 4. The systematic cold bias simulated throughout the water year appears to be an elevation-dependent phenomena with higher-elevations exhibiting an enhanced cold bias compared with PRISM. However, the river valley and relatively lower-elevation areas at the southern edge of the ERW, which includes Crested Butte Mountain, stands out as these regions are warmer than the PRISM dataset. Figure 4b shows precipitation in BSU-CFSR2 is wetter in the western regions, and drier in the eastern, of the ERW in comparison against PRISM. Figure S-3 shows comparisons between PRISM and the IPM configurations and

indicates no biases that are persistent across seasons. During summer, the BSU-CFSR2 simulation consistently produces more precipitation than PRISM.


Although the two-meter surface air temperature bias is evident, it doesn't vary significantly across either subgrid-scale physics scheme or meteorological forcing. Therefore, subsequent exploration in this study will be focused on precipitation. The bottom row in Figure 4 shows the grid-cell standard deviation of monthly precipitation across subgrid-scale physics schemes (i.e., UCD, NCAR and BSU simulations with CFSR2 meteorological forcing – bottom left) and BSU simulation driven by different

meteorological forcing datasets (ERA5, CFSR2, MERR2 and NCEP – bottom right). Similar to the conclusions drawn from Figure 3, Supplementary Figure S-6    also shows the monthly spatial standard deviations across subgrid-scale physics schemes are generally greater than meteorological forcing, particularly in regions of higher-elevation during the winter season.

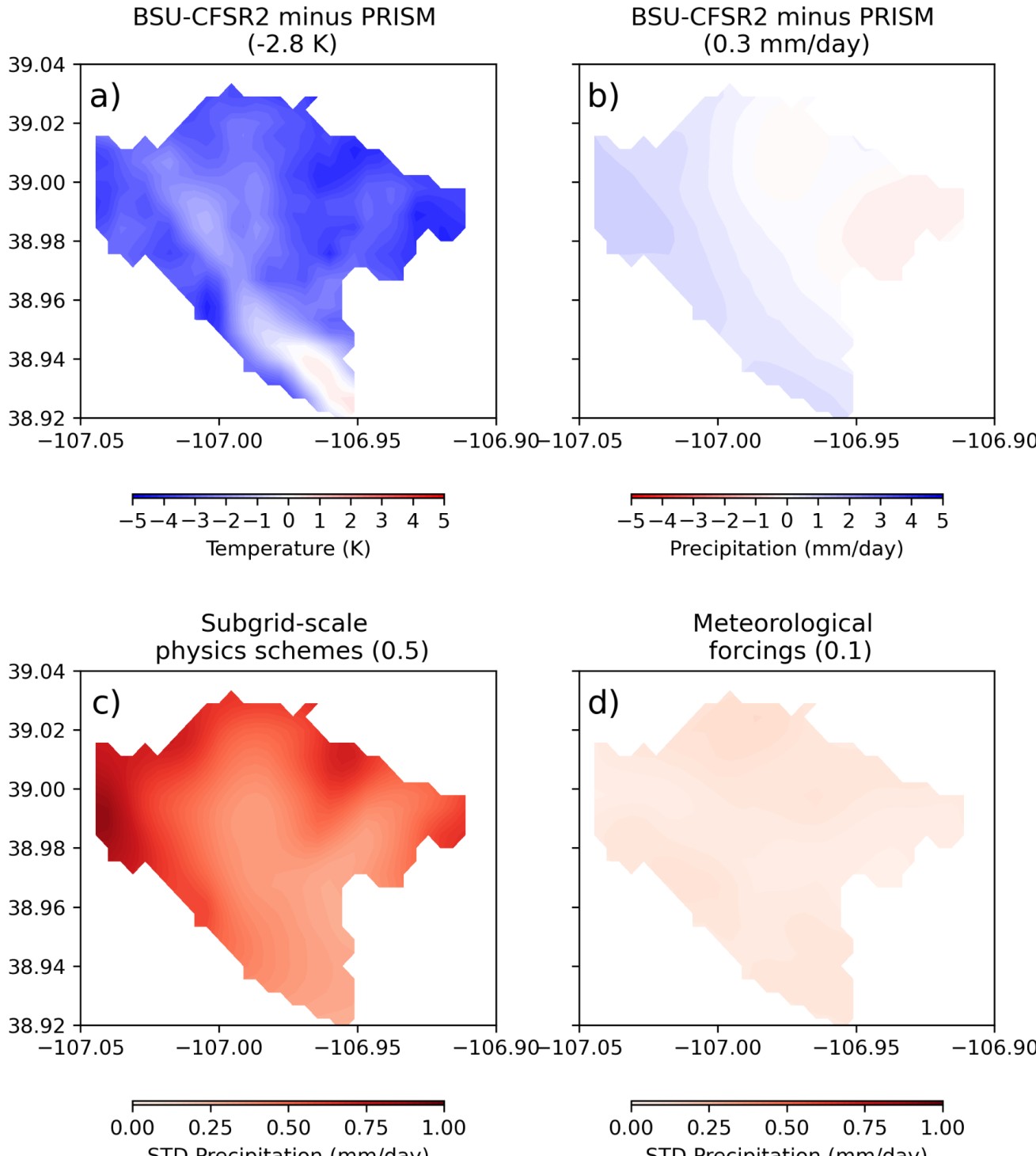


*Figure 4. Upper row: Differences in spatial distributions of annual average two-meter surface air temperature (a) and cumulative precipitation (b) between the BSU-CFSR2 WRF configuration and PRISM. Lower row: For all schemes, the standard deviation of annual cumulative precipitation is plotted for subgrid-scale physics schemes (c) and meteorological forcings (d). The values in the parentheses are the domain average differences over the water year. The standard deviations are the total annual precipitation in each ensemble simulations using different subgrid-scale physics schemes or large-scale meterological forcings.*

Quantitative statistics of the aggregated domain-average precipitation and temperature simulations for the WRF simulation across subgrid-scale physical schemes and large-scale meteorological forcings are presented in Table 3. Although NCAR-CFSR has a higher R2 than other simulations, NCAR-ERA5 has a very low R2. The BSU simulations provide a closer approximation of cumulative precipitation to PRISM. Specifically, BSU does better in simulating extreme precipitation events (i.e., 95th percentile). Therefore, we conclude that BSU WRF subgrid-scale physics schemes outperform the UCD and NCAR WRF subgrid-scale physics schemes in simulating both precipitation and temperature. On the other hand, the differences in precipitation and two-meter surface air temperatures across the four meteorological forcings are not statistically significant, and their standard deviations are much smaller than the differences in simulations across subgrid-scale physical schemes. While there are many metrics of model skill when selecting a meteorological forcing to simulate the hydrological processes in the ERW, we choose BSU-CFSR for the topographic radiation study in the next subsection due to its better match with PRISM, using our skill measures, in simulating both precipitation and two-meter surface air temperature.

*Table 3. Quantitative measures of precipitation and temperature of the WRF simulations among sub-grid physical schemes and meteorological forcings. R2 are the coefficient of determination for simulations and PRISM daily time series.*

| | Total Precipitation (mm) | Temperature (K) | Precipitation_R2 | Temperature_R2 | 95th percentile of daily precipitation (mm) |
|---|---|---|---|---|---|
| UCD-ERA5 | 1,706 | -3.14 | 0.26 | 0.79 | 20.84 |
| UCD-CFSR | 1,568 | -2.82 | 0.42 | 0.82 | 21.09 |
| NCAR-ERA5 | 1,435 | -2.80 | 0.16 | 0.82 | 19.40 |
| NCAR-CFSR | 1,308 | -2.50 | 0.50 | 0.85 | 18.10 |
| BSU-ERA5 | 1,273 | -2.31 | 0.32 | 0.86 | 17.70 |
| BSU-CFSR | 1,267 | -2.23 | 0.42 | 0.87 | 18.45 |

| | | | | | |
|---|---|---|---|---|---|
| BSU-MERRA | 1,296 | -2.20 | 0.36 | 0.87 | 19.51 |
| BSU-NCEP | 1,249 | -2.41 | 0.42 | 0.86 | 16.68 |
| PRISM | 1,202 | 0.59 | | | 17.61 |

## 4.2. 3D topographic radiation effects

05  Based on the assessment of simulated precipitation and two-meter surface air temperature compared with PRISM, the BSU-CFSR2 configuration is selected as a baseline to further explore the influence of topographic radiation scheme effects. The difference caused by turning on and off the 3D topographic radiation effects is similar in other WRF configurations; therefore, only the BSU-CFSR is presented. Figure 5 shows daily ERW spatial average time series over the water year for the major mountainous water and energy budget variables. By isolating the impacts of subgrid-scale physics schemes and meteorological

10  forcings across IPM simulations, it is easier to systematically intercompare cause-and-effect across different topographic radiation options. Consistent with previous findings, all configurations still overestimate cumulative precipitation and are too cold relative to PRISM.

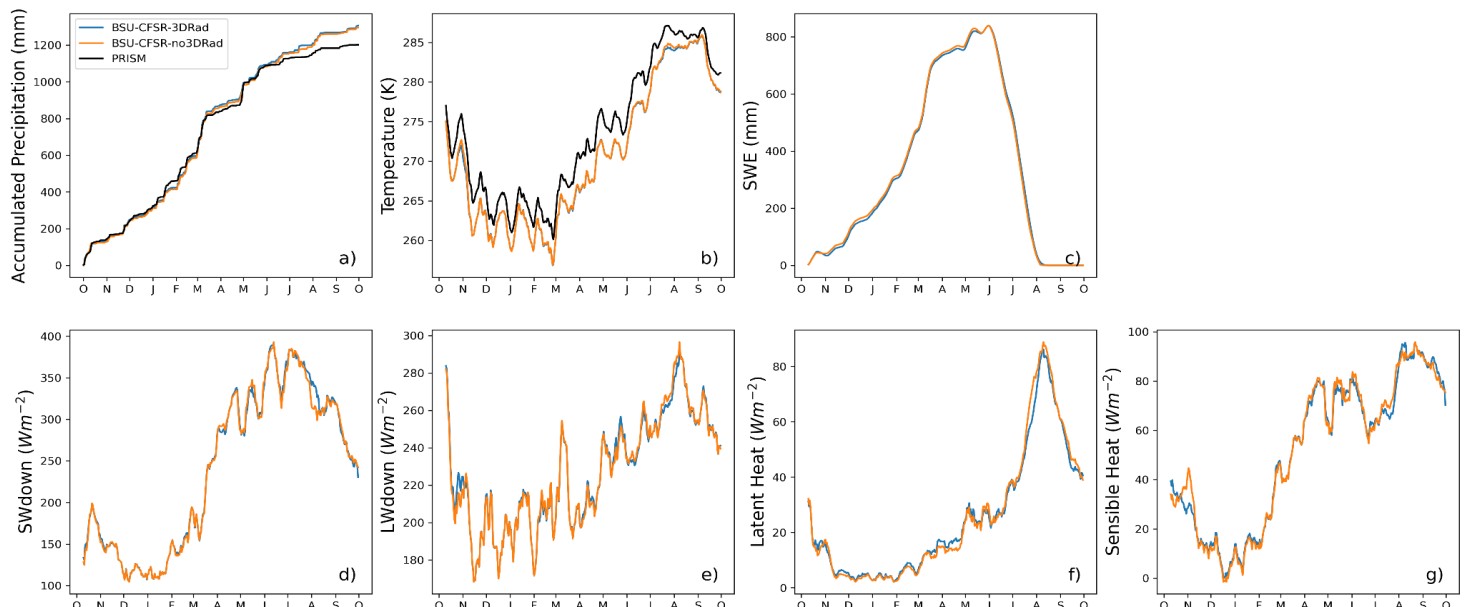

15  ***Figure 5.*** *Spatial-average cumulative precipitation, two-meter surface air temperature and snow water equivalent (SWE) (first row), shortwave and longwave radiation, and latent and sensible heat (second row) over the ERW as simulated by the IPM configurations with and without realistic topographic radiation effects, along with, where available, estimates from PRISM. 3DRad indicates a simulation with topo_shading and slope_rad turned on in the WRF inner domain but not the outer WRF*

*domains, no3DRad indicates a simulation with top_shading and slope_rad turned off in both the inner and outer WRF domains. 10-day moving averages are shown in b) temperature, and radiation variables (d, e, f, g).*

Figure 6 shows the seasonally-resolved shortwave radiation, two-meter surface air temperature, latent heat flux and SWE for different configurations of shortwave radiation in the simulation with and without topo_shaing and slope_rad options in the inner domain. While no3DRad does not adjust the SWdown (incoming shortwave radiation), 3DRad simulation recalculates the SWdown based on the shadows cast by nearby topography. Spatial differences in IPM-simulated shortwave radiation (Figure 6b) are seen in the northeast and western portions of the ERW, when topographic effect of shortwave radiation is included. As a result, a corresponding change in the spatial pattern of simulated two-meter surface air temperature and latent heat flux are seen, which are driven by the change in downwelling shortwave radiation with topographic shading (Figures 6a and 6d). Topographic shading makes a difference locally in LH flux, by redistributing the energy flux and thus affecting LH flux spatial distribution. Nevertheless, the domain averaged LH flux remains unchanged between cases. The resulting pattern change in SWE (Figure 6c) shows that the northern and northeastern sections of the ERW, where snowpack are concentrated, are sensitive to shortwave radiation. This is expected and consistent with previous findings that included topographic effects in shortwave radiation and found distinct spatial patterns of hydrologic variable sensitivity due to both shadows and surface reflection that produce time-varying effects on net surface radiation (Lee et al., 2015; Palazzi et al., 2019; Gu et al., 2020; Hao et al., 2021).

Although Figure 5 shows that realistic shortwave radiation produces small effects on the seasonal cycle of the surface energy and mass budgets when averaged over the entire watershed, including annual average SWE (Figure 5c), Figure 6c shows that mountains and valleys have different amounts of SWE. Furthermore, seasonal patterns show simulated latent heat is diminished at lower elevations from March to May, when snowmelt occurs in the valley, and the remaining snowpack in the mountains and late snowmelt in 3DRad simulation causes lower latent heat flux shown in July (Figure S-7). The 3D radiation shading scheme does not significantly affect the total water balance, but rather the spatial distribution of radiation fluxes. Thus, despite having minimal impacts on water impacting on the water balance, the scheme does have important localized impacts on SWE and surface energy budget spatial patterns. The 3DRad simulation has less SWE in the valleys during the accumulation season but more SWE at higher elevations during the melt season, which is a direct result of the differences in shortwave radiation redistribution. Figure S-5 also shows that the latent heat differences in north-facing and south-facing sides are most apparent in the snowmelt and warm seasons. This is consistent with previous findings (Lee et al., 2015; Palazzi et al., 2019; Gu et al., 2020; Hao et al., 2021), that a more realistic treatment of shortwave radiation, which includes shadows and projected insolation on sloped surfaces, results in lower shortwave insolation on the surface at this time of year. The lower shortwave radiation should, in turn, decrease the energy available for the IPM to produce snowmelt. In summary, the simulations show that, while local spatial differences in surface radiation with and without realistic topography are apparent in Figure 6, the domain spatial averages (even for SWE) are the same between shaded and non-shaded formulations. This suggests that while it may be striking

localized differences when shading is included, the impact of topographic shading on the entire water balance over a spatial domain like the ERW is negligible.

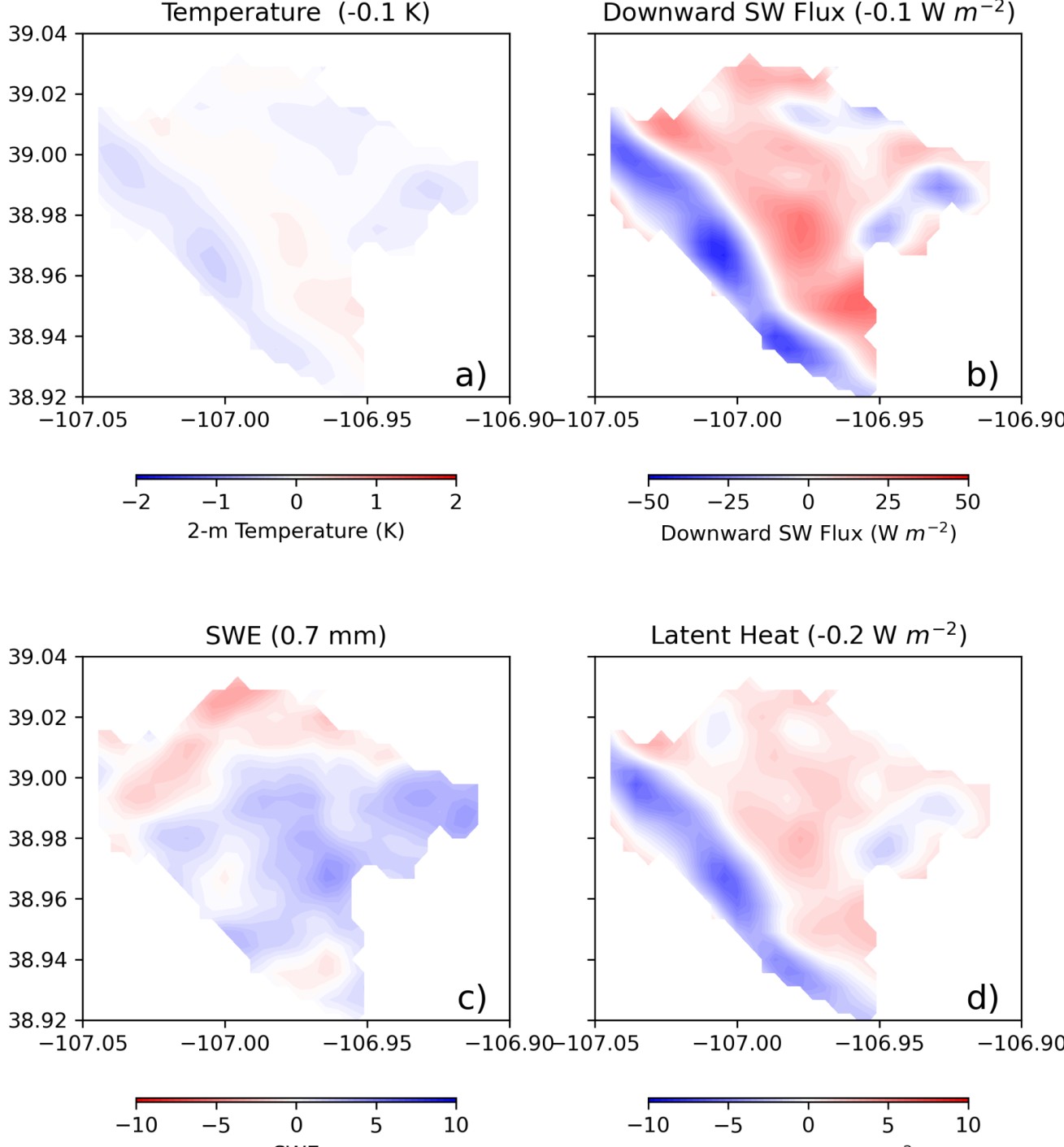

*Figure 6.* *Topographic radiation differences (3dRad minus no3dRad) annual average two-meter surface air temperature, shortwave (SW) and latent heat flux, and snow water equivalent (SWE) over the ERW. The values in the parentheses are the ERW average differences over the water year, which are small and consistent with Figure 5.*


### 4.3. Hydrological and streamflow responses

We have evaluated the aforementioned WRF configurations subgrid-scale physical scheme and large-scale meteorological forcings in representing precipitation, temperature, snowpack and radiation fluxes, and their impacts on the integrated water budget within the ParFlow-CLM. We also evaluated the simulated discharge from ParFlow-CLM forced by PRISM as a

comparison with WRF forcings. Figure 7 shows the simulated hydrologic output from the ParFlow-CLM model for watershed outlet discharge (top row) and watershed-average groundwater storage (bottom row). Discharge at the watershed outlet (see exact location on Figure 1) shows a different timing across the various WRF subgrid-scale physics scheme configurations and large-scale meteorological forcings that lead to a temporal shift in simulated streamflow, where the daily averaged time series (left) shows only minor differences through time. However, cumulative discharge by year-end reveals substantial differences

(right), especially after peak snowmelt where estimates of cumulative discharge begin to diverge. Differences across the WRF configurations are especially large; the difference across the three subgrid-scale physics scheme configurations with ERA5 (UCD, NCAR, and BSU) varies by 26% by year-end. Differences across meteorological forcing (using the BSU physics configuration as a control, shown in green) are also noteworthy, although smaller, approximately 6%. These results are consistent with variation of simulated precipitation in WRF described earlier, confirming that for this basin, meteorological

forcing drives less variance on hydrologic response than subgrid-scale physics scheme configuration.

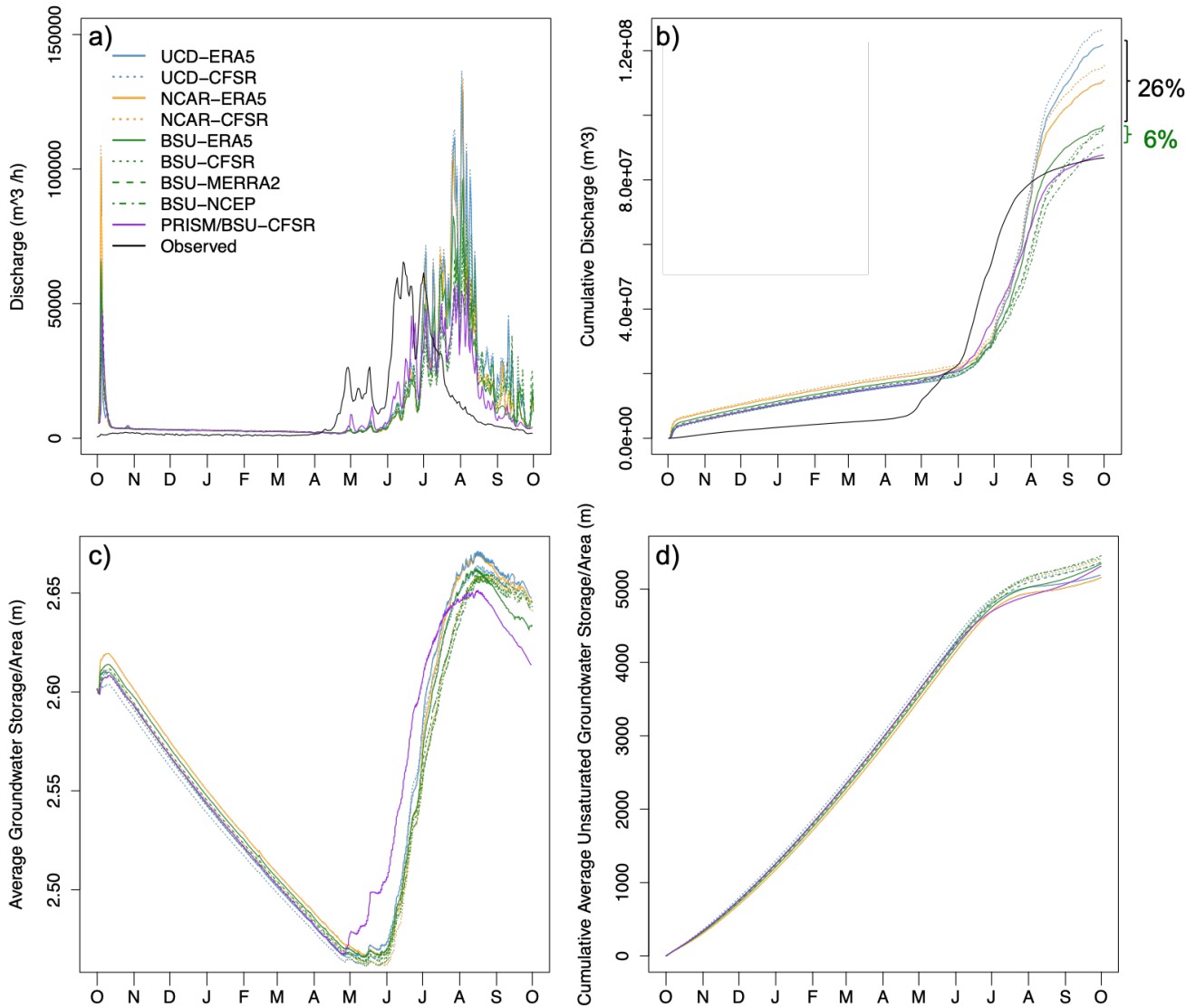

*Figure 7. Time-series of ParFlow-CLM simulations of discharge rate (a), cumulative discharge (b), groundwater storage per unit area of the watershed (c), and cumulative average unsaturated groundwater storage per area of the watershed (d) for the IPM configurations described in Table 2. The brackets on the far-right indicate the percent difference of cumulative discharge and unsaturated groundwater storage per area (b and d, respectively) for WRF simulations across different meteorological forcings (green) and subgrid-scale physics schemes (black).*


**In addition to the variance of cumulative discharge with WRF simulations across different conditions,** a comparison to observed discharge is also shown on Figure 7, which for all scenarios suggest a delayed snowmelt response in the IPM. While the objective of this study is not to replicate the observations, but rather determine sensitivity across IPM configuration choice, the mismatch in streamflow response suggests a systematic cold-bias from the WRF input into ParFlow-CLM which is consistent with the discussion surrounding Figure 3 in relationship to PRISM. An early-fall peak in simulated discharge is also seen in all WRF simulations, and not in observed discharge, although a significant increase in SNOTEL precipitation was measured in October of that year (see Figure S-1, S-2). This further supports a temperature bias, albeit opposite that of the cold-bias discussed previously, where precipitation around that October storm-event falling as rain (as opposed to snow) leads to a sharp increase in discharge. A sensitivity analysis of the BSU-ERA5 model run for a lower precipitation year (water year 2018, which was nearly half the precipitation of 2019), showed better agreement with observed discharge, which suggests the bias in timing may be a function of accumulated precipitation and/or snowmelt, and is reserved for future studies (not shown).

Basin-average groundwater storage, shown in Figure 7c in area-normalized units, shows a strong annual signal for all WRF configurations with minimal differences across IPM configurations. Here all groundwater, inclusive of saturated or unsaturated storage, is considered. The cumulative, area-normalized annual groundwater storage, when accounting for only vadose zone storage (Figure 7d), which is most responsive to sub-annual differences in precipitation inputs, is meaningful in this context because it relates a cumulative impact on near-surface groundwater storage due to IPM configuration. Similar to year-end cumulative discharge, year-end departures in vadose zone groundwater storage across the different simulations are evident. Differences across the IPM configurations of subgrid-scale physics schemes are larger than the difference across the forcing simulations (4% versus 2%, respectively). While the differences in groundwater signals are not as pronounced as the discharge signals, streamflow signals are very reactive, noisy, and change quickly, whereas groundwater signals are the product of slower processes via infiltration and vadose zone dynamics, often at longer timescales, which result in very different temporal signals as compared to streamflow.

Figure 8 shows maps of standard deviations in annual total evapotranspiration (ET) simulated by ParFlow-CLM across IPM configurations (top row), as well as the cell-binned relationship of those standard deviations of annual ET with land use and cover type, as well as elevation (bottom). Consistent with variations shown in the simulated discharge and groundwater storage, Parflow-CLM simulates greater variations of ET under WRF configurations driven by different subgrid-scale physics schemes (Figure 8a), compared to the simulations conducted with different meteorological forcings (Figure 8b). These results suggest that locations populated by high-water demanding vegetation (namely evergreen and deciduous forests) at mid-elevations result in the highest ET variability across IPM configurations. Conversely, low-water demanding vegetation (barren/sparsely vegetated land and grasses), which reside across a range of elevations in the study domain, result in the lowest variability in annual ET across IPM configurations. These differences in water demand essentially magnify any differences in atmospheric conditions.

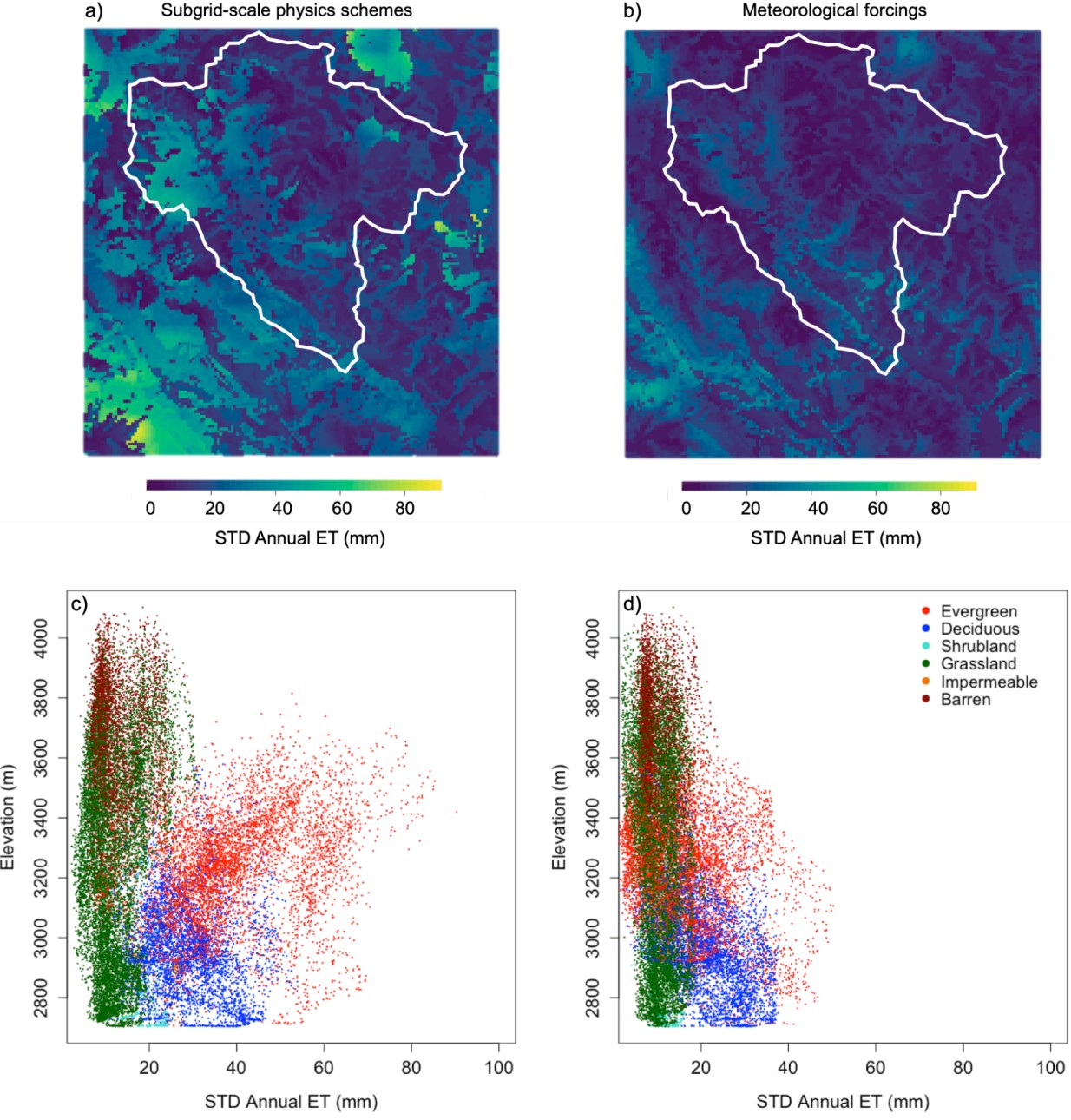

***Figure 8.*** *Pixel-level standard deviation in annual total evapotranspiration (ET) over the ParFlow-CLM domain from WRF with different subgrid-scale physics schemes (a,c) or meteorological forcing (b,d). The ERW outline is overlain in white in the*

*upper row, a-b. The standard deviation of simulated ET from ParFlow-CLM across different physics schemes (c) and meteorological forcing (d) are presented in the lower row, c-d. For each pixel in a-b, the relationship between annual ET standard deviation, elevation (y-axis), and land cover type (colors) are shown as scatter plots on the bottom row, c-d. See Figure 1 for maps of land cover types. Subgrid-scale physics schemes (a,c) have more variance compared to meteorological forcings (b,d), especially for mid-elevations and in evergreen forests.*

## 5. Discussion and Conclusions

### 5.1 Scientific Findings

In spite of previous efforts to characterize the sensitivity of WRF simulations to model configuration choices, the mountain climate and hydrology scientific community has not sufficiently explored the implications of those choices for surface and subsurface hydrology in high-altitude complex terrain. Here, we used an IPM with one-way feedbacks from WRF to ParFlow-CLM to assess the hydrometeorology of the ERW which is characterized by strong hydrological gradients indicative of mountain environments of the UCRB.

In this paper, we present a number of numerical experiment results that are informative for the scientific community to better understand atmosphere-through-bedrock process interactions, and the uncertainties of those    interactions between    climate and hydrological model experimental setup choices. First, the uncertainties associated with meteorological forcing choice are less important than subgrid-scale physics scheme choice, at least in the ERW. This finding has important implications for IPM in complex terrain, since it reveals that the differences in reanalysis products are less consequential for initializing and forcing IPMs than atmospheric configurations, and that efforts to advance IPMs such as collecting observations and using them to evaluate physical process parameterizations at the sub-HUC-8 scale could help to better constrain model performance. This result also shows that the large-scale meteorological forcing of the IPM simulation are less important in driving the magnitude and spatial variability of key hydrometeorological variables than the details in choosing and optimizing atmospheric subgrid-scale physics schemes (e.g., microphysics or boundary layer turbulence). Ultimately, we used the BSU-CFSR2 configuration to recreate    WY2019 in the ERW which allows researchers, in this case, to prioritize process studies and the development of associated observational constraints within the ERW. However, further investigation is needed to evaluate the systemic cold bias across IPM configurations, particularly at higher elevations, and the consequence of delayed snowmelt and timing of discharge peaks.

We recognize that numerous works in meteorological disciplines have demonstrated that "physical parameterization is much more important than lateral or initial conditions" (e.g., Solman and Pessacg, 2012; Pohl et al., 2011)). However, our findings are not redundant with the published literature, as those references either evaluated large-scale meteorological processes or did not focus on high-altitude complex terrain regions, which are central to our study. Additionally, most IPM studies to date do

not show how the range of reasonable IPM configurations (based on configurations that have been presented in the published literature) affects water management relevant processes such as discharge, ET and subsurface hydrology. With our set of one-way atmosphere-through-bedrock process modeling results, we now show how choices in atmospheric process model configurations impact the surface and subsurface hydrology. Specifically, we evaluate and quantify the sensitivity of discharge,

ET, and subsurface hydrology to IPM configurations, and we also address how 3D topographic radiation schemes affect both the spatial distribution and spatial average aspects of the mountainous hydrologic budget.

In the investigation of topographical and slope gradient effects on shortwave radiation, our study shows those considerations in WRF are essential in redistributing radiation flux over regions of complex terrain, even though the differences in spatial-

average performance over ERW is minimal. This is because the spatial redistribution of shortwave radiation leads to approximately +/- 30 W/m$^2$ difference in the east/west facing slopes that lead to +/- 1 K difference in two-meter surface air temperature in August and September (when snowpack is nonexistent). Throughout most of the water year when snowpack exists, the spatial heterogeneity of temperature differences is less apparent than for shortwave radiation. Latent heat is buffers differences in the shortwave radiation contribution to the radiation budget, and causes early snowmelt in the high

elevation mountains in those simulations with topographical and slope gradient shortwave radiation effects turned on. At the same time, the systemic cold bias and limitations of one-way feedback in this study is potentially indicative of challenges in extrapolating findings from one mountainous watershed to another. If atmospheric process details are significant for surface and subsurface hydrological modeling and if the findings regarding atmospheric processes in one study area are marginally or completely irrelevant to other mountainous watersheds, then additional field work would be needed in mountainous hydrology

research to address this issue, given that the extrapolation of fieldwork results remains a central challenge for field-based research and modeling activities.

### 5.2 Limitations and Future Works

A limitation of our study, given the computational constraints of running IPMs, is that it was infeasible to explore the full

parameter spaces of WRF and ParFlow-CLM exhaustively; thus, our conclusions are limited to the selected subgrid-scale physics schemes and meteorological forcing datasets analyzed. Additional work is needed to improve the systemic cold bias in two-meter surface air temperature throughout all experiments as this may have been the major driver in the delayed snowmelt and peak discharge simulated by the IPM.

Another methodological constraint is that our WRF and Parflow-CLM experiments were only one-way instead of two-way feedbacks, which ignores potentially important feedbacks from the subsurface hydrology to the atmosphere via ET and the radiation budget. For example, Givati et al. (2016) reported that simulated precipitation was improved with two-way coupling in WRF-Hydro compared to WRF-only and Forrester et al. (2018) showed that boundary layer dynamics were impacted in IPM simulations in regions where shallow water tables exist. On the other hand, ParFlow-CLM is essential in our experiment

for quantifying hydrological responses, including streamflow and groundwater storage. Although other fully-coupled integrated hydrology model (e.g., WRF-Hydro) provides some insights into streamflow, it still uses a simplified and prescribed stream network. Groundwater storage in WRF-Hydro is also highly simplified by using a bucket model while ParFlow-CLM simulates the full 3D continuum of variable saturation in three dimensions. Importantly, in a similar fashion as the hierarchy of climate models approach oft used in the climate community (Jeevanjee et al., 2017), we would also like to assess one-way coupling performance of our IPM prior to assessing two-way coupling IPM performance.

The East River watershed is already highly instrumented due to the presence of the long-standing Rocky Mountain Biological Laboratory (RMBL), the SNOTEL network, the United States Geological Survey's Next Generation Water Observing System (NGWOS), the National Science Foundation's Sublimation of Snow (SOS) project, and DOE Watershed Science Focus Area project which has been adding instrumentation to the watershed over the last ~7 years. While these observations focus primarily on surface and subsurface processes, the East River watershed has become even more instrumented in recent years (2021-2023) through the support of the U.S. DOE (SAIL campaign) and U.S. NOAA (SPLASH campaign) deployments of a comprehensive set of atmospheric instrumentations (e.g., radar and radiation measurements). Future work will include integration of data, either indirectly through IPM benchmarking or directly through data assimilation into the IPM, from the SAIL campaign. SAIL is collecting a wide-array of observations with the intent to advance understanding of precipitation, snow, aerosol, aerosol-cloud interaction, and radiation processes in complex terrain and establish the minimum-but-sufficient level of process understanding to develop a robust predictive understanding of seasonal surface water and energy budgets in the ERW (Feldman et al., 2021). SAIL aims to develop a wide range of hydrometeorological datasets to constrain atmosphere, surface, and subsurface processes simultaneously. Together, these resources are contributing to the establishment of a highly-instrumented and studied UCRB watershed. We look forward to building upon the knowledge learned from this manuscript to compare the most appropriately configured IPM to SAIL and SPLASH campaign observations. Our study highlights that the benchmarking provided by these data collections will be critical in addressing the systemic IPM cold bias by providing a more constrained estimate of radiation budgets in complex terrain that ultimately shape snowmelt and discharge.

*Data Availability:* All WRF model output files can be found at
https://portal.nersc.gov/archive/home/z/zexuanxu/Shared/www/IPM
Please notify corresponding author Zexuan Xu (zexuanxu@lbl.gov) if you used our data.

*Author contributions:* ZX, ESW, AMR and DF designed the study together. ZX performed the WRF simulations and analyzed the results. ESW performed the ParFlow-CLM simulations. All authors contributed to the writing and approve of this manuscript.

*Competing interests:* The contact author has declared that neither they nor their co-authors have any competing interests.

*Acknowledgments:* This work was supported by the Laboratory Directed Research and Development Program of Lawrence Berkeley National Laboratory. This work was also supported by the Watershed Function Scientific Focus Area project, and the Atmospheric Radiation Measurement User Facility Program, the U.S. Department of Energy, Office of Science, Office of Biological and Environmental Research, under U.S. Department of Energy Contract No. DE-AC02-05CH11231. This research used resources of the National Energy Research Scientific Computing Center, a DOE Office of Science User Facility supported by the Office of Science of the U.S. Department of Energy under that same contract. This research also used the Lawrencium computational cluster resource provided by the IT Division at the Lawrence Berkeley National Laboratory (supported by the Director, Office of Science, Office of Basic Energy Sciences, of the U.S. Department of Energy under contract no. DE-AC02-05CH11231). The authors acknowledge the helpful guidance provided by Professor Lejo Flores at Boise State University and Dr. Will Rudisill at Lawrence Berkeley National Laboratory regarding the WRF configurations conducted in this study over the ERW.

*Financial support:* Authors Xu, Siirilla-Woodburn and Feldman were supported by the Laboratory Directed Research and Development Program funded by Lawrence Berkeley National Laboratory, the Watershed Function Scientific Focus Area funded by the U.S. Department of Energy, Office of Science, Office of Biological and Environmental Research, and the Atmospheric Radiation Measurement User Facility Program of the U.S. Department of Energy, Office of Science, Office of Biological and Environmental Research, under U.S. Department of Energy Contract No. DE-AC02-05CH11231. Co-author Rhoades was funded by the Director, Office of Science, Office of Biological and Environmental Research of the U.S. Department of Energy Regional and Global Model Analysis (RGMA) Program through the Calibrated and Systematic Characterization, Attribution and Detection of Extremes (CASCADE) Science Focus Area (award No. DE-AC02-05CH11231), and the "An Integrated Evaluation of the Simulated Hydroclimate System of the Continental US" project (award no. DE-SC0016605).

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
