# Peer review of "Sensitivities of subgrid-scale physics schemes, meteorological forcing, and topographic radiation in atmosphere-through-bedrock integrated process models: A case study in the Upper Colorado River Basin"

_EGUsphere, 2022_

## Author Comment (AC1)

**Referee Comments  #1**

Summary
This study evaluates the influence of different meteorologic forcing (based on different reanalysis datasets), subgrid-scale physics schemes, and terrain shading on simulated hydrometeorology. They find that physics configurations result in more variance in simulated hydrometeorological conditions, and that meteorological forcing has a smaller impact. This type of sensitivity study is important to understanding where and how to focus further model development and observational field campaigns (as the authors note), and this particular study evaluates some sensitivities that I have not previously seen addressed. In my view, this has the potential to be a highly valuable contribution, but I believe it could use some sharpening of its framing, earlier recognition of the problems across all model configurations with respect to streamflow simulation, and more quantitative comparisons of some of the results.

Response: Thank you for acknowledging the contribution and novelty of this paper.  We appreciate the subsequent suggestions to sharpen the framing, recognize issues across all model configurations with respect to streamflow simulation, and look forward to developing more quantitative comparisons of some of the results.

Major comments
One major comment is around framing: at times, the authors imply that these results show an optimal IPM configuration but this is never clearly evaluated. At other times, the authors note that validation against observations is not a major goal of this study – in which case, it cannot indicate an optimal IPM configuration. My recommendation is to avoid implying that an optimal configuration is identified here. On a related note, I think the poor simulation of streamflow by the IPM should be mentioned earlier (perhaps in the abstract) – while it's ok that this is the case, having this result buried in Figure 7 felt a bit deceptive.

Response: We agree that framing is important and that the phrasing "optimal configuration" needs to be avoided.  The manuscript can be reframed to avoid implying WRF configuration optimization by making sure that the reader is aware that the space of WRF configurations is not exhaustively or even parsimoniously sampled, given the infeasibility of evaluating all the WRF configurations. Of the configurations we investigated, we found that the BSU_CFSR exhibits the most skill with respect to the observations and observational products we sampled – this wording was modified accordingly. Furthermore, per this reviewer's suggestion, we can present information about the streamflow simulation by the IPM earlier in this paper. We can also revise the paper to discuss the bias in streamflow simulation and report the quantitative measure of the biases in simulating streamflow.

Similarly, the authors refer in the introduction to recent arguments by Lundquist that models may be outperforming observations. In my view, they then miss a relatively easy opportunity to contribute to this debate: adding a ParFlow-CLM run forced by PRISM and reporting the results

in Figure 7 would provide a case study testing whether meteorological models or observations are indeed more accurate in this case (assuming we basically believe that ParFlow-CLM is not biasing the results so much as to invert this response). I'm loathe to be the reviewer who suggests the authors do a different study than the one they have done – but in this case, the introduction led in this direction, and one additional simulation would significantly enhance the value of the present work.

Response: We thank the reviewer for the insight regarding the opportunity to contribute to the debate highlighted in Lunquist et al, 2019, BAMS. We appreciate the suggestion and agree that ParFlow-CLM run forced by PRISM will enable us to make statements about whether, in this particular domain over this particular water year, using a state-of-the-science surface/subsurface hydrology model forced with observational-based products exhibits superior or inferior performance with respect to streamflow observations, as compared to that same hydrological model when it is forced with atmospheric process model simulations.

Finally, it would have been useful to see more quantitative model evaluation, and some description of model evaluation in the methods. I had two specific concerns about the identification of BSU-CFSR2 as the "best" model and that used for the topographic radiation evaluation. First, I didn't see a quantitative evaluation of models against PRISM to make this evaluation. Why not report an NSE or RMSE? Second, given the idea that PRISM is not necessarily more accurate than WRF, I'm not sure how important PRISM is as a benchmark here. Could you analyze the impact of topographic shading for two model configurations with very different results? Assuming you find similar results for a different configuration, it would just be helpful to have a sentence confirming that evaluating topographic results in a different WRF configuration had similar results.

Response: We agree with the reviewer and can report the quantitative measurements (e.g., RMSE against PRISM) in the revision. While uncertainty exists in the PRISM product, it has been evaluated against SNOTEL measurements and chosen as the benchmark for the assessments in this paper. Furthermore, there are a few SNOTEL sites that provide observations in the vicinity of the East River Watershed: the Butte station is within the Watershed, the Schofield Pass station is only a few km north of the East River Watershed northern boundary and captures some of the north-south gradient in snowfall while the Taylor Park Reservoir station is only a few km east of the East River Watershed and captures some of the east-west variability in snowfall.

As stated in the original version of the paper, the impacts of topographic shading had a minimal effect within the context of spatial-aggregated hydroclimate variables but had a significant effect for spatially resolved radiation fluxes which nonlinearly affect temperature and snowmelt. The topo_shading and slope_rad only redistribute the radiation flux on the topographic edges, but maintain the total surface energy budget at the watershed-scale. We can clarify and expand on these points in a revised manuscript.

Minor comments

Line 21 – Based on only the abstract, it's not clear to me how the "spatiotemporal variance in simulated hydrometeorological conditions" is defined. I think you mean the model response varies more across the model structure options than meteorologic – but from this sentence, another possible interpretation is that spatiotemporal variance itself (e.g., the variance of some response variable across grid cells) is greater in certain physics schemes. Is there a way to avoid this ambiguity?

Response: We are referring to the variances of simulated hydrogemeterological outputs over multiple sub-grid physical schemes or meteorological forcings in the WRF model. We can modify this sentence in the revised manuscript by clearly explaining the hydrometeorological variables that are used to analyze the numerical experiments in this study.

Line 27 – The conclusion that these findings provide guidance on the most accurate IPM was a bit of a jump from the prior sentences, which just described model sensitivity. To justify this, it would be better to describe what analysis supports this guidance (a calibration, I presume? Against what variables?). Alternatively, your concluding point could note that these sensitivity analyses show where more effort should be focused to constrain our process-based understanding.

Response: We concur with the Reviewer that the language regarding IPM accuracy requires modification in the revised manuscript.  The Reviewer's suggestion to reframe the concluding point to highlight the fact that these sensitivity analyses do help guide the scientific community as it develops observational constraints on process-understanding is very well-taken and we would like to include it in the revised manuscript.  To that point, our finding that the atmospheric drivers of uncertainty in discharge, ET and other hydrologic variables are associated with processes occurring within the study domain and not external to it (i.e., uncertainty in physical processes in the Upper Colorado River at the watershed scale dominates surface hydrological variable uncertainty over the uncertainty associated with large-scale dynamics that set the initial and boundary conditions of these watersheds) provides support for future research directions.

We view our study as a first exploration of these topics and we look forward to working more in the coming years with hydroclimatic scientists interested in advancing the predictive understanding of Upper Colorado River water balance to improve regional, continental, or even Earth System scale dynamics modeling. The sensitivity analyses that we performed here do show the value of observational constraints on process-based understanding and where the scientific community should be focusing its efforts moving forward.  We found definitively that the scientific community should be focusing on what is going on in the watersheds specifically, rather than focusing on improving large-scale meteorological prediction.  It is a significant finding and can be emphasized in the revised manuscript in the context of discussing the development of observational constraints on process-models.

Future work (such as the use of more configurations, forcings, water years, and watershed locations) could help to support this hypothesis. Future works are going to be focusing on using a baseline configuration for future process-based research concurrent to SAIL, rather than a

suite of configurations that we explored here. The reason for this is that we need to be sure to reference this manuscript for the WRF runs you are doing in support of SAIL.

Line 34 – remove "that"
Response: We agree with the suggestion and look forward to revising the manuscript per this suggestion.

Line 35 – "may have"? Could you express the reason this is stated with uncertainty?
Response: This is an estimation from multiple sources mentioned in (Milly and Dunne, 2020). We look forward to revising the manuscript to clarify this point.

Line 44 – Is "relevant" here meaning for larger-scales? Or respective relevant scales for each process?
Response: "relevant" means the observational datasets can be only used to improve the understanding of physical processes at their respective scale.

Line 46 - Tying the motivation for this article to recent discussions about the relative skill of process-based atmospheric models vs gridded interpolated datasets provides a great motivation for the present study.
Response: We agree with the reviewer and can add the review of climate model assessments against a few reanalysis datasets, specifically in the complex-topography Rocky Mountains/UCRB regions.

For examples, those references can be included in the revision
Alder, J. R., & Hostetler, S. W. (2019). The dependence of hydroclimate projections in snow‐dominated regions of the western United States on the choice of statistically downscaled climate data. Water Resources Research, 55(3), 2279-2300.
Buban, M. S., Lee, T. R., & Baker, C. B. (2020). A comparison of the US climate reference network precipitation data to the parameter-elevation regressions on independent slopes model (PRISM). Journal of Hydrometeorology, 21(10), 2391-2400.
Rahimi, S., Krantz, W., Lin, Y. H., Bass, B., Goldenson, N., Hall, A., ... & Norris, J. (2022). Evaluation of a Reanalysis‐Driven Configuration of WRF4 Over the Western United States From 1980 to 2020. Journal of Geophysical Research: Atmospheres, 127(4), e2021JD035699.

Line 58 – "To further compound…" I think this is a good point, but could you provide an example?

Response: First, the snow processes cross-scale interactions are hard to manage and often necessitate downscaling of WRF to force snow process models at the scale they need to be run. One reference for this is Winstral and Marks, 2014 (Winstral, A., and Marks, D. (2014). Long-term snow distribution observations in a mountain catchment: assessing variability, time stability, and the representativeness of an index site. Water Resour. Res. 50, 293–305. doi: 10.1002/2012WR013038). Also, Siirila-Woodburn et al. (2021) has provided detailed reviews of

the challenges of managing the scales of subsurface process modeling with the scales of atmospheric process modeling.

Citations:
Winstral, A., & Marks, D. (2014). Long‑term snow distribution observations in a mountain catchment: Assessing variability, time stability, and the representativeness of an index site. Water Resources Research, 50(1), 293-305.
Siirila-Woodburn, E. R., Rhoades, A. M., Hatchett, B. J., Huning, L. S., Szinai, J., Tague, C., ... & Kaatz, L. (2021). A low-to-no snow future and its impacts on water resources in the western United States. Nature Reviews Earth & Environment, 2(11), 800-819.

Line 68 – I'm a little uncomfortable with "properly-configured" unless you feel this analysis truly fixes equifinality issues. Maybe "appropriately-configured"?

Response: We will reword "properly-configured" to "appropriately-configured".

Line 120 – "We can establish" leaves the reader uncertain if you did this or not.

Response: We will change "We established" to "We can establish".

Line 121-126 – This motivation is very nicely stated (although I don't think it's a hypothesis in the context of this study) – could you state this explicitly in the abstract?

Response: We agree with the reviewer on this point. We look forward to revising the manuscript to state the hypothesis of this paper by stating "Our hypothesis is that synoptic-scale forcings produce a much larger spread in surface-through-subsurface hydrology fields than subgrid-scale physics scheme choice." We will then clarify the text to walk the reader through the implications if the hypothesis is confirmed or falsified by stating: "If our hypothesis is confirmed, then scientific efforts to advance the predictive hydrology, through modeling, of the UCRB should prioritize improving large-scale weather products and analyses. Conversely, if the hypothesis is falsified, model subgrid-scale physics scheme choice produces more variability in hydrologic response, so scientific efforts should prioritize development of smaller scale atmospheric and hydrological processes affected by surface heterogeneity in the ERW."

We also look forward to explicitly stating this hypothesis in the abstract of the revised manuscript.

Line 127 – Is "observations" here meant to refer to gridded reanalysis products? As the Lundquist paper points out, those are also models (generally statistical interpolations), so I'd suggest another word. I also note that this section doesn't say anything about identifying an optimal model configuration, which is an outcome highlighted in the abstract.

Response: We agree with the reviewer that the "observations" here can be misleading, and can reword the revised manuscript with the following language: "regridded reanalysis products and

in-situ sensor measurements". We also look forward to revising the manuscript to briefly summarize the objectives and outcomes of this paper at the end of the introduction section.

Line 141 – "representative" is a bit of a tough argument to make – consider "similar to many other basins in…"

Response: We agree with the reviewer and look forward to revising the manuscript with the suggested language.

Line 141 – "near" should be "nearly"

Response: We agree with the reviewer and look forward to revising the manuscript with the suggested language.

Line 153 – You noted a lack of observations earlier, which disconnects somewhat with the "heavily-instrumented" claim here. I think this could be mitigated by noting that the instrumentation is intense at this site, but it's extremely difficult to observe many processes with high accuracy at relevant scales.

Response: We agree with the reviewer and look forward to comparing our simulations with the precipitation, temperature and ET observations measured by the in-situ instrumentation in the East River watershed. We will add this tothe manuscript with the suggested language by noting the mismatch in scales directly measured by instrumentation.

Figure 1- As I read through the rest of the paper, I found I needed a more detailed study area map for the ERW specifically – with elevation and streamlines, perhaps?

Response: We appreciate the sentiments of the reviewer and will now include a Google Earth overlay of watershed boundaries and streams in the revised manuscript as an additional Supplementary Material figure.

[Figure]

Line 254 – I have trouble understanding why PRISM was used to assess model performance for meteorological fields, given the comments in the Lundquist et al. (2019) paper you cited. It seems fine to compare against PRISM, but perhaps not to "assess model performance."

Response: We included PRISM here as a reference dataset because it is one of the most widely-used gridded observationally-based datasets at sufficient resolution (800 meters) to evaluate the heteorgeneity of the UCRB. At the same time, we recognize the very issues that Lundquist et al. (2019) raised about this dataset, since those issues were strong factors in motivating the research described in this manuscript.  We look forward to revising the manuscript to modify the language from "... was used as the reference dataset to assess model performance of precipitation and temperature in this study" to "... was used as a point of comparison in evaluating model uncertainty across sub-grid physical schemes and meteorological forcing datasets for precipitation and temperature in this study."

Line 266 – Could you note the spatial resolution of the ASO product used here? At 50 m, point-to-grid errors could be one reason for the apparent underestimation by ASO relative to SNOTEL.

Response: The raw ASO product has 50 meters spatial resolution. The ASO product is regridded to the same grid resolution as WRF outputs (500 meters) for comparison purposes. We acknowledge that the underestimation by ASO could be due to the point-to-grid errors and

look forward to addressing this issue in the revision. We recognize the research on gridding SWE data (e.g., Fassnacht et al, 2003, doi: https://doi.org/10.1029/2002WR001512 and Dozier, 2011, doi:https://doi.org/10.1029/2011EO430001) and have followed the approach of the linear interpolation of the ASO data as documented in Oaida et al, 2019, doi:https://doi.org/10.1175/JHM-D-18-0009.1 and look forward to including this detail in the revised manuscript.

Line 272 - Results section would be easier to follow with if subheadings were included.

Response: We can add subtitles as "4.1. Sub-grid physical schemes vs meteorological forcings", and "4.2. 3D topographic radiation effects".

Figure 3 caption – would read more easily if you noted a-c in your descriptions of which variables are identified. The statistics used to evaluate these differences are essentially introduced in this caption; could you move that to the methods?

Response: We agree with the reviewer and can add labels in the captions, and also to describe the statistical methods in the methods section.

Figure 3 – I'm surprised the UCD configurations melt so much earlier when they don't appear to be warmer. Is it possible that the spatial averages here obscure spatial differences that would explain why the UCD simulations melt earlier? Figure S-4 kind of gets at this, but I think it needs more interpretation for the reader.

Response: We agree with the reviewer that the spatial average visualization does not show the importance of locally specific spatial differences and, therefore, is unable to explain the physical reasoning of earlier snowmelt. We can create another supplement figure of the locally specific spatial differences in the UCD configuration simulation, and add a brief discussion of the physical reasons that may have given rise to an earlier snowmelt.

Line 319 – run-on sentence.

Response: We can modify this sentence for clarity.

Figure 4 – Nice figure. Could you again add an introduction to these statistics in the methods so we know how you're evaluating variance earlier? Why do c and d have only two points marked on the x-axis?

Response: Thank you for the kudos about this figure! We look forward to revising the manuscript accordingly. The x-extents of a-d are identical; we can add the additional tick-marks to avoid confusion.

Line 335 – Were there any quantitative statistics provided to determine that BSU-CFSR2 agreed best with PRISM?

Response: We look forward to adding to the revised manuscript the quantitative statistic of RMSE (Root Mean Squared Error), as suggested earlier by the reviewer, for precipitation, temperature and SWE across all experiments to quantify how the BSU-CFSR2 configuration compared to the other configurations in terms of its agreement with PRISM.

Figure 6 – Some panels appear not to use their full color scale (e.g., Temperature). Is that due to outlier pixels? There's a lot of wasted white-space in these maps – why not use the full plotting area for each map?

Response: We can adjust the extent of the plotting area according to this comment.

Line 380 – This paragraph describes Figure 7, but the next paragraph also seems to introduce Figure 7 as though it's a new topic?

Response: This paragraph describes the variance of simulated streamflow across experiments, and the next paragraph introduces the comparison against in-situ streamflow observations. We look forward to revising the manuscript to add a better transition sentence to aid the reader in separating these paragraphs.

Line 401 – "The objective of this study is not to replicate the observations…" In that case, I strongly recommend changing the final sentence in the abstract, because that implies you're identifying the best model configuration.

Response: We agree with the reviewer and can revise the abstract to not explicitly state that the objective of this paper is to identify the optimal model configurations but rather an exercise of sensitivity analysis where one configuration will perform the best.

Line 413 – Are the differences notable or minimal? I would say minimal. Maybe better to describe quantitatively – you could note the among-model variance vs the seasonal variance?

Response: The reviewer is correct that the differences are minimal. Since this is a snow-dominated watershed and streamflow is predominantly controlled by snowmelt, the seasonal variance is not comparable with the among-model variances. We can revise the manuscript to describe the intra-model configuration variance in different seasons.

Line 417 – "are slightly larger…" The differences are twice as big for the subgrid-scale physics schemes but are small in both cases; I would suggest rephrasing to clarify.

Response: We agree with the reviewer and look forward to removing the word "slightly" in the revised manuscript to avoid any misunderstanding.

Line 420 – What is meant by "more muted-nature"? I think this sentence speculating about differences in groundwater signals across years would be better in the discussion.

Response: We can re-word this sentence and perhaps use the wording "less noisy" opposed to "more muted" to avoid any confusion. What's meant here is that streamflow signals are very reactive, noisy, and change quickly, whereas groundwater signals are the product of slower processes via infiltration and vadose zone dynamics, often at longer timescales, which result in very different temporal signals as compared to streamflow.

Figure 8 – Is this color gradient perceptually uniform? It appears not to be (e.g., see Figure 1b in Cramer et al., 2020). It would be helpful to see a perceptually uniform palette here if possible.

Response: We can replace with the perceptually uniform color bar for the plots, based on the suggestions in Cramer et al. (2020).

Line 448 – "with an eye towards how to represent…" Without calibration or serious validation efforts, I don't think this study tells us about how to represent these interactions in models. I do think it tells us about where the most important uncertainties are, though (in your next sentence).

Response: Here we mean that by evaluating the model uncertainties for simulating precipitation, temperature, and streamflow, we are able to identify the which process within the model has the most important uncertainties. We can revise this sentence accordingly.

Line 454 – I don't remember a prior discussion of boundary conditions – is this referring to boundary conditions at the land surface driven by differences in the subgrid-scale physics schemes?

Response: Here we mean the large-scale forcing dataset used as the initial and boundary conditions in the WRF model. We can revise this sentence and explicitly mention that in the revised manuscript.

Line 456 – This would be more convincing if statistics on BSU-CFSR2 vs other models were presented. How does identifying this configuration allow researchers to prioritize process studies and observational constraints? What would these be, specifically?

Response: We agree with the reviewer and look forward to adding the quantitative statistics RMSE for the experiments against the PRISM observationally-based dataset.

Figure S6 – Could you use a different color scheme that doesn't have a diverging gradient? I think the diverging gradient is most appropriate for your maps showing differences (e.g., value scales that center on zero).

Response: We agree with the reviewer and can revise the color scale.

Line 467 – "Latent heat is posited…" by whom? Are you? I think you could state with more

confidence than "posit" that other energy balance components (including but not exclusively latent) mediate the influence of shortwave spatial variability on temperature spatial variability.

Response: We agree with the reviewer and can revise this sentence as "Latent heat buffers differences in the shortwave radiation contribution to the radiation budget."

Line 470 – You lost me here. This paragraph is ostensibly about how terrain shading algorithms affect radiation flux? How does this affect our ability to extrapolate findings from one mountainous watershed to another? The multiple "if" statements in here are also a little confusing – did the present study show these things or not?

Response:  This paragraph discusses the systemic cold bias in our current IPM configuration, and the limitations of one-way vs. two-way coupling between WRF and ParFlow-CLM.  We will revise this paragraph to be more clear and to the point.

References
Crameri, F., Shephard, G.E. & Heron, P.J. The misuse of colour in science communication. Nat Commun 11, 5444 (2020). https://doi.org/10.1038/s41467-020-19160-7

---

## Author Comment (AC2)

**Referee Comments  #2**

The integrated process model (IPM) which resolves the processes extending from the atmosphere through the bedrock is a hot topic in recent years. Using the IPM, researchers try to investigate the interactions between atmosphere and underground hydrological processes (e.g., lateral flows, groundwater dynamics), which used to be neglected by traditional meteorological modeling works. The ParFlow-CLM model is also a famous tool that couples the one-dimensional and sophisticated land surface model (CLM) with the three-dimensional groundwater model (ParFlow). Xu et al. Tested the sensitive of some hydrometeorological variables, which were simulated by the WRF model coupled with an integrated hydrologic model, to the choices of physical parameterizations, meteorological forcings that provide lateral boundary conditions, and terrain shading options. The author found that, physical parameterizations contributes to the largest spatial temporal variance in simulating the temperature, precipitation and other related hydrological variables. Although the topic is important and the introduction is well written, I still think the innovation is not strong and the manuscript needs major revision. My major concerns are below:

1. The author emphasize the necessity and importance of IPM in the introduction and also take the IPM as one of the innovation of this research. However, the simulation work is based on one-way coupling (use the WRF simulated meteorological forcings to drive ParFlow-CLM). Whether this one-way coupling can be called as IPM is confused, as there is no feedback between meteorology and the underground hydrology.

Response: We thank the reviewer for her/his comments noting the sophisticated tools used in this study, as well as the importance and high quality of the manuscript writing. The concern about the use of the phrase "integrated process modeling" is potentially one of semantics in this case, although we're happy to revise to avoid any potential confusion for the reviewer and/or other readers. There is a small body of literature that does use "integrated process model" terminology (e.g., Zhang et al, 2016 doi:https://doi.org/10.5194/hess-20-529-2016; Davison et al, 2017 doi:https://doi.org/10.1002/2017MS001052) that demonstrates the utility of coupling process models built to explore discipline-specific processes as a mechanism to advance interdisciplinary research.  There is also a literature comparing and contrasting one-way coupling vs two-way coupling for mountainous hydrology: e.g., Camera et al, 2020 doi:https://doi.org/10.5194/nhess-20-2791-2020 and Rudisill et al, 2022 doi:https://doi.org/10.1002/hyp.14578 where the latter paper finds that in snow-dominated watersheds such as the ERW, which found that the representation of uncertainties in the representation orographic precipitation is the single largest driver of hydrological uncertainty while the inclusion or exclusion of two-way coupling has little effect on atmosphere-through-bedrock state evolution. Nonetheless, to avoid confusion we would like to clarify that we are referring to a one-way coupled IPM (or WRF-Parflow-CLM) in the revised manuscript.

2. Another issue is that the finding that "physical parameterization is much more important than lateral or initial conditions"has been revealed by numerous works in meteorological discipline. For example, Solman and Pessacg (2012) found that the largest spread among WRF ensemble simulation members is caused by different combinations of physical parameterizations. Pohl et al (2011) tested the uncertainties of WRF simulation caused by physical parameterizations, lateral forcings, domain geometry. And they also suggested that physical parameterizations have the largest influence on precipitation. So, from the perspective of meteorology, the current finding is not surprising. The author should review the previous works and rethink the added value of the current work.

Response: We thank the reviewer for raising this issue and will add the cited references in the revision where appropriate, especially given the concurrence of our findings with those references. However, our findings are not redundant with the published literature: those references either evaluated large-scale meteorological processes or did not focus on high-altitude complex terrain regions, which are central to our study. Additionally, most previous studies do not show how the range of reasonable IPM configurations (based on configurations that have been presented in the published literature) affects discharge, ET and subsurface hydrology. We would argue that these aspects to our work represent an additional set of novel contributions that will be of interest to the readers of HESS.

With this set of one-way atmosphere-through-bedrock process modeling results, we can and have uncovered how choices in atmospheric process model configurations impact the surface and subsurface hydrology.  Specifically,  we evaluate and quantify the sensitivity of discharge, ET, and subsurface hydrology to IPM configurations, and we also address how 3D topographic radiation schemes affect both spatial distribution and spatial average hydroclimate simulations. More importantly, this study aims to guide the plan of field observational activities in the future by (1) uncovering how uncertainties in the representation of atmospheric processes impact surface and subsurface process modeling and (2) providing direction for those field observations with the greatest potential to constrain atmospheric processes. We will revise the paper to highlight the added value of the contributions of our work to the existing, relevant literature.

3. Since the ERW is a heavily-instrumented catchment with a growing atmosphere-through-bedrock observation network (emphasized in abstract) and the "The goal of this work is to provide the mountain hydrology research community with a properly-configured IPM that can inform ongoing and future field campaigns and their process-modeling needs in the UCRB.", why don't you use the in-situ observations to evaluation the T2m and precipitation.

Response: We agree with the reviewer and look forward to adding the comparison against the precipitation, two-meter air temperature and ET in-situ observations in the paper. We agree adding those comparisons against in-situ observations will help quantify the model performance in terms of surface air temperature and precipitation and will emphasize the value of

observational networks supporting model evaluation. The SAIL-based observations will be used in a future study to compare with IPM skill once the SAIL campaign is completed (2021-2023).

4. Moreover, I am really confused about the use of Parflow-CLM here. Is it used to only provide streamflow and groundwater storage? The simulated snow and ET a in re provided by CLM-Parflow or the default land surface model in WRF? Actually, the Parflow-CLM is often used to investigate the potential influence of three-dimensional groundwater on the responses of terrestrial hydrological processes to meteorological forcings (e.g., numerous high impact works performed by Maxwell and Condon). However, here, I did not see what will be different if we used the traditional one-dimensional land surface model to investigate the same issue. I suggested the author to compare the difference when using the results from default WRF land surface simulation and that from Parflow-CLM (such as ET, total water in the soil column). This may help enhance the innovation of current work.

Response: Yes, the primary reason for using ParFlow-CLM is to allow for the quantification of streamflow and groundwater storage. As the reviewer is probably aware, standalone WRF does not simulate these processes, although branches of the code (WRF-Hydro) do provide some insight into at least streamflow, although with a simplified and prescribed stream network. Groundwater in WRF-Hydro is highly simplified (shallow soil layers and a bucket model) while ParFlow simulates the full continuum of variably saturated flow in three dimensions.

We are interested here in developing, testing, and analyzing simulations with realistic representations of atmospheric processes and to explore how they interface with representations of surface and subsurface processes that are as realistic as possible. For this study, we would like to avoid complicating the analysis with additional model structural errors where we can, so that in the future, we can ultimately relax towards the more simplified representations of surface/subsurface processes in models such as WRF-Hydro.

The innovation of this work is to better understand how these choices in atmospheric parameterizations, forcing, and other configurations/options impact the greater hydrologic cycle. We respectfully disagree that the one-dimensional land surface model alone cannot yield this information, which is why we performed the additional modeling steps with ParFlow-CLM.

5. The experimental design needs more detailed information. I suggest the author to provide more introduction to the experimental design. For example, why do you only use the CFSR2 and ERA5 in the UCD and NCAR simulation? Why does the no3DRad_inner radiation scheme is only used in BSU_CFSR2 and BSU_ERA5?

Response: We designed the experiments with the intent to evaluate different subgrid-scale physical scheme configurations, atmospheric boundary forcings, and topographic specific subgrid-scale parameterizations. However, we recognized that this study would be computationally constrained given our prioritization of the use of sub-km horizontal resolution IPM simulations, hence why the model configuration matrix was not completely sampled. We

first chose to run a series of experiments with the three most prominent WRF subgrid-scale physical scheme configurations in the literature. We learned that the BSU configuration is the optimal physical scheme relative to the others chosen. As such, and due to computational restraints, we then chose this configuration to interrogate the topographic specific subgrid-scale parameterizations (e.g., noD3Rad_inner). We hope to fill some of the gaps in the simulation matrix in future work, particularly when we compare these simulations with the SAIL observational campaign (once completed in 2023). We plan on including more descriptive language on the experimental design in the revised manuscript.

6. I suggest to show the topograpnhy of the inner domain in Figure 1 which will be helpful to better understand the influence of 3D-radiative scheme. Currently, I am confused why the valley gets more radiation after considering the topographic shading and slope effect.

Response: To address this comment, we can upate Figure 1 and present the topography of the inner domain in more detail and providing more descriptive text for Figure 1. This will help show how and why the east side of the valley receives more radiation because it's west facing, and the western side of the valley gets less radiation in the early morning due to solar geometry. Similar conclusions have been observed in Arthur et al. 2018.

7. Moreover, the author should proofread the manuscript. For example:
● The Figure S-4 in L 299 should be Figure S1?
● "Figure S-3 and Figure S-4"should be "Figure S-3"?
● There is no description or analysis on the Figure 8c-8d.
● I also noticed some grids are masked out in Fig S-5 and Fig S-6, but no interpretation is given.

Response: We thank the reviewer for catching these typos and will revise all of them accordingly in our revision.

Reference:
Solman and Pessacg. (2012). Evaluating uncertainties in regional climate simulations over South America at the seasonal scale.
Pohl et al. (2011). Testing WRF capability in simulating the atmospheric water cycle over Equatorial East Africa

---

## Author Comment (AC3)

**Referee Comments #3**

This work presents a model study in a headwaters catchment in the Upper Colorado Watershed. In general the work is interesting and well-written but the presentation is somewhat confusing. There are some major points I think the authors need to address before the work's suitability for publication can be assessed. They are detailed below.

We thank the reviewer for his or her comments on this work being interesting and well-written. We have addressed all points the reviewer requests clarification on below.

General Comments:

-The terminology of so-called IMP's used is confusing along with the reference to these as coupled models, which I would argue they are not. It's very confusing to discuss this work in the framework of different models as opposed to just forcing used to drive the hydrologic model. The language around the different products used is very confusing and makes much of the discussion hard to follow. Some of the meterological forcing datasets appear to be used to drive the models direclty, but in the introduction it appears that only WRF simulations are used to drive models. A completely re-write of this entire section is needed to make this clear. What did the authors do with the hydrologic outputs from the WRF simulations? Why are the Noah and Noah-MP models used interchangeably but the results are not compared to ParFlow-CLM?

Response: We thank the reviewer for her/his comments noting the sophisticated tools used in this study, as well as the importance and high quality of the manuscript writing. The concern about the use of the phrase "integrated process modeling" is potentially one of semantics in this case, although we're happy to revise to avoid any potential confusion for the reviewer and/or other readers. There is a small body of literature that does use "integrated process model" terminology (e.g., Zhang et al, 2016 doi:https://doi.org/10.5194/hess-20-529-2016; Davison et al, 2017 doi:https://doi.org/10.1002/2017MS001052) that demonstrates the utility of coupling process models built to explore discipline-specific processes as a mechanism to advance interdisciplinary research. There is also a literature comparing and contrasting one-way coupling vs two-way coupling for mountainous hydrology: e.g., Camera et al, 2020 doi:https://doi.org/10.5194/nhess-20-2791-2020 and Rudisill et al, 2022 doi:https://doi.org/10.1002/hyp.14578 where the latter paper finds that in snow-dominated watersheds such as the ERW, which found that the representation of uncertainties in the representation orographic precipitation is the single largest driver of hydrological uncertainty while the inclusion or exclusion of two-way coupling has little effect on atmosphere-through-bedrock state evolution.

We agree that the use of the word "coupled" should be used with caution, as some of these models are formally coupled within their source-code infrastructure, and others are "step-wise coupled". To avoid any potential confusion, we have modified the manuscript to only use the

word "coupled" when there is a formal two-way and self-consistent pairing of codes (for example, ParFlow and CLM). In contrast, the associated meteorological variables used to drive ParFlow-CLM are now referred to as "forcing" only, as this is a one-way interaction between the codes/outputs.  Nonetheless, to avoid confusion we would like to clarify that we are referring to a one-way coupled IPM (or WRF-Parflow-CLM) in the revised manuscript.

The name of the meteorological products are the forcing datasets for the WRF runs, not the forcing data for ParFlow-CLM. Precipitation, temperature and other hydroclimate simulations from WRF are used as the atmospheric drivers for the ParFlow-CLM simulations. We will revise the manuscript to explicitly clarify that the meteorological forcing datasets are used as the initial and boundary conditions for the WRF model, and the WRF model outputs are then used to force ParFlow-CLM.

The Noah and Noah-MP hydrologic output from the WRF simulations included ET, surface and subsurface runoff. We agree with the referee (and referee #2 who identified this too) and will compare the Noah and Noah-MP simulated ET against ParFlow-CLM.  The primary reason for using ParFlow-CLM is to allow for the quantification of streamflow and groundwater storage. As the reviewer is probably aware, standalone WRF does not simulate these processes, although branches of the code (WRF-Hydro) do provide some insight into at least streamflow, although with a simplified and prescribed stream network. Groundwater in WRF-Hydro is highly simplified (shallow soil layers and a bucket model) opposed to ParFlow, which simulates the full continuum of variably saturated flow in three dimensions. Thus, while we indeed have generated multiple datasets with the different LSMs, this is not our primary objective for intercoparison and we believe showing more of these results would be distracting. We focus on the set of LSM outputs from WRF for simplicity (and to avoid confusion) and show the groundwater and streamflow outputs from ParFlow-CLM.

Except for groundwater which isn't discussed very much in the manuscript, all the same results should be in the WRF simulations.  Why didn't the authors just run the WRF-ParFlow model or even mention it's existence?  They talk about everything in a coupled sense but the models are in no way formally coupled (unless something is happening that is not discussed in the manuscript); the output files from WRF simulations are saved and somehow reformatted (this is not clear) and used to drive ParFlow-CLM.  They could drive any hydrologic model and it wouldn't be considered a coupled platform, likewise the standard forcing products the authors might choose to drive the simulations off the shelf are also generated with atmospheric models, yet I would never think of this as an IMP.  I suggest the authours are much more transparent about this aspect and remove the terminology from a revision.  They should also provide some clear language about what is actual being done here, is this a comparison between forcing generated with WRF v other approaches?  Why didn't the authors just run forced by PRISM?

Response: As is hopefully now more clear based on the last answer, we use WRF outputs to force ParFlow-CLM as a mechanism to simulate the integrated hydrologic response of the watershed. We do this because WRF is not capable of simulating two key processes of interest: not just groundwater storage changes but also streamflow. We did not run the fully coupled

WRF-ParFlow model platform because these simulations are extremely computationally expensive. We agree that it would be interesting to perform these 2-way coupled simulations in the future, but believe that an important first step is to determine what the hydrologic response is without the two-way interaction (i.e. WRF-forced ParFlow-CLM as we've done here). Fully coupled 2-way simulations are by no way the "standard" in hydrologic modeling, and the amount of fidelity we've included in this study stands to dwarf that of other approaches. That being said, we acknowledge that the existence of the code could be pointed to as an interesting next step. Two-way integrated meteorological and surface/subsurface hydrology simulations in complex terrain are especially limited, and a new frontier in the field.

Furthermore, we do appreciate the sentiments of the reviewer and will ensure that we will more clearly outline various methodological approaches taken in our model simulations and analyses in our revision.  For example, the WRF simulations were regridded using bilinear interpolation and used as the forcing dataset to run the ParFlow-CLM model.  Therefore, the simulations are, as the reviewer points out, a one-way coupled IPM.  We will ensure that we mention that WRF-Parflow is also available as a two-way coupled IPM too (and cite the relevant literature).

Finally, we agree with the reviewer and can run the ParFlow-CLM simulation forced by PRISM. We plan to regrid the 800-meter resolution PRISM daily precipitation and temperature variables and temporally interpolate them using an adjustment to the hourly distributions provided by the WRF hourly output. Other than precipitation and temperature (only variables made available from PRISM), we will use the WRF hourly output to force the ParFlow-CLM simulation.

-Coupled v uncoupled processes and feedbacks.  There has been a lot of work to understand the role of feedbacks between two-way coupled hydrologic models and atmospheric models. Examples include WRF-Hydro-WRF (e.g. Arnault 2016), COSMO-CLM-Parflow (e.g. Keune 2016, 2019), WRF-ParFlow (e.g. Maxwell 2012, Forrester 2020), feedbacks over complex terrain (Ban 2014),  and other more conceptual approaches (e.g. Miguez-Macho 2007).  This is not an exhustive list, but demonstrates that much work has been done to study these feedbacks,  Some of these studies are in complex terrain and even suggest that the approach used by the authors may not be valid at high resolution without lateral flow.  These studies all systematically compare different types of model physics (e.g. free drainage, the standalone atmospheric model, fully coupled system) and use varying metrics to diagnose coupling strength and changes in the atmosphere.  I suggest the authors read these prior studies carefully and develop a new section that summarizes (rather than ignores the existence of) this body of work and uses this to put the current study in context.  This will help frame the current work and help it look much less like a patchwork of runs that are loosely tied together.  This will also help clarify my point above, to help the reader follow what is being done and what runs are conducted in the current work.

Response: We agree with reviewer that there are multiple two-way coupled IPM configurations and will ensure that our revision better contextualizes our one-way coupled IPM configuration within the broader two-way coupled literature. We can develop a new paragraph in the introduction to demonstrate the details of those models in the paper, and

-Variability in point processes compared to integrated or averaged measures. I mention this as a specific instance below, but it is also a general point, there are instances where the authors present differences locally (at a point) that do not persist synoptically. Do the different forcing products or microphysics (I think this is the point the authors make) make some difference locally for e.g. precip, radiation, but does some averaged quantity remain unaffected. It appears this is the case for much of the analysis. That is, topographic shading makes a difference locally in LH flux but the domain averaged LH flux remains unchanged between cases. The authors draw one conclusion (local differences) without acknowledging the other (same net energy flux over the domain).

Response: We agree with the reviewer and look forward to adding more discussion of locally specific versus watershed average differences in our IPM results. For example, the importance oftopographic shading on the spatial distribution of radiation flux. The variances across different forcing products and subgrid-scale physical schemes are shown in our paper through both spatial distributions and average quantities. However, the 3D topographic radiation schemes only redistribute the energy flux thus affect LH flux spatial distribution, but do not show significant difference in the watershed average quantities. In the revision, we will more clearly highlight these points.

-Atmospheric uncertainty. There has been much work on differences in model physics in a model such as WRF that allows different physical parameterizations to be "swapped out" easily in simulations by changing the namelist. This is an important aspect of uncertainty, but it is almost always put in the context of one of the major forms of uncertainty in the atmosphere, propogation of intial conditions. One should always determine that such a physics change is robust using (e.g.) time-shifted uncertainty in an ensemble type approach (e.g. Walser 2004). Often upon inclusion of uncertainty in the intial model state (in the atmosphere) the differences in physical paramterization no longer dominate.

Response: We acknowledge the reviewer's sentiments about using a traditional experimental design to isolate structural versus internal variability uncertainties in climate modeling approaches, specifically through the use of initial condition perturbation ensemble members (internal variability) and holding subgrid-scale physics parameterizations constant (structural uncertainty). We will add a few sentences in the Methods section pointing to the reviewer's important points, explicitly cite Walser and Schaer (2004) and be more explicit about how our experimentally design differed from the more traditional approach. With that said, we also would like to acknowledge that although our experimental design didn't assess internal variability through the more traditionally employed initial condition perturbation approach (i.e., alter the initial conditions provided to the atmospheric model by a rounding error) we did isolate large-scale boundary conditions through the use of several atmospheric reanalyses that, in a way, also isolate slight differences in large-scale boundary conditions (e.g., integrated vapor transport).

We believe that a major finding of our work that hasn't necessarily been explored in the literature extensively over the East River watershed is that these slight differences in large-scale boundary conditions (analogous to perturbing initial conditions by rounding errors) did not markedly change water year precipitation totals unlike changes in subgrid-scale physical parameterization choice.

-Can the authors compare meterological forcings at the site? A heavily-instrumented catchment (abstract line 16-) should have observations of meterological variables and snow outside of the SNOTEL (which I don't think are used for comparison and should contain precipitation and temperature), even preciptation and temperature at gage locations would be very instrumental. It appears that the authors treat the PRISM product like observationsm, which is an unfortunate and hopefully accidental. The PRISM product is a model, even if statistical, that takes into account observations in a region. One would assume that then PRISM is ingesting precip from the SNOTEL sites in the domain but this isn't stated (are there even any obsevations that PRISM is using and is it thus totally unconstrained?).

Response: The large-scale meteorological forcings used in the WRF simulations have coarser grid resolutions than the WRF inner domain, so comparing IMP simulation results with meteorological forcing at the sites do not add scientific significance in model evaluation.

The reviewer is right about the PRISM product that it used SNOTEL datasets and was generated using statistlcal methods. PRISM dataset has been widely used to evaluate climate models in the complex-terrain region, with many thoughtful uncertainty analysis (e.g., Lunquist et al., 2020). We used interpolated precipitation and temperature product PRISM to evaluate the model performance over the whole domain. In the supplementary material, we presented the comparison of precipitation and temperature against snotel stations. In the paper revision, we can compare with in-situ observation data, including precipitation and temperature measurements taken at sensors installed in the East River watershed. We did not provide these in the initial manuscript to avoid distracting from the main message of the paper, but see the relevance and could add these to the supplemental in the revision.

-The authors should compare to ET observations in Ryken et al 2022 to results of current work (both WRF and ParFlow-CLM). Additionally, it apears that the Ryken et al 2022 paper has meterological observations of precip, temperature and radation that might be useful to partly address my comment above.

Response: We agree with the reviewer and can add the comparison against the ET measurement at the eddy-covariance tower at the East River, but again do not want to distract from the main purpose of this study by making this an ET intercomparison so this can potentially be added to the appendix

Specific comments

line 65: is PF-CLM being cited using Maxwell et al 2015 (cited on line 617)?  That paper references a simulation over large scale that as I read it is forced externally and does not use or describe the CLM model.

Response: Yes, while this is the same ParFlow, we agree a better citation can be used, the standard are: (Ashby and Falgout, 1996; Jones and Woodward, 2001; Maxwell, 2013). We will cite more appropriately in the revised version, including explicit mention of previous version of the model in the East River (Maina et al., 2022; Foster and Maxwell, 2018; Pribulick et al., 2016).

Citations:
Ashby, S.F., Falgout, R.D., 1996. A parallel multigrid preconditioned conjugate gradient algorithm for groundwater flow simulations. Nucl. Sci. Eng. 124 (1), 145–159.
Kuffour, B. N., Engdahl, N. B., Woodward, C. S., Condon, L. E., Kollet, S., & Maxwell, R. M. (2020). Simulating coupled surface–subsurface flows with ParFlow v3. 5.0: capabilities, applications, and ongoing development of an open-source, massively parallel, integrated hydrologic model. Geoscientific Model Development, 13(3), 1373-1397.
M. Foster, L., & M. Maxwell, R. (2019). Sensitivity analysis of hydraulic conductivity and Manning's n parameters lead to new method to scale effective hydraulic conductivity across model resolutions. Hydrological Processes, 33(3), 332-349.
Jones, J.E., Woodward, C.S., 2001. Newton–Krylov-multigrid solvers for large-scale, highly heterogeneous, variably saturated flow problems. Adv. Water Resour. 24 (7), 763–774.
Maina, F. Z., Wainwright, H. M., Dennedy-Frank, P. J., and Siirila-Woodburn, E. R.: On the similarity of hillslope hydrologic function: a clustering approach based on groundwater changes, Hydrol. Earth Syst. Sci., 26, 3805–3823, https://doi.org/10.5194/hess-26-3805-2022, 2022.
Maxwell, R.M., 2013. A terrain-following grid transform and preconditioner for parallel, large-scale, integrated hydrologic modeling. Adv. Water Resour. 53, 109–117.
Pribulick, C. E., Foster, L. M., Bearup, L. A., Navarre‐Sitchler, A. K., Williams, K. H., Carroll, R. W., & Maxwell, R. M. (2016). Contrasting the hydrologic response due to land cover and climate change in a mountain headwaters system. Ecohydrology, 9(8), 1431-1438.

Line 240+ This section describes the PF-CLM model in general but I could not find specifics for the model domain used in this study?  What is the resolution or model configuration for the PF-CLM domain?  How deep is the subsurface?  What is the lateral resolution?  How was this matched to the forcing datasets or the WRF outputs?  Was there a balance of water and fluxes between the grids?  How were model parameters determined?  Are there references to prior work on this model?  Calibration?  If not the authors might include a description of these aspects in the current manuscript and as supplemental material.

Response: The ParFlow-CLM subsurface domain is 30-meter deep, 100-meter horizontal resolution. The WRF output are re-gridded using bilinear interpolation to match the ParFlow-CLM grid cells. The model parameters are based on a variety of geological and soil parameters, and calibarated using streamflow measurements. More details can be found in some previous

papers (e.g., Foster et al., 2019, Pribulick et al., 2016, see the answers to the previous question). We will add more detailed descriptions of the ParFlow-CLM model in the manuscript

Lines 275-281.  The UCD datasets appear to have the most precip but the ear
Figure 3 caption (~line 305): a, b, c are used to identify plots in the figure but are not used in the caption.  Also, it does not appear that 3c is described in the caption.

Response: We agree with the reviewer and can revise the figures accordingly.

Figures 5, 6 and associated discussion.  An interesting point that might be made here is that while local spatial differences are apparent in Figure 6, the domain averages (even for SWE) are the same between shaded and non-shaded formulations.  This suggests that while it may be striking visually to include shading, the upscaled water balance for the catchment isn't sensitive.

Response:  We agree with the reviewer that turning on and off the 3D shaded radiation scheme do not significantly affect the domain average. The reviewer is correct that the 3D shading radiation scheme do not affect the upscaling water balance, but redistribute the spatial distribution of radiation fluxes thus SWE and energy spatial pattern. The east side of the valley gets more radiation because it's facing west, and the western side of the valley gets less radiation as it gets less radiation in the early morning. Similar conclusions have been observed in Arthur et al. 2018.

lines 383- I'm not sure I agree with these conclusions.  While the cumulative variability in outflow resulting from the different forcing products creates different cumulative outflows, Figure 7a indicates that there is no difference in timing across all the forcing datasets.  My suspicion is that the differences in outflow are due to total water quantity (Figure 5a suggests this as well) and are simply a precip bias artifact in the different WRF runs.

Response: We agree with the reviewer that Figure 7a indicates the difference in timing in all the IPM across all forcing datasets. The difference of precipitation in WRF simulations leads to the differences in streamflow simulation in ParFlow-CLM. We can revise this sentence to clarify and avoid misunderstanding.

line 443: "Here, we ... coupling WRF and ParFlow" rephrase, this sentence isn't correct, the models were not coupled

Response: We can reword this as 'one-way coupling IPM' or "WRF-ParFlow-CLM" to avoid misunderstanding.

line 476+: This text appears to acknowledge the lack of coupling in the current work (as an aside, what is "one-way coupled" this is not actually coupled at all, as it appears the results from WRF were simply used as forcing for the PF-CLM model.  This isn't bad, but as mentioned above should be discussed up front.  The arguments here regarding computational expense as an excuse for not running coupled simualtions are incorrect, prior studies have shown with the

e.g. WRF-PF model that ParFlow is approximately 1% of the total computational time compared to WRF which is 99% of the computational time.  Thus if the authors ran WRF for this domain, the additional expense to run with WRF-PF is a negligible increase in cost.  Also the authors might want to correclty identify that the Forrester et al study (line 483) was run with WRF-PF and the authors might want to read and cite Forrester 2020 which discusses limitations of running high resolution, uncoupled WRF simulations in mountain terriain (the CO headwaters was studied) where the lack of lateral flow caused changes in the surface energy budget and hight of the boundary layer.

Responses: The reviewer is correct that the WRF simulation used most of the computational and throughput time. However, we are unaware of any study which shows only a 1% CPU demand for ParFlow-WRF opposed to standalone WRF. Personal communication with the primary ParFlow developer confirmed this is not a strict rule of thumb, and that there are various factors which determine the additional CPU time to include ParFlow in a WRF run. This is especially true in complex terrain where there are sharp wetting fronts and higher demands on the Richards' equation solver. Thus, it's difficult to know the exact additional demands of the fully coupled code without performing the simulations.

We agree to update the statement as "one-way coupling IPM" in the revision. We also concur with the reviewer that computational expense is not the major excuse for not running coupled simulations, moreso that we wished to establish a baseline set of simulations without fully coupled feedbacks before considering the 2-way interacting model.

We agree with the reviewer to cite Forrester 2020 paper, which discusses how the subsurface groundwater flow in ParFlow-CLM (lateral groundwater flow and subsurface lower boundary conditions) affects the atmospheric model. This impacts were particularly enhanced during summertime, while we run the simulation for a whole water year. We agree to highlight the findings from previous fully-coupled WRF-PF model, and extend the discussion of one-way and two-way coupling in the revision.

line 498: Is the watershed highly instrumented now or will it be?  This seems at odds with statement in the abstract (line 16)?

Response: To clarify, the East River watershed is already highly instrumented due to the presence of the long-standing Rocky Mountain Biological Laboratory (RMBL), the SNOTEL network, and DOE Watershed Science Focus Area project which has been adding instrumentation to the watershed over the last ~7 years. While these observations focus primarily on surface and subsurface processes, the East River watershed has become even more instrumented in recent years (2021-2023) through the support of the U.S. DOE (SAIL campaign) and U.S. NOAA (SPLASH campaign) deployments of a comprehensive set of atmospheric instrumentations (e.g., radar and radiation measurements).

However, these SAIL and SPLASH campaign measurements were provided after our simulations were conducted and we began to prepare this manuscript.  In future work, we plan

to build upon the knowledge learned from this manuscript to compare the most optimally configured IPM to SAIL and SPLASH campaign observations. We can revise this sentence to best reflect the added significance of SAIL and SPLASH campaigns in future IPM studies in the UCRB region, and look forward to using those datasets to improve mountainous hydrologic cycle process understanding and model development.

references

Arnault, J., Wagner, S., Rummler, T., Fersch, B., Bliefernicht, J., Andresen, S., & Kunstmann, H, 2016: Role of Runoff–Infiltration Partitioning and Resolved Overland Flow on Land–Atmosphere Feedbacks: A Case Study with the WRF-Hydro Coupled Modeling System for West Africa, Journal of Hydrometeorology

Ban, N., Schmidli, J., and Schär, C. 2014: Evaluation of the convection-resolving regional climate modeling approach in decade-long simulations, Journal Geophyscal Research Atmosphere

Forrester, M. and Maxwell, R. 2020: Impact of lateral groundwater flow and subsurface lower boundary conditions on atmospheric boundary layer development over complex terrain, Journal of Hydrometeorology

Frei, C. and Schär, C. 1998: A precipitation climatology of the Alps from high-resolution rain-gauge observations. Inter ational Journal of Climatology,

Keune, J., Gasper, F., Goergen, K., Hense, A., Shrestha, P., Sulis, M. and Kollet, S. 2016: Studying the influence of groundwater representations on land surface-atmosphere feedbacks during the European heat wave in 2003, Journal of Geophysical Research - Atmospheres

Keune, J., Sulis, M., and Kollet, S. J. 2019: Potential added value of incorporating human water use on the simulation of evapotranspiration and precipitation in a continental  scale bedrock  to   atmosphere modeling system–A validation study considering observational uncertainty. Journal of Advances in Modeling Earth Systems

Miguez-Macho, G., Y. Fan, C. P. Weaver, R. Walko, and A. Robock, 2007: Incorporating water table dynamics in climate modeling: 2. Formulation, validation, and soil moisture simulation. Journal of Geophysical Research

Maxwell, R.,  J. K. Lundquist, J. D. Mirocha, S. G. Smith, C. S. Woodward, and A. F. B. Tompson, 2011: Development of a coupled groundwater–atmosphere model, Monthly Weather Review

Ryken, A. C., Gochis, D., & Maxwell, R. 2022: Unravelling groundwater contributions to evapotranspiration and constraining water fluxes in a high-elevation catchment. Hydrological Processes

Walser, A., and C. Schaer, 2004: Convection-resolving precipitation forecasting and its predictability in Alpine river catchments, Journal of Hydrology

---

## Author Comment (AC4)

**Referee Comments  #1**

Summary
This study evaluates the influence of different meteorologic forcing (based on different reanalysis datasets), subgrid-scale physics schemes, and terrain shading on simulated hydrometeorology. They find that physics configurations result in more variance in simulated hydrometeorological conditions, and that meteorological forcing has a smaller impact. This type of sensitivity study is important to understanding where and how to focus further model development and observational field campaigns (as the authors note), and this particular study evaluates some sensitivities that I have not previously seen addressed. In my view, this has the potential to be a highly valuable contribution, but I believe it could use some sharpening of its framing, earlier recognition of the problems across all model configurations with respect to streamflow simulation, and more quantitative comparisons of some of the results.

Response: Thank you for acknowledging the contribution and novelty of this paper.  We appreciate the subsequent suggestions to sharpen the framing, recognize issues across all model configurations with respect to streamflow simulation, and look forward to developing more quantitative comparisons of some of the results.

Response after revision: Please find those responses after revision in blue

Major comments
One major comment is around framing: at times, the authors imply that these results show an optimal IPM configuration but this is never clearly evaluated. At other times, the authors note that validation against observations is not a major goal of this study – in which case, it cannot indicate an optimal IPM configuration. My recommendation is to avoid implying that an optimal configuration is identified here. On a related note, I think the poor simulation of streamflow by the IPM should be mentioned earlier (perhaps in the abstract) – while it's ok that this is the case, having this result buried in Figure 7 felt a bit deceptive.

Response: We agree that framing is important and that the phrasing "optimal configuration" needs to be avoided.  The manuscript can be reframed to avoid implying WRF configuration optimization by making sure that the reader is aware that the space of WRF configurations is not exhaustively or even parsimoniously sampled, given the infeasibility of evaluating all the WRF configurations. Of the configurations we investigated, we found that the BSU_CFSR exhibits the most skill with respect to the observations and observational products we sampled – this wording was modified accordingly. Furthermore, per this reviewer's suggestion, we can present information about the streamflow simulation by the IPM earlier in this paper. We can also revise the paper to discuss the bias in streamflow simulation and report the quantitative measure of the biases in simulating streamflow.

Response after revision: We revised "optimal configuration" as "assessed several literature supported configurations". We also revised the paper to discuss and report quantitative measures of bias. The abstract was revised as follows, "While the simulated discharge peaks are delayed earlier due to cold bias, discharge shows greater variance in response to the WRF simulations across subgrid-scale physics schemes (26%) rather than meteorological forcing (6%)"

Similarly, the authors refer in the introduction to recent arguments by Lundquist that models may be outperforming observations. In my view, they then miss a relatively easy opportunity to contribute to this debate: adding a ParFlow-CLM run forced by PRISM and reporting the results in Figure 7 would provide a case study testing whether meteorological models or observations are indeed more accurate in this case (assuming we basically believe that ParFlow-CLM is not biasing the results so much as to invert this response). I'm loathe to be the reviewer who suggests the authors do a different study than the one they have done – but in this case, the introduction led in this direction, and one additional simulation would significantly enhance the value of the present work.

Response: We thank the reviewer for the insight regarding the opportunity to contribute to the debate highlighted in Lunquist et al, 2019, BAMS. We appreciate the suggestion and agree that ParFlow-CLM run forced by PRISM will enable us to make statements about whether, in this particular domain over this particular water year, using a state-of-the-science surface/subsurface hydrology model forced with observational-based products exhibits superior or inferior performance with respect to streamflow observations, as compared to that same hydrological model when it is forced with atmospheric process model simulations.

Response after revision: We have performed an Parflow-CLM simulation forced with PRISM and added these results into Figure 7 and provided a few new sentences in the main text regarding these results. The discharge simulated in ParFlow-CLM shows a similar behavior with the hydrological simulations forced by WRF models. The text following text has been added into Section 3.2, "We also forced ParFlow-CLM with PRISM precipitation and temperature fields by evenly distributing daily precipitation and temperature across a diurnal cycle of 24 hours within a day."

Finally, it would have been useful to see more quantitative model evaluation, and some description of model evaluation in the methods. I had two specific concerns about the identification of BSU-CFSR2 as the "best" model and that used for the topographic radiation evaluation. First, I didn't see a quantitative evaluation of models against PRISM to make this evaluation. Why not report an NSE or RMSE? Second, given the idea that PRISM is not necessarily more accurate than WRF, I'm not sure how important PRISM is as a benchmark here. Could you analyze the impact of topographic shading for two model configurations with very different results? Assuming you find similar results for a different configuration, it would just be helpful to have a sentence confirming that evaluating topographic results in a different WRF configuration had similar results.

Response: We agree with the reviewer and can report the quantitative measurements (e.g., RMSE against PRISM) in the revision. While uncertainty exists in the PRISM product, it has been evaluated against SNOTEL measurements and chosen as the benchmark for the assessments in this paper.  Furthermore, there are a few SNOTEL sites that provide observations in the vicinity of the East River Watershed: the Butte station is within the Watershed, the Schofield Pass station is only a few km north of the East River Watershed northern boundary and captures some of the north-south gradient in snowfall while the Taylor Park Reservoir station is only a few km east of the East River Watershed and captures some of the east-west variability in snowfall.

As stated in the original version of the paper, the impacts of topographic shading had a minimal effect within the context of spatial-aggregated hydroclimate variables but had a significant effect for spatially resolved radiation fluxes which nonlinearly affect temperature and snowmelt. The topo_shading and slope_rad only redistribute the radiation flux on the topographic edges, but maintain the total surface energy budget at the watershed-scale. We can clarify and expand on these points in a revised manuscript.

Response after revision: We reported the quantitative measurements RMSE and R^2 for WRF precipitation and temperature simulations against PRISM in Table 1. We also clarified the 3D radiation effects in the text as, "The 3D radiation shading scheme does not significantly affect the total water balance, but rather the spatial distribution of radiation fluxes. Thus, despite having minimal impacts on the water balance, the scheme does have important localized impacts on SWE and surface energy budget spatial patterns", and, "In summary,  the simulations show that, while local spatial differences in surface radiation with and without realistic topography are apparent in Figure 6, the domain spatial averages (even for SWE) are the same between shaded and non-shaded formulations.  This suggests that while there may be striking localized differences when shading is included, the impact of topographic shading on the entire water balance over a spatial domain like the ERW is negligible" in section 4.2.

Minor comments
Line 21 – Based on only the abstract, it's not clear to me how the "spatiotemporal variance in simulated hydrometeorological conditions" is defined. I think you mean the model response varies more across the model structure options than meteorologic – but from this sentence, another possible interpretation is that spatiotemporal variance itself (e.g., the variance of some response variable across grid cells) is greater in certain physics schemes. Is there a way to avoid this ambiguity?

Response: We are referring to the variances of simulated hydrometerological outputs over multiple subgrid-scale physical schemes or meteorological forcings in the WRF model. We can modify this sentence in the revised manuscript by clearly explaining the hydrometeorological variables that are used to analyze the numerical experiments in this study.
Response after revision: This sentence has now been revised as follows, "Results reveal that subgrid-scale physics configuration lead to larger spatiotemporal variance in simulated hydrometeorological conditions"

Line 27 – The conclusion that these findings provide guidance on the most accurate IPM was a bit of a jump from the prior sentences, which just described model sensitivity. To justify this, it would be better to describe what analysis supports this guidance (a calibration, I presume? Against what variables?). Alternatively, your concluding point could note that these sensitivity analyses show where more effort should be focused to constrain our process-based understanding.

Response: We concur with the Reviewer that the language regarding IPM accuracy requires modification in the revised manuscript. The Reviewer's suggestion to reframe the concluding point to highlight the fact that these sensitivity analyses do help guide the scientific community as it develops observational constraints on process-understanding is very well-taken and we would like to include it in the revised manuscript. To that point, our finding that the atmospheric drivers of uncertainty in discharge, ET and other hydrologic variables are associated with processes occurring within the study domain and not external to it (i.e., uncertainty in physical processes in the Upper Colorado River at the watershed scale dominates surface hydrological variable uncertainty over the uncertainty associated with large-scale dynamics that set the initial and boundary conditions of these watersheds) provides support for future research directions.

We view our study as a first exploration of these topics and we look forward to working more in the coming years with hydroclimatic scientists interested in advancing the predictive understanding of Upper Colorado River water balance to improve regional, continental, or even Earth System scale dynamics modeling. The sensitivity analyses that we performed here do show the value of observational constraints on process-based understanding and where the scientific community should be focusing its efforts moving forward. We found definitively that the scientific community should be focusing on what is going on in the watersheds specifically, rather than focusing on improving large-scale meteorological prediction. It is a significant finding and can be emphasized in the revised manuscript in the context of discussing the development of observational constraints on process-models.

Future work (such as the use of more configurations, forcings, water years, and watershed locations) could help to support this hypothesis. Future works are going to be focusing on using a baseline configuration for future process-based research concurrent to SAIL, rather than a suite of configurations that we explored here. The reason for this is that we need to be sure to reference this manuscript for the WRF runs you are doing in support of SAIL.

Response after revision: The abstract was revised as "Our hypothesis is that synoptic-scale forcings produce a much larger spread in surface-through-subsurface hydrologic fields than subgrid-scale physics scheme choice. Results reveal that sub-grid scale physics configuration lead to larger spatiotemporal variance in simulated hydrometeorological conditions, whereas variance across meteorological forcing with common sub-grid scale physics configurations is more spatiotemporally constrained. While the simulated discharge peaks are delayed earlier due to model cold biases, discharge shows greater variance in response to the WRF simulations across subgrid-scale physics schemes (26%) rather than meteorological forcing

(6%). Topographic radiation option has minor effects on the watershed-average hydrometeorological processes, but adds profound spatial heterogeneity to local energy budgets (+/-30 W/m2 in shortwave radiation and 1 K air temperature differences in late summer).  This is the first presentation of sensitivity analyses that provide support to   help guide the scientific community to develop observational constraints on atmopshere-through-bedrock processes and their interactions."

Line 34 – remove "that"
Response: We agree with the suggestion and look forward to revising the manuscript per this suggestion.

Response after revision: Remove "that".

Line 35 – "may have"? Could you express the reason this is stated with uncertainty?

Response: This is an estimation from multiple sources mentioned in (Milly and Dunne, 2020). We look forward to revising the manuscript to clarify this point.

Response after revision: Removed "may have".

Line 44 – Is "relevant" here meaning for larger-scales? Or respective relevant scales for each process?

Response: "relevant" means the observational datasets can be only used to improve the understanding of physical processes at their respective scale.

Response after revision: This sentence has been revised as follows, "there is a dearth of observational data that can constrain these processes at their respective scales, which has resulted in persistent model simulation biases in predicting the mountainous hydrologic cycle with direct implications for water resource management".

Line 46 - Tying the motivation for this article to recent discussions about the relative skill of process-based atmospheric models vs gridded interpolated datasets provides a great motivation for the present study.

Response: We agree with the reviewer and can add the review of climate model assessments against a few reanalysis datasets, specifically in the complex-topography Rocky Mountains/UCRB regions.

For examples, those references can be included in the revision
Alder, J. R., & Hostetler, S. W. (2019). The dependence of hydroclimate projections in snow‐dominated regions of the western United States on the choice of statistically downscaled climate data. Water Resources Research, 55(3), 2279-2300.

Buban, M. S., Lee, T. R., & Baker, C. B. (2020). A comparison of the US climate reference network precipitation data to the parameter-elevation regressions on independent slopes model (PRISM). Journal of Hydrometeorology, 21(10), 2391-2400.
Rahimi, S., Krantz, W., Lin, Y. H., Bass, B., Goldenson, N., Hall, A., ... & Norris, J. (2022). Evaluation of a Reanalysis‐Driven Configuration of WRF4 Over the Western United States From 1980 to 2020. Journal of Geophysical Research: Atmospheres, 127(4), e2021JD035699.

Response after revision: A few additional references have been added, and the text is revised as follows, "A wide range of physics based and statistical models have been used over the complex terrain of the western U.S. For example, Alder et al. (2019) and Rahimi et al. (2022) have evaluated the choice of downscaled climate data and the sensitivities of grid resolution. Buban et al. (2022) also investigated the use of PRISM as a reference dataset to assess climate model performance. Observational campaigns, combined with coordinated modeling activities, represent a potential path forward towards enhancing our predictive understanding of the hydrologic cycle in complex terrain and, ultimately, advancing model development that can better aid water resource management (Lundquist et al, 2019; Feldman et al., 2021)."

Line 58 – "To further compound…" I think this is a good point, but could you provide an example?

Response: First, the snow processes cross-scale interactions are hard to manage and often necessitate downscaling of WRF to force snow process models at the scale they need to be run.  One reference for this is Winstral and Marks, 2014 (Winstral, A., and Marks, D. (2014). Long-term snow distribution observations in a mountain catchment: assessing variability, time stability, and the representativeness of an index site. Water Resour. Res. 50, 293–305. doi: 10.1002/2012WR013038). Also, Siirila-Woodburn et al. (2021) has provided detailed reviews of the challenges of managing the scales of subsurface process modeling with the scales of atmospheric process modeling.

Citations:
Winstral, A., & Marks, D. (2014). Long‐term snow distribution observations in a mountain catchment: Assessing variability, time stability, and the representativeness of an index site. Water Resources Research, 50(1), 293-305.
Siirila-Woodburn, E. R., Rhoades, A. M., Hatchett, B. J., Huning, L. S., Szinai, J., Tague, C., ... & Kaatz, L. (2021). A low-to-no snow future and its impacts on water resources in the western United States. Nature Reviews Earth & Environment, 2(11), 800-819.

Response after revision: Several new references are provided in the paper and the text is revised as follows, "Additionally, advances in process modeling in complex terrain must recognize connections between processes in the atmosphere, at the surface and in the subsurface. At the same time, making connections between processes across the atmosphere-through-bedrock continuum is highly non-trivial. Furthermore, snow processes must be resolved at much finer scales than atmospheric processes, such that snow process investigations and accurate snow process modeling requires high-resolution downscaling of

WRF (e.g. Winstral and Marks, 2014). Cross-scale interactions in complex terrain are challenging to resolve at their native scales with currently available advanced computing resources (Siirila-Woodburn et al., 2021)."”

Line 68 – I'm a little uncomfortable with "properly-configured" unless you feel this analysis truly fixes equifinality issues. Maybe "appropriately-configured"?

Response: We will reword "properly-configured" to "appropriately-configured".
Response after revision: revised as "appropriately-configured"

Line 120 – "We can establish" leaves the reader uncertain if you did this or not.

Response: We will change "We can established" to "We establish".
Response after revision: revised as "We establish"

Line 121-126 – This motivation is very nicely stated (although I don't think it's a hypothesis in the context of this study) – could you state this explicitly in the abstract?

Response: We agree with the reviewer on this point. We look forward to revising the manuscript to state the hypothesis of this paper by stating "Our hypothesis is that synoptic-scale forcings produce a much larger spread in surface-through-subsurface hydrology fields than subgrid-scale physics scheme choice." We will then clarify the text to walk the reader through the implications if the hypothesis is confirmed or falsified by stating: "If our hypothesis is confirmed, then scientific efforts to advance the predictive hydrology, through modeling, of the UCRB should prioritize improving large-scale weather products and analyses. Conversely, if the hypothesis is falsified, model subgrid-scale physics scheme choice produces more variability in hydrologic response, so scientific efforts should prioritize development of smaller scale atmospheric and hydrological processes affected by surface heterogeneity in the ERW."

We also look forward to explicitly stating this hypothesis in the abstract of the revised manuscript.

Response after revision: This hypothesis sentence has been added in the Abstract as follows, "Our hypothesis is that synoptic-scale forcings produce a much larger spread in surface-through-subsurface hydrologic fields than subgrid-scale physics scheme choice."

Line 127 – Is "observations" here meant to refer to gridded reanalysis products? As the Lundquist paper points out, those are also models (generally statistical interpolations), so I'd suggest another word. I also note that this section doesn't say anything about identifying an optimal model configuration, which is an outcome highlighted in the abstract.

Response: We agree with the reviewer that the "observations" here can be misleading, and can reword the revised manuscript with the following language: "regridded reanalysis products and

in-situ sensor measurements". We also look forward to revising the manuscript to briefly summarize the objectives and outcomes of this paper at the end of the introduction section.

Response after revision: Since the discussion of observation vs. reanalysis products are beyond the scope of this paper, we revised this sentence as follows, "Our hypothesis is that synoptic-scale forcings produce a much larger spread in surface-through-subsurface hydrology fields than subgrid-scale physics scheme choice. If our hypothesis is confirmed, then scientific efforts to advance the predictive hydrology, through modeling, of the UCRB should prioritize improving large-scale weather products and analyses. Conversely, if the hypothesis is falsified, model subgrid-scale physics scheme choice produces more variability in hydrologic response, then scientific efforts should prioritize the development of smaller scale atmospheric and hydrological process representations affected by surface heterogeneity in the ERW."

Line 141 – "representative" is a bit of a tough argument to make – consider "similar to many other basins in…"

Response: We agree with the reviewer and look forward to revising the manuscript with the suggested language.

Response after revision: Revised as suggested.

Line 141 – "near" should be "nearly"

Response: We agree with the reviewer and look forward to revising the manuscript with the suggested language.

Response after revision: Revised as suggested.

Line 153 – You noted a lack of observations earlier, which disconnects somewhat with the "heavily-instrumented" claim here. I think this could be mitigated by noting that the instrumentation is intense at this site, but it's extremely difficult to observe many processes with high accuracy at relevant scales.

Response: We agree with the reviewer and look forward to comparing our simulations with the precipitation, temperature and ET observations measured by the in-situ instrumentation in the East River watershed. We will add this tothe manuscript with the suggested language by noting the mismatch in scales directly measured by instrumentation.

Response after revision: The text has now been revised as follows to avoid any misunderstanding, "The ERW has become one of the most heavily-instrumented mountainous watersheds in the world, which makes it an ideal location for this research given the potentially large number of observational constraints available for the IPM efforts presented here. Although a wide range of precipitation, temperature and hydrological data have been collected, it is still

challenging to use these to characterize atmospheric, surface and subsurface processes and their interactions at relevant scales".

Figure 1- As I read through the rest of the paper, I found I needed a more detailed study area map for the ERW specifically – with elevation and streamlines, perhaps?

Response: We appreciate the sentiments of the reviewer and will now include a Google Earth overlay of watershed boundaries and streams in the revised manuscript as an additional Supplementary Material figure.

[Figure]

Response after revision: As shown in the figure above, we now have added a subplot of satellite imagery with watershed demarcations in Figure 1c.

Line 254 – I have trouble understanding why PRISM was used to assess model performance for meteorological fields, given the comments in the Lundquist et al. (2019) paper you cited. It seems fine to compare against PRISM, but perhaps not to "assess model performance."

Response: We included PRISM here as a reference dataset because it is one of the most widely-used gridded observationally-based datasets at sufficient resolution (800 meters) to evaluate the heteorgeneity of the UCRB. At the same time, we recognize the very issues that Lundquist et al. (2019) raised about this dataset, since those issues were strong factors in motivating the research described in this manuscript. We look forward to revising the

manuscript to modify the language from "... was used as the reference dataset to assess model performance of precipitation and temperature in this study" to "... was used as a point of comparison in evaluating model uncertainty across sub-grid physical schemes and meteorological forcing datasets for precipitation and temperature in this study."

Response after revision: This sentence has been revised in the following way, "The Parameter-elevation Relationships on Independent Slopes Model (PRISM) dataset (Daly et al., 2008) was used here as a point of comparison in evaluating model uncertainty across subgrid-scale physical schemes and meteorological forcing datasets for precipitation and temperature."

Line 266 – Could you note the spatial resolution of the ASO product used here? At 50 m, point-to-grid errors could be one reason for the apparent underestimation by ASO relative to SNOTEL.

Response: The raw ASO product has 50 meters spatial resolution. The ASO product is regridded to the same grid resolution as WRF outputs (500 meters) for comparison purposes. and look forward to addressing this issue in the revision. We recognize the research on gridding SWE data (e.g., Fassnacht et al, 2003, doi: https://doi.org/10.1029/2002WR001512 and Dozier, 2011, doi:https://doi.org/10.1029/2011EO430001) and have followed the approach of the linear interpolation of the ASO data as documented in Oaida et al, 2019, doi:https://doi.org/10.1175/JHM-D-18-0009.1 and look forward to including this detail in the revised manuscript.

Response after revision: Text was added and revised as follows to demonstrate the uncertainty of the ASO product in our model-to-observation comparisons, "The raw ASO product has 50 meters spatial resolution, and is regridded to the same grid resolution as the WRF outputs (500 meters) for comparison purposes using bilinear interpolation, as documented in Oaida et al. (2019). Since the spatial resolution of the ASO data is significantly finer than the WRF outputs, we acknowledge that the underestimation by ASO could be due to point-to-grid errors."

Line 272 - Results section would be easier to follow with if subheadings were included.

Response: We can add subtitles as "4.1. Subgrid-scale physical schemes vs meteorological forcings", and "4.2. 3D topographic radiation effects".

Response after revision: Added subsection subheading as suggested.

Figure 3 caption – would read more easily if you noted a-c in your descriptions of which variables are identified. The statistics used to evaluate these differences are essentially introduced in this caption; could you move that to the methods?

Response: We agree with the reviewer and can add labels in the captions, and also to describe the statistical methods in the methods section.

Response after revision: We have added a-c) captions as suggested by the reviewer.

Figure 3 – I'm surprised the UCD configurations melt so much earlier when they don't appear to be warmer. Is it possible that the spatial averages here obscure spatial differences that would explain why the UCD simulations melt earlier? Figure S-4 kind of gets at this, but I think it needs more interpretation for the reader.

Response: We agree with the reviewer that the spatial average visualization does not show the importance of locally specific spatial differences and, therefore, is unable to explain the physical reasoning of earlier snowmelt. We can create another supplement figure of the locally specific spatial differences in the UCD configuration simulation, and add a brief discussion of the physical reasons that may have given rise to an earlier snowmelt.

Response after revision: To address the reviewers concerns, we now provide Figure S-4 which compares the difference in monthly precipitation and temperature against PRISM, and the monthly snow water equivalent in the UCD-ERA5 WRF simulation. We have also added additional analysis in Section 4.1 to help provide readers with physical intuition for the simulated early snowmelt.  The following sentences were added to the manuscript to discuss these important points brought up by the reviewer, "Comparing the monthly average between UCD-ERA5 (Figure S-4) and BSU-ERA5 (Figure S-5), the early snowmelt observed in the UCD scheme is likely a result of warmer temperatures at low-altitude region that melt the snow earlier in the water year. However, the high-altitude regions remain cold enough to maintain snowpack through early-mid summer."

Line 319 – run-on sentence.

Response: We can modify this sentence for clarity.

Response after revision: This sentence has now been broken up and revised as follows, "Although the two-meter surface air temperature bias is evident, it doesn't vary significantly across either subgrid-scale physics scheme or meteorological forcing. Therefore, subsequent exploration in this study will be focused on precipitation."

Figure 4 – Nice figure. Could you again add an introduction to these statistics in the methods so we know how you're evaluating variance earlier? Why do c and d have only two points marked on the x-axis?

Response: Thank you for the kudos about this figure! We look forward to revising the manuscript accordingly. The x-extents of a-d are identical; we can add the additional tick-marks to avoid confusion.

Response after revision: The specific methods used to generate the statistics mentioned by the reviewer have now been added to the Figure 4 caption, "The standard deviations are the total

annual precipitation in each ensemble simulation, using either different subgrid-scale physics schemes or large-scale meterological forcings." Further, additional tickmarks have now been added to Figure 4c and 4d.

Line 335 – Were there any quantitative statistics provided to determine that BSU-CFSR2 agreed best with PRISM?

Response: We look forward to adding to the revised manuscript the quantitative statistic of RMSE (Root Mean Squared Error), as suggested earlier by the reviewer, for precipitation, temperature and SWE across all experiments to quantify how the BSU-CFSR2 configuration compared to the other configurations in terms of its agreement with PRISM.

Response after revision: The quantitative measurements requested by the reviewer have now been added to Table 3, with a new paragraph of text describing them as follows, "Quantitative statistics of the aggregated domain-average precipitation and temperature simulations for the WRF simulation across subgrid-scale physical schemes and large-scale meteorological forcings are presented in Table 3. Clearly, BSU subgrid-scale physical schemes outperform the UCD and NCAR schemes in both simulations of precipitation and temperature. On the other hand, the differences in precipitation and two-meter surface air temperatures across the four meteorological forcings are not statistically significant, and their standard deviations are much smaller than the differences in simulations across subgrid-scale physical schemes. While there are many metrics of model skill when selecting a meteorological forcing to simulate the hydrological processes in the ERW, we choose BSU-CFSR for the topographic radiation study in the next subsection due to its better match with PRISM, using our skill measures, in simulating both precipitation and two-meter surface air temperature ."

Figure 6 – Some panels appear not to use their full color scale (e.g., Temperature). Is that due to outlier pixels? There's a lot of wasted white-space in these maps – why not use the full plotting area for each map?

Response: We can adjust the extent of the plotting area according to this comment.

Response after revision: As suggested by the reviewer, the extent of the plotting area has now been enlarged in the subplots.

Line 380 – This paragraph describes Figure 7, but the next paragraph also seems to introduce Figure 7 as though it's a new topic?

Response: This paragraph describes the variance of simulated streamflow across experiments, and the next paragraph introduces the comparison against in-situ streamflow observations. We look forward to revising the manuscript to add a better transition sentence to aid the reader in separating these paragraphs.

Response after revision: A section subheading has now been added to break up the two subsections. The text has been revised as follows, "With the evaluation of the aforementioned WRF configurations and forcings on precipitation, temperature, snowpack and radiation fluxes, their impacts on the integrated water budget was evaluated in the ParFlow-CLM."

Line 401 – "The objective of this study is not to replicate the observations…" In that case, I strongly recommend changing the final sentence in the abstract, because that implies you're identifying the best model configuration.

Response: We agree with the reviewer and can revise the abstract to not explicitly state that the objective of this paper is to identify the optimal model configurations but rather an exercise of sensitivity analysis where one configuration will perform the best.

Response after revision: The last sentence in the abstract has been changed in the following way, "This is the first presentation of the sensitivity analyses that provide support to help guide the scientific community to develop observational constraints on atmopshere-through-bedrock processes and their interactions."

Line 413 – Are the differences notable or minimal? I would say minimal. Maybe better to describe quantitatively – you could note the among-model variance vs the seasonal variance?

Response: The reviewer is correct that the differences are minimal. Since this is a snow-dominated watershed and streamflow is predominantly controlled by snowmelt, the seasonal variance is not comparable with the among-model variances. We can revise the manuscript to describe the intra-model configuration variance in different seasons.

Response after revision: This sentence has now been revised as follows, "Basin-average groundwater storage, shown in Figure 7c in area-normalized units, shows a strong annual signal for all WRF configurations with minimal differences across IPM configurations."

Line 417 – "are slightly larger…" The differences are twice as big for the subgrid-scale physics schemes but are small in both cases; I would suggest rephrasing to clarify.

Response: We agree with the reviewer and look forward to removing the word "slightly" in the revised manuscript to avoid any misunderstanding.

Response after revision: Removed "slightly" as suggested.

Line 420 – What is meant by "more muted-nature"? I think this sentence speculating about differences in groundwater signals across years would be better in the discussion.

Response: We can re-word this sentence and perhaps use the wording "less noisy" opposed to "more muted" to avoid any confusion. What's meant here is that streamflow signals are very reactive, noisy, and change quickly, whereas groundwater signals are the product of slower

processes via infiltration and vadose zone dynamics, often at longer timescales, which result in very different temporal signals as compared to streamflow.

Response after revision: This sentence has now been revised as follows, "streamflow signals are very reactive, noisy, and change quickly, whereas groundwater signals are the product of slower processes via infiltration and vadose zone dynamics, often at longer timescales, which result in very different temporal signals as compared to streamflow."

Figure 8 – Is this color gradient perceptually uniform? It appears not to be (e.g., see Figure 1b in Cramer et al., 2020). It would be helpful to see a perceptually uniform palette here if possible.

Response: We can replace with the perceptually uniform color bar for the plots, based on the suggestions in Cramer et al. (2020).

Response after revision: The figure color bar has now been replaced with a perceptually uniform gradient.

Line 448 – "with an eye towards how to represent…" Without calibration or serious validation efforts, I don't think this study tells us about how to represent these interactions in models. I do think it tells us about where the most important uncertainties are, though (in your next sentence).

Response: Here we mean that by evaluating the model uncertainties for simulating precipitation, temperature, and streamflow, we are able to identify the which process within the model has the most important uncertainties. We can revise this sentence accordingly.

Response after revision: We removed the following phrasing in the sentence, "with an eye towards how to represent", and revised the sentence to read as follows, "In this paper, we present a number of numerical experiment results that are informative for the scientific community to better understand atmosphere-through-bedrock process interactions, and the uncertainties of those interactions between climate and hydrological model experimental setup choices"

Line 454 – I don't remember a prior discussion of boundary conditions – is this referring to boundary conditions at the land surface driven by differences in the subgrid-scale physics schemes?

Response: Here we mean the large-scale forcing dataset used as the initial and boundary conditions in the WRF model. We can revise this sentence and explicitly mention that in the revised manuscript.

Response after revision: The sentence has now been revised to, "This result also shows that the large-scale meteorological forcing of the IPM simulation"

Line 456 – This would be more convincing if statistics on BSU-CFSR2 vs other models were presented. How does identifying this configuration allow researchers to prioritize process studies and observational constraints? What would these be, specifically?

Response: We agree with the reviewer and look forward to adding the quantitative statistics RMSE for the experiments against the PRISM observationally-based dataset.

Response after revision: As mentioned in an earlier comment to the reviewer, the quantitative statistics are now presented in Section 4.1 and Table 3 in the manuscript.

Figure S6 – Could you use a different color scheme that doesn't have a diverging gradient? I think the diverging gradient is most appropriate for your maps showing differences (e.g., value scales that center on zero).

Response: We agree with the reviewer and can revise the color scale.

Response after revision: The color scale in Figure S6 has now been updated using an non-diverging gradient.

Line 467 – "Latent heat is posited…" by whom? Are you? I think you could state with more confidence than "posit" that other energy balance components (including but not exclusively latent) mediate the influence of shortwave spatial variability on temperature spatial variability.

Response: We agree with the reviewer and can revise this sentence as "Latent heat buffers differences in the shortwave radiation contribution to the radiation budget."

Response after revision: We have revised this sentence as follows to be more confident in causality, "Latent heat buffers differences in the shortwave radiation contribution to the radiation budget."

Line 470 – You lost me here. This paragraph is ostensibly about how terrain shading algorithms affect radiation flux? How does this affect our ability to extrapolate findings from one mountainous watershed to another? The multiple "if" statements in here are also a little confusing – did the present study show these things or not?

Response:  This paragraph discusses the systemic cold bias in our current IPM configuration, and the limitations of one-way vs. two-way coupling between WRF and ParFlow-CLM.  We will revise this paragraph to be more clear and to the point.

Response after revision: The sentence has been revised, with more clarity in mind, as follows, "At the same time, the systemic cold bias and limitations of one-way feedback in this study is potentially indicative of challenges in extrapolating findings from one mountainous watershed to another."

References

Crameri, F., Shephard, G.E. & Heron, P.J. The misuse of colour in science communication. Nat Commun 11, 5444 (2020). https://doi.org/10.1038/s41467-020-19160-7

---

## Author Comment (AC5)

**Referee Comments  #2**

The integrated process model (IPM) which resolves the processes extending from the atmosphere through the bedrock is a hot topic in recent years. Using the IPM, researchers try to investigate the interactions between atmosphere and underground hydrological processes (e.g., lateral flows, groundwater dynamics), which used to be neglected by traditional meteorological modeling works. The ParFlow-CLM model is also a famous tool that couples the one-dimensional and sophisticated land surface model (CLM) with the three-dimensional groundwater model (ParFlow). Xu et al. Tested the sensitive of some hydrometeorological variables, which were simulated by the WRF model coupled with an integrated hydrologic model, to the choices of physical parameterizations, meteorological forcings that provide lateral boundary conditions, and terrain shading options. The author found that, physical parameterizations contributes to the largest spatial temporal variance in simulating the temperature, precipitation and other related hydrological variables. Although the topic is important and the introduction is well written, I still think the innovation is not strong and the manuscript needs major revision. My major concerns are below:

1. The author emphasize the necessity and importance of IPM in the introduction and also take the IPM as one of the innovation of this research. However, the simulation work is based on one-way coupling (use the WRF simulated meteorological forcings to drive ParFlow-CLM). Whether this one-way coupling can be called as IPM is confused, as there is no feedback between meteorology and the underground hydrology.

Response: We thank the reviewer for her/his comments noting the sophisticated tools used in this study, as well as the importance and high quality of the manuscript writing. The concern about the use of the phrase "integrated process modeling" is potentially one of semantics in this case, although we're happy to revise to avoid any potential confusion for the reviewer and/or other readers. There is a small body of literature that does use "integrated process model" terminology (e.g., Zhang et al, 2016 doi:https://doi.org/10.5194/hess-20-529-2016; Davison et al, 2017 doi:https://doi.org/10.1002/2017MS001052) that demonstrates the utility of coupling process models built to explore discipline-specific processes as a mechanism to advance interdisciplinary research.  There is also a literature comparing and contrasting one-way coupling vs two-way coupling for mountainous hydrology: e.g., Camera et al, 2020 doi:https://doi.org/10.5194/nhess-20-2791-2020 and Rudisill et al, 2022 doi:https://doi.org/10.1002/hyp.14578 where the latter paper finds that in snow-dominated watersheds such as the ERW, which found that the representation of uncertainties in the representation orographic precipitation is the single largest driver of hydrological uncertainty while the inclusion or exclusion of two-way coupling has little effect on atmosphere-through-bedrock state evolution. Nonetheless, to avoid confusion we would like to clarify that we are referring to a one-way coupled IPM (or WRF-Parflow-CLM) in the revised manuscript.

Response after revision: A few IPM references have now been added to the Introduction. In addition, we now explicitly define the meaning of IPM as the use of WRF via a one-way feedback to ParFlow-CLM in the Introduction (to acknowledge the reviewers point about the extensive literature using two-way feedbacks). The section in the Introduction now reads as follows,"

While discipline-specific process models, such as those used to explore and predict atmospheric or subsurface processes have advanced scientific understanding in a myriad of ways through sustained engagement with extensive user communities (Gutowski et al., 2020), Integrated Process Models (IPMs), in which these discipline-specific process models are coupled, are relatively novel and are still being vetted for various scientific applications in complex terrain. Zhang et al. (2016) and Davison et al. (2017) demonstrated the utility of coupling process models built to explore discipline-specific processes as a mechanism to advance interdisciplinary research. Furthermore, Camera et al (2020) discussed the one-way vs. two-way coupling of IPMs to understand process interactions in the mountainous hydrologic cycle. The capabilities and details of the IPM has been discussed in a series of findings…."

2.  Another issue is that the finding that "physical parameterization is much more important than lateral or initial conditions"has been revealed by numerous works in meteorological discipline. For example, Solman and Pessacg (2012) found that the largest spread among WRF ensemble simulation members is caused by different combinations of physical parameterizations. Pohl et al (2011) tested the uncertainties of WRF simulation caused by physical parameterizations, lateral forcings, domain geometry. And they also suggested that physical parameterizations have the largest influence on precipitation. So, from the perspective of meteorology, the current finding is not surprising. The author should review the previous works and rethink the added value of the current work.

Response: We thank the reviewer for raising this issue and will add the cited references in the revision where appropriate, especially given the concurrence of our findings with those references. However, our findings are not redundant with the published literature: those references either evaluated large-scale meteorological processes or did not focus on high-altitude complex terrain regions, which are central to our study. Additionally, most previous studies do not show how the range of reasonable IPM configurations (based on configurations that have been presented in the published literature) affects discharge, ET and subsurface hydrology. We would argue that these aspects to our work represent an additional set of novel contributions that will be of interest to the readers of HESS.

With this set of one-way atmosphere-through-bedrock process modeling results, we can and have uncovered how choices in atmospheric process model configurations impact the surface and subsurface hydrology.  Specifically,  we evaluate and quantify the sensitivity of discharge, ET, and subsurface hydrology to IPM configurations, and we also address how 3D topographic radiation schemes affect both spatial distribution and spatial average hydroclimate simulations. More importantly, this study aims to guide the plan of field observational activities in the future by (1) uncovering how uncertainties in the representation of atmospheric processes impact

surface and subsurface process modeling and (2) providing direction for those field observations with the greatest potential to constrain atmospheric processes. We will revise the paper to highlight the added value of the contributions of our work to the existing, relevant literature.

Response after revision: A paragraph was added to the Introduction to more clearly introduce the novel aspects that our paper provides to the IPM community, "We recognize that numerous works in meteorological disciplines have demonstrated that "physical parameterization is much more important than lateral or initial conditions" (e.g., Solman and Pessacg, 2012 and Pohl et al., 2011). However, our findings are not redundant with the published literature, as those references either evaluated large-scale meteorological processes or did not focus on high-altitude complex terrain regions, which are central to our study. Additionally, most IPM studies to date do not show how the range of reasonable IPM configurations (based on configurations that have been presented in the published literature) affects water management relevant processes such as discharge, ET and subsurface hydrology. With our set of one-way atmosphere-through-bedrock process modeling results, we now show how choices in atmospheric process model configurations impact the surface and subsurface hydrology. Specifically, we evaluate and quantify the sensitivity of discharge, ET, and subsurface hydrology to IPM configurations, and we also address how 3D topographic radiation schemes affect both the spatial distribution and spatial average aspects of the mountainous hydrologic budget"

3. Since the ERW is a heavily-instrumented catchment with a growing atmosphere-through-bedrock observation network (emphasized in abstract) and the "The goal of this work is to provide the mountain hydrology research community with a properly-configured IPM that can inform ongoing and future field campaigns and their process-modeling needs in the UCRB.", why don't you use the in-situ observations to evaluation the T2m and precipitation.

Response: We agree with the reviewer and look forward to adding the comparison against the precipitation, two-meter air temperature and ET in-situ observations in the paper. We agree adding those comparisons against in-situ observations will help quantify the model performance in terms of surface air temperature and precipitation and will emphasize the value of observational networks supporting model evaluation. The SAIL-based observations will be used in a future study to compare with IPM skill once the SAIL campaign is completed (2021-2023).

Response after revision: The statement referred to by the reviewer has been revised in the following way, "The ERW has become one of the most heavily-instrumented mountainous watersheds in the world, which makes it an ideal focus for this research given the potentially large number of observational constraints available for the IPM efforts presented here. For example, The SAIL-based observations (Feldman et al., 2022) will be used in a future study to compare with IPM skill once the SAIL campaign is completed (2021-2023). Although a wide range of precipitation, temperature and hydrological data have been collected, it is still challenging to use these to characterize atmospheric, surface and subsurface processes and their interactions at their relevant scales."

4. Moreover, I am really confused about the use of Parflow-CLM here. Is it used to only provide streamflow and groundwater storage? The simulated snow and ET a in re provided by CLM-Parflow or the default land surface model in WRF? Actually, the Parflow-CLM is often used to investigate the potential influence of three-dimensional groundwater on the responses of terrestrial hydrological processes to meteorological forcings (e.g., numerous high impact works performed by Maxwell and Condon). However, here, I did not see what will be different if we used the traditional one-dimensional land surface model to investigate the same issue. I suggested the author to compare the difference when using the results from default WRF land surface simulation and that from Parflow-CLM (such as ET, total water in the soil column). This may help enhance the innovation of current work.

Response: Yes, the primary reason for using ParFlow-CLM is to allow for the quantification of streamflow and groundwater storage. As the reviewer is probably aware, standalone WRF does not simulate these processes, although branches of the code (WRF-Hydro) do provide some insight into at least streamflow, although with a simplified and prescribed stream network. Groundwater in WRF-Hydro is highly simplified (shallow soil layers and a bucket model) while ParFlow simulates the full continuum of variably saturated flow in three dimensions.

We are interested here in developing, testing, and analyzing simulations with realistic representations of atmospheric processes and to explore how they interface with representations of surface and subsurface processes that are as realistic as possible. For this study, we would like to avoid complicating the analysis with additional model structural errors where we can, so that in the future, we can ultimately relax towards the more simplified representations of surface/subsurface processes in models such as WRF-Hydro.

The innovation of this work is to better understand how these choices in atmospheric parameterizations, forcing, and other configurations/options impact the greater hydrologic cycle. We respectfully disagree that the one-dimensional land surface model alone can yield this information given the importance of lateral groundwater flow contributions to streamflow, especially in complex mountainous terrain like this watershed, which is why we performed the additional modeling steps with ParFlow-CLM.

Response after revision: We extended the introduction of one-dimensional land surface model in WRF/WRF-Hydro and 3D ParFlow-CLM model in the introduction as, "However, neither standalone WRF nor WRF-Hydro do not explicitly simulate streamflow and three-dimensional groundwater proceses. Groundwater in WRF-Hydro is highly simplified (shallow soil layers and a bucket model) while ParFlow simulates the full continuum of variably saturated flow in three dimensions. Therefore, one-dimensional land surface model alone cannot be used to better understand the configuration impacts on the greater hydrologic cycle, given the importance of lateral groundwater flow contributions to streamflow, especially in complex mountainous terrain."

5. The experimental design needs more detailed information. I suggest the author to provide more introduction to the experimental design. For example, why do you only use the CFSR2 and ERA5 in the UCD and NCAR simulation? Why does the no3DRad_inner radiation scheme is only used in BSU_CFSR2 and BSU_ERA5?

Response: We designed the experiments with the intent to evaluate different subgrid-scale physical scheme configurations, atmospheric boundary forcings, and topographic specific subgrid-scale parameterizations. However, we recognized that this study would be computationally constrained given our prioritization of the use of sub-km horizontal resolution IPM simulations, hence why the model configuration matrix was not completely sampled. We first chose to run a series of experiments with the three most prominent WRF subgrid-scale physical scheme configurations in the literature. We learned that the BSU configuration is the optimal physical scheme relative to the others chosen. As such, and due to computational restraints, we then chose this configuration to interrogate the topographic specific subgrid-scale parameterizations (e.g., noD3Rad_inner). We hope to fill some of the gaps in the simulation matrix in future work, particularly when we compare these simulations with the SAIL observational campaign (once completed in 2023). We plan on including more descriptive language on the experimental design in the revised manuscript.

Response after revision: We extended the analysis of BSU-CFSRS2 (among other experiments) in Section 4.1 and added the following paragraph, "Quantitative statistics of the aggregated domain-average precipitation and temperature simulation results for the WRF simulation across subgrid-scale physical schemes and meteorological forcings are presented in Table 3. Clearly, BSU subgrid-scale physical schemes outperforms the UCD and NCAR schemes in both precipitation and temperature simulations. On the other hand, the differences of precipitation and temperatures across the four meteorological forcings are not statistically significant, and their standard deviations are much smaller than the differences in simulations across subgrid-scale physical schemes. While there are many metrics of skill for selecting a meteorological forcing for simulating the hydrological processes in the ERW, we choose BSU-CFSR for the topographic radiation study in the next subsection due to its overall performance in precipitation and temperature simulation as compared with PRISM."

Additionally, at the beginning of Section 4.2, we revised the text to justify that BSU-CFSR is only used for the 3D topographic radiation effect analysis, "Based on the assessment of simulated precipitation and two-meter surface air temperature compared with PRISM, the BSU-CFSR2 configuration is selected as a baseline to further explore the influence of topographic radiation scheme effects. The difference caused by turning on and off the 3D topographic radiation effects is similar in other WRF configurations; therefore, only the BSU-CFSR is presented."

6. I suggest to show the topograpnhy of the inner domain in Figure 1 which will be helpful to better understand the influence of 3D-radiative scheme. Currently, I am confused why the valley gets more radiation after considering the topographic shading and slope effect.

Response: To address this comment,  we can upate Figure 1 and present the topography of the inner domain in more detail and providing more descriptive text for Figure 1. This will help show how and why the east side of the valley receives more radiation because it's west facing, and the western side of the valley gets less radiation in the early morning due to solar geometry. Similar conclusions have been observed in Arthur et al. 2018.

Response after revision: A satellite image with topographic and watershed delineations has now been added as Figure 1c.

7. Moreover, the author should proofread the manuscript. For example:
● The Figure S-4 in L 299 should be Figure S1?
● "Figure S-3 and Figure S-4"should be "Figure S-3"?
● There is no description or analysis on the Figure 8c-8d.
● I also noticed some grids are masked out in Fig S-5 and Fig S-6, but no interpretation is given.

Response: We thank the reviewer for catching these typos and will revise all of them accordingly in our revision.

Response after revision: All reviewer suggestions have been incorporated into the manuscript.

Reference:
Solman and Pessacg. (2012). Evaluating uncertainties in regional climate simulations over South America at the seasonal scale.
Pohl et al. (2011). Testing WRF capability in simulating the atmospheric water cycle over Equatorial East Africa

---

## Author Comment (AC6)

**Referee Comments  #3**

This work presents a model study in a headwaters catchment in the Upper Colorado Watershed. In general the work is interesting and well-written but the presentation is somewhat confusing. There are some major points I think the authors need to address before the work's suitability for publication can be assessed.  They are detailed below.

We thank the reviewer for his or her comments on this work being interesting and well-written. We have addressed all points the reviewer requests clarification on below.

General Comments:
-The terminology of so-called IMP's used is confusing along with the reference to these as coupled models, which I would argue they are not.  It's very confusing to discuss this work in the framework of different models as opposed to just forcing used to drive the hydrologic model. The language around the different products used is very confusing and makes much of the discussion hard to follow.  Some of the meterological forcing datasets appear to be used to drive the models direclty, but in the introduction it appears that only WRF simulations are used to drive models.  A completely re-write of this entire section is needed to make this clear.  What did the authors do with the hydrologic outputs from the WRF simulations?  Why are the Noah and Noah-MP models used interchangeably but the results are not compared to ParFlow-CLM?

Response: We thank the reviewer for her/his comments noting the sophisticated tools used in this study, as well as the importance and high quality of the manuscript writing. The concern about the use of the phrase "integrated process modeling" is potentially one of semantics in this case, although we're happy to revise to avoid any potential confusion for the reviewer and/or other readers. There is a small body of literature that does use "integrated process model" terminology (e.g., Zhang et al, 2016 doi:https://doi.org/10.5194/hess-20-529-2016; Davison et al, 2017 doi:https://doi.org/10.1002/2017MS001052) that demonstrates the utility of coupling process models built to explore discipline-specific processes as a mechanism to advance interdisciplinary research.  There is also a literature comparing and contrasting one-way coupling vs two-way coupling for mountainous hydrology: e.g., Camera et al, 2020 doi:https://doi.org/10.5194/nhess-20-2791-2020 and Rudisill et al, 2022 doi:https://doi.org/10.1002/hyp.14578 where the latter paper finds that in snow-dominated watersheds such as the ERW, which found that the representation of uncertainties in the representation orographic precipitation is the single largest driver of hydrological uncertainty while the inclusion or exclusion of two-way coupling has little effect on atmosphere-through-bedrock state evolution.

We agree that the use of the word "coupled" should be used with caution, as some of these models are formally coupled within their source-code infrastructure, and others are "step-wise coupled". To avoid any potential confusion, we have modified the manuscript to only use the word "coupled" when there is a formal two-way and self-consistent pairing of codes (for

example, ParFlow and CLM). In contrast, the associated meteorological variables used to drive ParFlow-CLM are now referred to as "forcing" only, as this is a one-way interaction between the codes/outputs.  Nonetheless, to avoid confusion we would like to clarify that we are referring to a one-way coupled IPM (or WRF-Parflow-CLM) in the revised manuscript.

The name of the meteorological products are the forcing datasets for the WRF runs, not the forcing data for ParFlow-CLM. Precipitation, temperature and other hydroclimate simulations from WRF are used as the atmospheric drivers for the ParFlow-CLM simulations. We will revise the manuscript to explicitly clarify that the meteorological forcing datasets are used as the initial and boundary conditions for the WRF model, and the WRF model outputs are then used to force ParFlow-CLM.

The Noah and Noah-MP hydrologic output from the WRF simulations included ET, surface and subsurface runoff. We agree with the referee (and referee #2 who identified this too) and will compare the Noah and Noah-MP simulated ET against ParFlow-CLM.  The primary reason for using ParFlow-CLM is to allow for the quantification of streamflow and groundwater storage. As the reviewer is probably aware, standalone WRF does not simulate these processes, although branches of the code (WRF-Hydro) do provide some insight into at least streamflow, although with a simplified and prescribed stream network. Groundwater in WRF-Hydro is highly simplified (shallow soil layers and a bucket model) opposed to ParFlow, which simulates the full continuum of variably saturated flow in three dimensions. Thus, while we indeed have generated multiple datasets with the different LSMs, this is not our primary objective for intercoparison and we believe showing more of these results would be distracting. We focus on the set of LSM outputs from WRF for simplicity (and to avoid confusion) and show the groundwater and streamflow outputs from ParFlow-CLM.

Response after revision: A few IPM references have now been added to the Introduction. In addition, we now explicitly define the meaning of IPM as the use of WRF via a one-way feedback to ParFlow-CLM in the Introduction (to acknowledge the reviewers point about the extensive literature using two-way feedbacks). The section in the Introduction now reads as follows,"
While discipline-specific process models, such as those used to explore and predict atmospheric or subsurface processes have advanced scientific understanding in a myriad of ways through sustained engagement with extensive user communities (Gutowski et al., 2020), Integrated Process Models (IPMs), in which these discipline-specific process models are coupled, are relatively novel and are still being vetted for various scientific applications in complex terrain. Zhang et al. (2016) and Davison et al. (2017) demonstrated the utility of coupling process models built to explore discipline-specific processes as a mechanism to advance interdisciplinary research. Furthermore, Camera et al (2020) discussed the one-way vs. two-way coupling of IPMs to understand process interactions in the mountainous hydrologic cycle. The capabilities and details of the IPM has been discussed in a series of findings…."

Except for groundwater which isn't discussed very much in the manuscript, all the same results should be in the WRF simulations.  Why didn't the authors just run the WRF-ParFlow model or

even mention it's existence?  They talk about everything in a coupled sense but the models are in no way formally coupled (unless something is happening that is not discussed in the manuscript); the output files from WRF simulations are saved and somehow reformatted (this is not clear) and used to drive ParFlow-CLM.  They could drive any hydrologic model and it wouldn't be considered a coupled platform, likewise the standard forcing products the authors might choose to drive the simulations off the shelf are also generated with atmospheric models, yet I would never think of this as an IMP.  I suggest the authours are much more transparent about this aspect and remove the terminology from a revision.  They should also provide some clear language about what is actual being done here, is this a comparison between forcing generated with WRF v other approaches?  Why didn't the authors just run forced by PRISM?

Response: As is hopefully now more clear based on the last answer, we use WRF outputs to force ParFlow-CLM as a mechanism to simulate the integrated hydrologic response of the watershed. We do this because WRF is not capable of simulating two key processes of interest: not just groundwater storage changes but also streamflow. We did not run the fully coupled WRF-ParFlow model platform because these simulations are extremely computationally expensive. We agree that it would be interesting to perform these 2-way coupled simulations in the future, but believe that an important first step is to determine what the hydrologic response is without the two-way interaction (i.e. WRF-forced ParFlow-CLM as we've done here). Fully coupled 2-way simulations are by no way the "standard" in hydrologic modeling, and the amount of fidelity we've included in this study stands to dwarf that of other approaches. That being said, we acknowledge that the existence of the code could be pointed to as an interesting next step. Two-way integrated meteorological and surface/subsurface hydrology simulations in complex terrain are especially limited, and a new frontier in the field.

Furthermore, we do appreciate the sentiments of the reviewer and will ensure that we will more clearly outline various methodological approaches taken in our model simulations and analyses in our revision.  For example, the WRF simulations were regridded using bilinear interpolation and used as the forcing dataset to run the ParFlow-CLM model.  Therefore, the simulations are, as the reviewer points out, a one-way coupled IPM.  We will ensure that we mention that WRF-Parflow is also available as a two-way coupled IPM too (and cite the relevant literature).

Finally, we agree with the reviewer and can run the ParFlow-CLM simulation forced by PRISM. We plan to regrid the 800-meter resolution PRISM daily precipitation and temperature variables and temporally interpolate them using an adjustment to the hourly distributions provided by the WRF hourly output. Other than precipitation and temperature (only variables made available from PRISM), we will use the WRF hourly output to force the ParFlow-CLM simulation.

Response after revision: In the Discussion section, we have tried to be more transparent about what is meant when we use the terms ParFlow-CLM, WRF-hydro and WRF-PF: "On the other hand, ParFlow-CLM is essential in our experiment for quantifying hydrological responses, including streamflow and groundwater storage. Although WRF-Hydro provides some insights into streamflow, it still uses a simplified and prescribed stream network. Groundwater storage in WRF-Hydro is also highly simplified through its use of a bucket model, while ParFlow-CLM

simulates the full 3D continuum of variably saturated soils. We also recognize that the WRF-ParFlow model can simulate two-way coupling between the atmospheric and hydrological processes in the surface and subsurface domain, though it is computationally expensive and requires signficant efforts to set up.  Importantly, in a similar fashion as the heirarchy of climate models approach oft used in the climate community (Jeevanjee et al., 2017), we would also like to assess one-way coupling performance of our IPM prior to assessing two-way coupling IPM performance."

We also performed new ParFlow-CLM experiments forced by PRISM and included these results in Figure 7, and the text has been revised as follows, "We have evaluated the aforementioned WRF subgrid-scale physical schemes and large-scale meteorological forcings in representing precipitation, temperature, snowpack and radiation fluxes, and their impacts on the integrated water budget within ParFlow-CLM. We also evaluated the simulated discharge from ParFlow-CLM forced by PRISM as a comparison with WRF forcings."

-Coupled v uncoupled processes and feedbacks.  There has been a lot of work to understand the role of feedbacks between two-way coupled hydrologic models and atmospheric models. Examples include WRF-Hydro-WRF (e.g. Arnault 2016), COSMO-CLM-Parflow (e.g. Keune 2016, 2019), WRF-ParFlow (e.g. Maxwell 2012, Forrester 2020), feedbacks over complex terrain (Ban 2014),  and other more conceptual approaches (e.g. Miguez-Macho 2007).  This is not an exhustive list, but demonstrates that much work has been done to study these feedbacks,  Some of these studies are in complex terrain and even suggest that the approach used by the authors may not be valid at high resolution without lateral flow.  These studies all systematically compare different types of model physics (e.g. free drainage, the standalone atmospheric model, fully coupled system) and use varying metrics to diagnose coupling strength and changes in the atmosphere.  I suggest the authors read these prior studies carefully and develop a new section that summarizes (rather than ignores the existence of) this body of work and uses this to put the current study in context.  This will help frame the current work and help it look much less like a patchwork of runs that are loosely tied together.  This will also help clarify my point above, to help the reader follow what is being done and what runs are conducted in the current work.

Response: We agree with reviewer that there are multiple two-way coupled IPM configurations and will ensure that our revision better contextualizes our one-way coupled IPM configuration within the broader two-way coupled literature. We can develop a new paragraph in the introduction to demonstrate the details of those models in the paper.

Response after revision: A new paragraph has been developed in the Discussion section to better contextualize one-way and two-way feedback mechanisms in the IPM community as follows, "Another methodological constraint is that our WRF and Parflow-CLM experiments were only one-way instead of two-way feedbacks, which ignores potentially important feedbacks from the subsurface hydrology to the atmosphere via ET and the radiation budget. For example, Givati et al. (2016) reported that simulated precipitation was improved with two-way coupling in WRF-Hydro compared to WRF-only and Forrester et al. (2018) showed that boundary layer

dynamics were impacted in IPM simulations in regions where shallow water tables exist. On the other hand, ParFlow-CLM is essential in our experiment for quantifying hydrological responses, including streamflow and groundwater storage. Although other fully-coupled integrated hydrology model (e.g., WRF-Hydro) provides some insights into streamflow, it still uses a simplified and prescribed stream network. Groundwater storage in WRF-Hydro is also highly simplified by using a bucket model while ParFlow-CLM simulates the full continuum of variable saturation in three dimensions. "

-Variability in point processes compared to integrated or averaged measures. I mention this as a specific instance below, but it is also a general point, there are instances where the authors present differences locally (at a point) that do not persist synoptically. Do the different forcing products or microphysics (I think this is the point the authors make) make some difference locally for e.g. precip, radiation, but does some averaged quantity remain unaffected. It appears this is the case for much of the analysis. That is, topographic shading makes a difference locally in LH flux but the domain averaged LH flux remains unchanged between cases. The authors draw one conclusion (local differences) without acknowledging the other (same net energy flux over the domain).

Response: We agree with the reviewer and look forward to adding more discussion of locally specific versus watershed average differences in our IPM results. For example, the importance of topographic shading on the spatial distribution of radiation flux. The variances across different forcing products and subgrid-scale physical schemes are shown in our paper through both spatial distributions and average quantities. However, the 3D topographic radiation schemes only redistribute the energy flux thus affect LH flux spatial distribution, but do not show significant difference in the watershed average quantities. In the revision, we will more clearly highlight these points.

Response after revision: The text has now been revised to, "Topographic shading makes a difference locally in LH flux, by redistributing the energy flux and thus affecting LH flux spatial distribution. Nevertheless, the domain averaged LH flux remains unchanged between cases."

-Atmospheric uncertainty. There has been much work on differences in model physics in a model such as WRF that allows different physical parameterizations to be "swapped out" easily in simulations by changing the namelist. This is an important aspect of uncertainty, but it is almost always put in the context of one of the major forms of uncertainty in the atmosphere, propogation of intial conditions. One should always determine that such a physics change is robust using (e.g.) time-shifted uncertainty in an ensemble type approach (e.g. Walser 2004). Often upon inclusion of uncertainty in the intial model state (in the atmosphere) the differences in physical paramterization no longer dominate.

Response: We acknowledge the reviewer's well-taken point about the value of using a traditional experimental design to isolate and compare model structural uncertainty to model internal variability uncertainty to improve modeling skill and predictive power. We concur that the specific use of initial condition perturbation ensemble members enable the quantification of

model internal variability while varying subgrid-scale physics parameterizations tests model structural uncertainty. The use of different meteorological forcings do comprise different initial conditions, though this does differ from explicit initial condition perturbation analysis. We will add a few sentences in the Methods section pointing to the reviewer's important points, explicitly cite Walser and Schaer (2004) and be more explicit about how our experimental design differed from the more traditional approach. As part of what we will add to the Methods section we would like to acknowledge that although our experimental design didn't assess internal variability through the more traditionally employed initial condition perturbation approach (i.e., alter the initial conditions provided to the atmospheric model by a rounding error) we did test the WRF-ParFlowCLM configuration with a range of realistic large-scale initial boundary conditions derived from the use of several atmospheric reanalyses. This approach, while coarse, does separately assess the influence of initial and boundary conditions (e.g., integrated vapor transport) ranges from the role of atmospheric physics representations in WRF-ParFlowCLM to isolate slight differences in large-scale boundary conditions.

We believe that a major finding of our work that hasn't been explored in the literature extensively over the East River watershed is that these slight differences in large-scale boundary conditions (analogous to perturbing initial conditions by rounding errors) do not markedly change water year precipitation totals unlike changes in subgrid-scale physical parameterization choice.

Response after revision: The discussion of atmospheric uncertainty and the initial condition has now been added to, "We recognize that the output from WRF simulations may be dependent on initial conditions, which are inherently difficult to constrain (e.g., Walser and Schär, 2004), but the experimental configuration described here seeks to be insulated from that dependency by running WRF simulations with initial conditions derived from different meteorological forcings."

-Can the authors compare meterological forcings at the site? A heavily-instrumented catchment (abstract line 16-) should have observations of meterological variables and snow outside of the SNOTEL (which I don't think are used for comparison and should contain precipitation and temperature), even preciptation and temperature at gage locations would be very instrumental. It appears that the authors treat the PRISM product like observationsm, which is an unfortunate and hopefully accidental. The PRISM product is a model, even if statistical, that takes into account observations in a region. One would assume that then PRISM is ingesting precip from the SNOTEL sites in the domain but this isn't stated (are there even any obsevations that PRISM is using and is it thus totally unconstrained?).

Response: The large-scale meteorological forcings used in the WRF simulations have coarser grid resolutions than the WRF inner domain, so comparing IPM simulation results with meteorological forcing at the sites do not add scientific significance in model evaluation.

The reviewer is right about the PRISM product: it uses SNOTEL datasets and was generated using statistical methods. PRISM dataset has been widely used to evaluate climate models in the complex-terrain region, with many thoughtful uncertainty analysis (e.g., Lunquist et al., 2020). We used interpolated precipitation and temperature product PRISM to evaluate the

model performance over the whole domain. In the supplementary material, we presented the comparison of precipitation and temperature against snotel stations. In the paper revision, we can compare with in-situ observation data, including precipitation and temperature measurements taken at sensors installed in the East River watershed. We did not provide these in the initial manuscript to avoid distracting from the main message of the paper, but see the relevance and could add these to the supplemental in the revision.

Response after revision: The precipitation, temperature and snow water equivalent comparison among WRF simulations vs. measurements at two SNOTEL stations are now presented in Figure S3.

-The authors should compare to ET observations in Ryken et al 2022 to results of current work (both WRF and ParFlow-CLM). Additionally, it apears that the Ryken et al 2022 paper has meterological observations of precip, temperature and radation that might be useful to partly address my comment above.

Response: We agree with the reviewer and can add the comparison against the ET measurement at the eddy-covariance tower at the East River, but again do not want to distract from the main purpose of this study by making this an ET intercomparison so this can potentially be added to the appendix.

Response after revision: Per the reviewer's comment, we performed a comparison of WRF and ParFlow-CLM against the eddy-covariance tower in the East River. As shown below, the WRF simulations generally agree with the observation at the tower (black line), with the exception of the BSU-ERA5 simulation:

[Figure]

However, a comparison of the ParFlow-CLM ET simulations revealed lower annual ET fluxes, with annual ET on the order of 100-200 mm. This difference is largely due to a number of factors in both the ParFlow-CLM and WRF simulations. First, land cover type: WRF parameterization of land cover in this region is coarser than that of ParFlow, and represents a coniferous forest over this region, whereas ParFlow is based on NLCD land cover resolved to 30 m, which was then upscaled to 100 (m) to match the resolution of the ParFlow-CLM model. The land cover type of the cell containing the flux tower in ParFlow-CLM is show centered in the figure below with a white star, and is represented in the model as "open shrubland." Neighboring needleleaf forests cells in ParFlow-CLM yield similar annual ET totals to that of WRF and the flux tower, which shows consistency in the predictability of the WRF simulations.

[Figure]

A second consideration is the flux tower fetch (see Figure 5 of Ryken et al., 2022, pasted below), which shows segments of the tower footprint over the meandering reaches of the East River that will bias the observation of ET at the tower high. Personal communication with David Gochis (who installed the tower with Ryken and Maxwell) confirmed this presumption, stating that if a land model parameterized the land cover similar to that of the surrounding land cover type (grasses and shrubs), it would likely calculate a lower flux compared to the observed ET measurement of the tower. Thus, it's sensical that ParFlow-CLM estimates are low when only assuming the cell-based comparison of ET.

[Figure]

Thus, the use of the tower to benchmark the model performance at this scale is not appropriate, and we have chosen not to include the discussion in the paper.

Specific comments

line 65: is PF-CLM being cited using Maxwell et al 2015 (cited on line 617)?  That paper references a simulation over large scale that as I read it is forced externally and does not use or describe the CLM model.

Response: Yes, while this is the same ParFlow, we agree a better citation can be used, the standard are: (Ashby and Falgout, 1996; Jones and Woodward, 2001; Maxwell, 2013). We will cite more appropriately in the revised version, including explicit mention of previous version of the model in the East River (Maina et al., 2022; Foster and Maxwell, 2018; Pribulick et al., 2016).

Citations:

Ashby, S.F., Falgout, R.D., 1996. A parallel multigrid preconditioned conjugate gradient algorithm for groundwater flow simulations. Nucl. Sci. Eng. 124 (1), 145–159.

Kuffour, B. N., Engdahl, N. B., Woodward, C. S., Condon, L. E., Kollet, S., & Maxwell, R. M. (2020). Simulating coupled surface–subsurface flows with ParFlow v3. 5.0: capabilities, applications, and ongoing development of an open-source, massively parallel, integrated hydrologic model. Geoscientific Model Development, 13(3), 1373-1397.

M. Foster, L., & M. Maxwell, R. (2019). Sensitivity analysis of hydraulic conductivity and Manning's n parameters lead to new method to scale effective hydraulic conductivity across model resolutions. Hydrological Processes, 33(3), 332-349.

Jones, J.E., Woodward, C.S., 2001. Newton–Krylov-multigrid solvers for large-scale, highly heterogeneous, variably saturated flow problems. Adv. Water Resour. 24 (7), 763–774.

Maina, F. Z., Wainwright, H. M., Dennedy-Frank, P. J., and Siirila-Woodburn, E. R.: On the similarity of hillslope hydrologic function: a clustering approach based on groundwater changes, Hydrol. Earth Syst. Sci., 26, 3805–3823, https://doi.org/10.5194/hess-26-3805-2022, 2022.

Maxwell, R.M., 2013. A terrain-following grid transform and preconditioner for parallel, large-scale, integrated hydrologic modeling. Adv. Water Resour. 53, 109–117.

Pribulick, C. E., Foster, L. M., Bearup, L. A., Navarre-Sitchler, A. K., Williams, K. H., Carroll, R. W., & Maxwell, R. M. (2016). Contrasting the hydrologic response due to land cover and climate change in a mountain headwaters system. Ecohydrology, 9(8), 1431-1438.

Response after revision: The introduction of ParFlow-CLM is now done in the following way, "result in differences in surface and subsurface hydrologic metrics when used to force the integrated hydrologic model (ParFlow-CLM; Ashby and Falgout, 1996; Jones and Woodward, 2001; Maxwell, 2013, Maxwell et al., 2015), which has been widely applied in the UCRB (Maina et al., 2022; Foster and Maxwell, 2018; Pribulick et al., 2016). We expand upon those various sensitivity analyses in this study, including the influences of large-scale meteorological forcing and subgrid-scale physics scheme choice on the surface-through-subsurface response of the integrated hydrologic model."

Line 240+ This section describes the PF-CLM model in general but I could not find specifics for the model domain used in this study? What is the resolution or model configuration for the PF-CLM domain? How deep is the subsurface? What is the lateral resolution? How was this matched to the forcing datasets or the WRF outputs? Was there a balance of water and fluxes between the grids? How were model parameters determined? Are there references to prior work on this model? Calibration? If not the authors might include a description of these aspects in the current manuscript and as supplemental material.

Response: The ParFlow-CLM subsurface domain is 30-meter deep, 100-meter horizontal resolution. The WRF output are re-gridded using bilinear interpolation to match the ParFlow-CLM grid cells. The model parameters are based on a variety of geological and soil parameters, and calibarated using streamflow measurements. More details can be found in some previous papers (e.g., Foster et al., 2019, Pribulick et al., 2016, see the answers to the previous question). We will add more detailed descriptions of the ParFlow-CLM model in the manuscript

Response after revision: The description of ParFlow-CLM has now been revised as follows: "The ParFlow-CLM subsurface domain is 30-meter deep at 100-meter horizontal resolution. The WRF outputs are re-gridded using bilinear interpolation to match the ParFlow-CLM grid cells. The model parameters are based on a variety of geological and soil parameters, and calibrated using streamflow measurements. More details can be found in Foster et al., (2019) and Pribulick et al., (2016)."

Lines 275-281. The UCD datasets appear to have the most precip but the ear

Figure 3 caption (~line 305): a, b, c are used to identify plots in the figure but are not used in the caption.  Also, it does not appear that 3c is described in the caption.

Response: We agree with the reviewer and can revise the figures accordingly.

Response after revision: Revised as suggested.

Figures 5, 6 and associated discussion.  An interesting point that might be made here is that while local spatial differences are apparent in Figure 6, the domain averages (even for SWE) are the same between shaded and non-shaded formulations.  This suggests that while it may be striking visually to include shading, the upscaled water balance for the catchment isn't sensitive.

Response:  We agree with the reviewer that turning on and off the 3D shaded radiation scheme does not significantly affect the domain average. The reviewer is correct that the 3D shading radiation scheme does not affect the upscaling water balance, but rather redistributes the spatial distribution of radiation fluxes thus SWE and energy spatial pattern. The east side of the valley gets more radiation because it's facing west, and the western side of the valley gets less radiation as it gets less radiation in the early morning. Similar conclusions have been observed in Arthur et al. 2018.

Response after revision: The text has now been revised as follows, "The 3D  radiation shading scheme does not significantly affect the total water balance, but rather the spatial distribution of radiation fluxes. Thus, despite having minimal impacts on water impacting on the water balance, the scheme does have important localized impacts on SWE and surface energy budget spatial patterns" and "In summary,  the simulations show that, while local spatial differences in surface radiation with and without realistic topography are apparent in Figure 6 , the domain averages (even for SWE) are the same between shaded and non-shaded formulations.  This suggests that while it may be striking visually to include shading, the impact of topographic shading on upscaled water balance for a domain like the ERW is negligible."

lines 383- I'm not sure I agree with these conclusions.  While the cumulative variability in outflow resulting from the different forcing products creates different cumulative outflows, Figure 7a indicates that there is no difference in timing across all the forcing datasets.  My suspicion is that the differences in outflow are due to total water quantity (Figure 5a suggests this as well) and are simply a precip bias artifact in the different WRF runs.

Response: We agree with the reviewer that Figure 7a indicates the difference in timing in all the IPM across all forcing datasets. The difference of precipitation in WRF simulations leads to the differences in streamflow simulation in ParFlow-CLM. We can revise this sentence to clarify and avoid misunderstanding.

Response after revision: We revised the sentence as follows, "Discharge at the watershed outlet (see exact location on Figure 1) shows a different timing across the various WRF subgrid-scale physics scheme configurations and large-scale meteorological forcings that leads to a temporal

shift in simulated streamflow, where the daily averaged time series (left) shows only minor differences through time."

line 443: "Here, we ... coupling WRF and ParFlow" rephrase, this sentence isn't correct, the models were not coupled

Response: We can reword this as 'one-way coupling IPM' or "WRF-ParFlow-CLM" to avoid misunderstanding.

Response after revision: We have revised this sentence in the text as follows, "here, we used an IPM with one-way feedbacks from WRF to ParFlow-CLM".

line 476+: This text appears to acknowledge the lack of coupling in the current work (as an aside, what is "one-way coupled" this is not actually coupled at all, as it appears the results from WRF were simply used as forcing for the PF-CLM model. This isn't bad, but as mentioned above should be discussed up front. The arguments here regarding computational expense as an excuse for not running coupled simualtions are incorrect, prior studies have shown with the e.g. WRF-PF model that ParFlow is approximately 1% of the total computational time compared to WRF which is 99% of the computational time. Thus if the authors ran WRF for this domain, the additional expense to run with WRF-PF is a negligible increase in cost. Also the authors might want to correclty identify that the Forrester et al study (line 483) was run with WRF-PF and the authors might want to read and cite Forrester 2020 which discusses limitations of running high resolution, uncoupled WRF simulations in mountain terriain (the CO headwaters was studied) where the lack of lateral flow caused changes in the surface energy budget and hight of the boundary layer.

Responses: The reviewer is correct that the WRF simulation used most of the computational and throughput time. However, we are unaware of any study which shows only a 1% CPU demand for ParFlow-WRF opposed to standalone WRF. Personal communication with the primary ParFlow developer confirmed this is not a strict rule of thumb, and that there are various factors which determine the additional CPU time to include ParFlow in a WRF run. This is especially true in complex terrain where there are sharp wetting fronts and higher demands on the Richards' equation solver. Thus, it's difficult to know the exact additional demands of the fully coupled code without performing the simulations.

We agree to update the statement as "one-way coupling IPM" in the revision. We also concur with the reviewer that computational expense is not the major excuse for not running coupled simulations, moreso that we wished to establish a baseline set of simulations without fully coupled feedbacks before considering the 2-way interacting model.

We agree with the reviewer to cite Forrester 2020 paper, which discusses how the subsurface groundwater flow in ParFlow-CLM (lateral groundwater flow and subsurface lower boundary conditions) affects the atmospheric model. These impacts were particularly enhanced during summertime, while we run the simulation for a whole water year. We agree to highlight the

findings from previous fully-coupled WRF-PF model, and extend the discussion of one-way and two-way coupling in the revision.

Response after revision: The text has now been revised as follows, "A limitation of our study, given the computational constraints of running IPMs, is that it was infeasible to explore the full parameter spaces of WRF and ParFlow-CLM exhaustively; thus, our conclusions are limited to the selected subgrid-scale physics schemes and meteorological forcing datasets analyzed. Additional work is needed to improve the systemic cold bias in two-meter surface air temperature throughout all experiments as this may have been the major driver in the delayed snowmelt and peak discharge simulated by the IPM. Another methodological constraint is that our WRF and Parflow-CLM experiments were only one-way instead of two-way feedbacks, which ignores potentially important feedbacks from the subsurface hydrology to the atmosphere via ET and the radiation budget. For example, Givati et al. (2016) reported that simulated precipitation was improved with two-way coupling in WRF-Hydro compared to WRF-only and Forrester et al. (2018) showed that boundary layer dynamics were impacted in IPM simulations in regions where shallow water tables exist. On the other hand, ParFlow-CLM is essential in our experiment for quantifying hydrological responses, including streamflow and groundwater storage. Although WRF-Hydro provides some insights into streamflow, it still uses a simplified and prescribed stream network. Groundwater storage in WRF-Hydro is also highly simplified by using a bucket model while ParFlow-CLM simulates the full continuum of variable saturation in three dimensions. We also recognize that a coupled version of WRF and ParFlow exists, with the capability of simulating two-way coupling between the atmospheric and hydrological processes in the surface and subsurface domain, though it was not used in this study and could be explored in future efforts. "

line 498: Is the watershed highly instrumented now or will it be? This seems at odds with statement in the abstract (line 16)?

Response: To clarify, the East River watershed is already highly instrumented due to the presence of the long-standing Rocky Mountain Biological Laboratory (RMBL), the SNOTEL network, and DOE Watershed Science Focus Area project which has been adding instrumentation to the watershed over the last ~7 years. While these observations focus primarily on surface and subsurface processes, the East River watershed has become even more instrumented in recent years (2021-2023) through the support of the U.S. DOE (SAIL campaign) and U.S. NOAA (SPLASH campaign) deployments of a comprehensive set of atmospheric instrumentations (e.g., radar and radiation measurements).

However, these SAIL and SPLASH campaign measurements were provided after our simulations were conducted and we began to prepare this manuscript. In future work, we plan to build upon the knowledge learned from this manuscript to compare the most optimally configured IPM to SAIL and SPLASH campaign observations. We can revise this sentence to best reflect the added significance of SAIL and SPLASH campaigns in future IPM studies in the UCRB region, and look forward to using those datasets to improve mountainous hydrologic cycle process understanding and model development.

Response after revision: The last paragraph of the conclusion section has been revised to "The East River watershed is already highly instrumented due to the presence of the long-standing Rocky Mountain Biological Laboratory (RMBL), the SNOTEL network, the United States Geological Survey's Next Generation Water Observing System (NGWOS), the National Science Foundation's Sublimation of Snow (SOS) project, and DOE Watershed Science Focus Area project which has been adding instrumentation to the watershed over the last ~7 years. While these observations focus primarily on surface and subsurface processes, the East River watershed has become even more instrumented in recent years (2021-2023) through the support of the U.S. DOE (SAIL campaign) and U.S. NOAA (SPLASH campaign) deployments of a comprehensive set of atmospheric instrumentations (e.g., radar and radiation measurements). Future work will include integration of data, either indirectly through IPM benchmarking or directly through data assimilation into the IPM, from the SAIL campaign. . SAIL is collecting a wide-array of observations with the intent to advance understanding of precipitation, snow, aerosol, aerosol-cloud interaction, and radiation processes in complex terrain and establish the minimum-but-sufficient level of process understanding to develop a robust predictive understanding of seasonal surface water and energy budgets in the ERW (Feldman et al., 2021). SAIL aims to develop a wide range of hydrometeorological datasets to constrain atmosphere, surface, and subsurface processes simultaneously. Together, these resources are contributing to the establishment of a highly-instrumented and studied UCRB watershed. We look forward to building upon the knowledge learned from this manuscript to compare the most appropriately configured IPM to SAIL and SPLASH campaign observations. Our study highlights that the benchmarking provided by these data collections will be critical in addressing the systemic IPM cold bias by providing a more constrained estimate of radiation budgets in complex terrain that ultimately shape snowmelt and discharge."

references
Arnault, J., Wagner, S., Rummler, T., Fersch, B., Bliefernicht, J., Andresen, S., & Kunstmann, H, 2016: Role of Runoff–Infiltration Partitioning and Resolved Overland Flow on Land–Atmosphere Feedbacks: A Case Study with the WRF-Hydro Coupled Modeling System for West Africa, Journal of Hydrometeorology
Ban, N., Schmidli, J., and Schär, C. 2014: Evaluation of the convection-resolving regional climate modeling approach in decade-long simulations, Journal Geophyscal Research Atmosphere
Forrester, M. and Maxwell, R. 2020: Impact of lateral groundwater flow and subsurface lower boundary conditions on atmospheric boundary layer development over complex terrain, Journal of Hydrometeorology
Frei, C. and Schär, C. 1998: A precipitation climatology of the Alps from high-resolution rain-gauge observations. Inter ational Journal of Climatology,
Keune, J., Gasper, F., Goergen, K., Hense, A., Shrestha, P., Sulis, M. and Kollet, S. 2016: Studying the influence of groundwater representations on land surface-atmosphere feedbacks during the European heat wave in 2003, Journal of Geophysical Research - Atmospheres

Keune, J., Sulis, M., and Kollet, S. J. 2019: Potential added value of incorporating human water use on the simulation of evapotranspiration and precipitation in a continental  scale bedrock  to   atmosphere modeling system–A validation study considering observational uncertainty. Journal of Advances in Modeling Earth Systems

Miguez-Macho, G., Y. Fan, C. P. Weaver, R. Walko, and A. Robock, 2007: Incorporating water table dynamics in climate modeling: 2. Formulation, validation, and soil moisture simulation. Journal of Geophysical Research

Maxwell, R.,  J. K. Lundquist, J. D. Mirocha, S. G. Smith, C. S. Woodward, and A. F. B. Tompson, 2011: Development of a coupled groundwater–atmosphere model, Monthly Weather Review

Ryken, A. C., Gochis, D., & Maxwell, R. 2022: Unravelling groundwater contributions to evapotranspiration and constraining water fluxes in a high-elevation catchment. Hydrological Processes

Walser, A., and C. Schaer, 2004: Convection-resolving precipitation forecasting and its predictability in Alpine river catchments, Journal of Hydrology

---

## Referee Report (RR1)

The authors have done a very thorough job revising the manuscript, and it is much improved. I have only minor comments before I believe the manuscript will be ready for publication.

**Major comments:**
I find the phrase "subgrid-scale physics schemes" a little general. For a coupled model, this could refer to the representation of hydrology in ParFlow-CLM (or other hydrologic models), but in this study it refers to physics schemes within WRF. In the abstract (or even the title), could you clarify that the physics schemes here refer to the physics within WRF?

I also have one lingering comment about the poor agreement between streamflow simulations and observations, as well as the related temperature bias. The poor streamflow fit suggests that even if regional climate models like WRF outperform observations with respect to precipitation, a temperature bias could confound this with respect to their ability to act as accurate hydrologic model forcings. I think it would be valuable to mention as early as the abstract (and possibly expand in discussion) that all the WRF configurations have a temperature bias relative to PRISM that's comparable to what we would expect due to climate change impacts. Maybe that will just tell readers that bias correction is still important, but it seems like information that should be available early.

**Page 1**
Line 21 – should be "uncertainty from synoptic-scale forcings…"?

Line 25 – "delayed earlier" – This is a contradiction in my view. Were flows delayed or earlier?

**Page 4**
Line 101 – "However, neither… do not…" Did you intend the double-negative here?

**Page 18**
Table 3 – The standout conclusion from this table, in my view, is that all models have pretty terrible $R^2$ of precipitation relative to PRISM. Is this due to stochasticity in the daily time series generation, or something else? If it's event-scale stochasticity, would a different metric be more appropriate?

Figure 4 provides a time-averaged spatially explicit comparison between PRISM and modeled temperature; why don't we have the same thing provided for precipitation?

**Page 19**
First line of page - Based on Table 3, NCAR-CFSR seems like a more attractive fit to PRISM than the BSU simulations. The temperature $R^2$ is slightly worse, but the precipitation is substantially better. Are there other factors that made the BSU simulations better?

---

## Author Response (AR2)

The authors have done a very thorough job revising the manuscript, and it is much improved. I have only minor comments before I believe the manuscript will be ready for publication.

Major comments:

I find the phrase "subgrid-scale physics schemes" a little general. For a coupled model, this could refer to the representation of hydrology in ParFlow-CLM (or other hydrologic models), but in this study it refers to physics schemes within WRF. In the abstract (or even the title), could you clarify that the physics schemes here refer to the physics within WRF?
Response: We added "in WRF" on line 19 and line 23 of the abstract.

I also have one lingering comment about the poor agreement between streamflow simulations and observations, as well as the related temperature bias. The poor streamflow fit suggests that even if regional climate models like WRF outperform observations with respect to precipitation, a temperature bias could confound this with respect to their ability to act as accurate hydrologic model forcings. I think it would be valuable to mention as early as the abstract (and possibly expand in discussion) that all the WRF configurations have a temperature bias relative to PRISM that's comparable to what we would expect due to climate change impacts. Maybe that will just tell readers that bias correction is still important, but it seems like information that should be available early.
Response: We agree with the reviewer and revised the abstract as "Despite reasonably simulating precipitation, a delay in simulated peak discharge is due to a systematic cold bias across WRF simulations, suggesting the need for bias correction."

Line 21 – should be "uncertainty from synoptic-scale forcings…"?
Response: Revised as suggested.

Line 25 – "delayed earlier" – This is a contradiction in my view. Were flows delayed or earlier?
Response: Deleted "earlier"

Line 101 – "However, neither… do not…" Did you intend the double-negative here?
Response: Yes, we removed the "do not"

Table 3 – The standout conclusion from this table, in my view, is that all models have pretty terrible $R_2$ of precipitation relative to PRISM. Is this due to stochasticity in the daily time series generation, or something else? If it's event-scale stochasticity, would a different metric be more appropriate?
Response: We agree with the reviewers that the $R_2$ of precipitation relative to PRISM are relatively low. The misfit is mostly due to overestimation of a few precipitation events from February through April. In the revision, we added the 95th percentile of daily precipitation in Table 3 to analyze the event-scale stochasticity of precipitation simulation.

The text on line 393 are updated as "Although NCAR-CFSR has a higher R2 than other simulations, NCAR-ERA5 has a very low R2.  The BSU simulations provide a closer approximation of cumulative precipitation to PRISM.  Specifically, BSU does better in simulating extreme precipitation events (i.e., 95th percentile).  Therefore, we conclude that BSU WRF subgrid-scale physics schemes outperform the UCD and NCAR WRF subgrid-scale physics schemes in simulating both precipitation and temperature."

Figure 4 provides a time-averaged spatially explicit comparison between PRISM and modeled temperature; why don't we have the same thing provided for precipitation?
Response: I'm not sure if I understand the reviewer's comment or the reviewer is missing anything. The time-averaged spatially explicit comparison between PRISM and modeled precipitation is in Figure 4b).

First line of page - Based on Table 3, NCAR-CFSR seems like a more attractive fit to PRISM than the BSU simulations. The temperature $R_2$ is slightly worse, but the precipitation is substantially better. Are there other factors that made the BSU simulations better?
Response: While the NCAR-CFSR has a higher R2 than others, but the NCAR-ERA5 has a very low R2 so we are concerning about its capabilities under the uncertainties of various large-scale meteorological forcings datasets. Although the BSU scenarios have relative lower R2 of precipitation, the cumulative precipitations are consistently closer to the PRISM. Also compared to BSU, the NCAR scenarios have relatively severe colder bias, which is one of the major concerns identified in this study. Overall, we decided to choose the BSU scheme.

The text on line 393 are updated as "Although NCAR-CFSR has a higher R2 than other simulations, NCAR-ERA5 has a very low R2.  The BSU simulations provide a closer approximation of cumulative precipitation to PRISM.  Specifically, BSU does better in simulating extreme precipitation events (i.e., 95th percentile).  Therefore, we conclude that BSU WRF subgrid-scale physics schemes outperform the UCD and NCAR WRF subgrid-scale physics schemes in simulating both precipitation and temperature."